# Near-Minimax Multi-Objective RL under Predictable Adversarial Preferences and Preference-Free Exploration in Linear MDPs

**Mingxi Hu** [1]  **Meiling Yu** [2]

## Abstract

Multi-objective reinforcement learning (MORL) must often support preferences that change online or are specified only after data collection. We study finite-horizon MORL with vector feedback in linear MDPs under two protocols: (i) predictable adversarial preferences revealed before each episode, and (ii) reward-free preference-free exploration (PFE), where exploration observes only transitions and must later answer arbitrary preference queries. Standard reductions are protocol-unsafe: re-scalarizing past stochastic rewards with future weights breaks the martingale structure needed for self-normalized confidence bounds, and hypervolume evaluation must account for episode-start randomization, which yields a deployable convex hull of return vectors. We propose a protocol-safe reward interface that estimates each reward coordinate via regression and performs scalarization only at query time, and we formalize deployable hypervolume semantics with a stability chain from support-function error to hypervolume error. Consequently, we obtain filtration-safe regret bounds for any predictable preference sequence without discretizing the simplex (only $\log m$ dependence) and matching near-minimax rates in linear MDPs, as well as sharp reward-free PFE guarantees: a (near-)minimax decision-optimal query answering rate $\tilde{O}(d^2 U_{\mathrm{ret}}^2/\varepsilon^2)$ and a tight separation from explicit transition-model recovery $\Theta(d(|\mathcal{S}| - 1)/\varepsilon_P^2)$. These results connect online learning, preference-free deployment, and hypervolume-aware evaluation through a single protocol-aligned theory.

[1]School of Data Science, Fudan University [2]College of Artificial Intelligence, Nankai University. Correspondence to: Mingxi Hu <23110980005@m.fudan.edu.cn>, Meiling Yu <1120230250@mail.nankai.edu.cn>.

*Proceedings of the 43$^{rd}$ International Conference on Machine Learning*, Seoul, South Korea. PMLR 306, 2026. Copyright 2026 by the author(s).

## 1. Introduction

Multi-objective reinforcement learning (MORL) extends reinforcement learning to vector-valued feedback, where each action yields an $m$-dimensional reward capturing competing criteria such as utility, risk, and resource usage (Roijers et al., 2013; Hayes et al., 2022). A common way to operationalize preferences is *linear scalarization*: a weight vector $w \in \Delta_m$ defines the scalar reward $\langle w, r \rangle$, and learning aims to support many possible weights. In many applications, however, preferences are not fixed: an online system may face a sequence of users with different utilities, and offline data may need to support weights chosen only *after* data collection.

We study episodic finite-horizon MORL in linear MDPs (Jin et al., 2020b) under two protocols that make this uncertainty explicit. (i) *Predictable* adversarial preferences: before episode $k$, an adversary chooses $w_k$ based on the past but not on episode-$k$ randomness (Figure 1), extending the "picky customers" setting (Wu et al., 2021) to unknown reward models. (ii) *Reward-free* preference-free exploration (PFE): exploration observes only transitions (no rewards) and must later answer arbitrary preference queries once the corresponding *mean* scalar reward is revealed (Jin et al., 2020a; Wagenmaker et al., 2022). A guiding viewpoint throughout is to treat preference as a *query-time object*: interaction learns preference-agnostic quantities (reward coordinates or transition structure), and preferences enter only when solving a scalarized planning or evaluation problem. A concise reading map is: M1 asks how to learn safely when the preference can change from episode to episode; M2 asks which return set is deployable after learning; and M3 asks what must be learned during reward-free exploration to answer future preference queries.

Protocol alignment matters: several standard MORL shortcuts are unsafe or ill-posed under these protocols. First (C1), post-hoc re-scalarization of past stochastic rewards with a future weight is not filtration-safe and can break the martingale structure needed for self-normalized confidence transfer. Second (C2), deployment permits episode-start randomization, so the set of deployable return vectors is a convex hull; consequently, hypervolume should be evaluated on $\mathrm{HV}(\mathrm{conv}(\cdot))$ (Zitzler et al., 2007). Third (C3), in

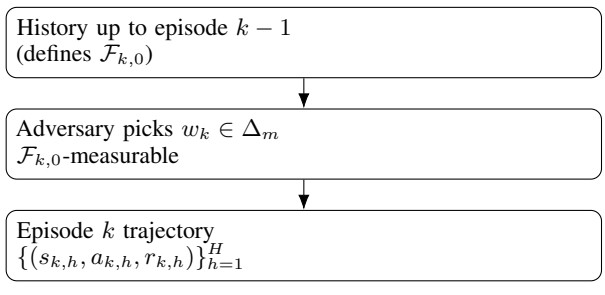

History up to episode $k-1$
(defines $\mathcal{F}_{k,0}$)

$\downarrow$

Adversary picks $w_k \in \Delta_m$
$\mathcal{F}_{k,0}$-measurable

$\downarrow$

Episode $k$ trajectory
$\{(s_{k,h}, a_{k,h}, r_{k,h})\}_{h=1}^{H}$

Pitfall: using a future weight to re-scalarize past noise destroys MDS / sub-Gaussianity.

*Figure 1.* Filtration timeline. Predictability means $w_k$ is $\mathcal{F}_{k,0}$-measurable (fixed before episode $k$). Post-hoc scalarization of past noise with a future weight breaks martingale arguments; Appendix D.1 gives a counterexample. Our protocol-safe reward interface estimates reward coordinates and combines them with the queried $w_k$ only when planning for episode $k$.

reward-free PFE, uniform decision-optimal query answering can avoid explicit transition-model recovery. Our goal is near-minimax statistical guarantees in linear MDPs with constants and dependencies that are valid under the stated protocols.

**Contributions.** We build on existing near-minimax optimistic planning in linear MDPs and reward-free exploration primitives; our *new* technical contributions are: (M1) a protocol-safe reward interface—coordinate-wise reward regression with query-time scalarization—and a preference-safe wrapper around optimistic linear-MDP planners, yielding filtration-safe regret bounds for any predictable preference sequence without discretizing $\Delta_m$ and with only a $\log m$ overhead. (M2) a deployable hypervolume semantics based on $\mathrm{HV}(\mathrm{conv}(\cdot))$ together with a stability chain from uniform support-function error to hypervolume error, enabling auditable hypervolume certificates and quantifying the weight-discretization curse. (M3) under reward-free feedback, (near-)minimax rates for decision-optimal PFE and matching bounds for explicit-model output, yielding an information-theoretic separation and a bridge from uniform query answering to deployable hypervolume certification.

**Roadmap.** Section 2 discusses protocol-aligned prior work; Section 3 defines the model; Section 4 develops deployable hypervolume tools; Section 5 presents protocol-safe algorithms; and Section 6 states the main theorems (proofs in the appendix).

## 2. Related Work & Protocol Alignment

We briefly relate our three main lines (M1–M3) to prior work, emphasizing that guarantees depend on the interaction protocol: what feedback is observed, when preferences are revealed, and what is required at deployment.

For online learning with changing preferences, the closest protocol is predictable per-episode scalarization. The "picky customers" model of Wu et al. (2021) studies adversarial per-episode preferences in tabular MDPs but assumes the multi-objective reward functions are known. Our setting differs in that the reward coordinates are unknown and must be estimated jointly with the transition model in a linear MDP. Thus the new technical issue is not scalar planning itself, but maintaining filtration-safe reward confidence transfer when $w_k$ adapts to the past: we learn stable reward coordinates from vector feedback, introduce $w_k$ only at episode start, and replan for that episode. On the dynamics side, near-minimax regret for linear MDPs with a known feature map is well understood (Jin et al., 2020b; He et al., 2023); our contribution is a protocol-safe reward interface that wraps such optimistic planners while remaining valid under predictable preferences. Other objectives with "adversarial weights" (e.g., max–min fairness) optimize a single stationary criterion, and dynamic-regret analyses for nonstationary rewards address reward drift rather than preference shifts; these protocols do not capture the filtration issues central to M1.

Preference-free exploration (PFE) under reward-free feedback is closely related to reward-free (goal-free) exploration (Jin et al., 2020a; Wagenmaker et al., 2022), which collects transition data without reward observations and later plans once a reward function is revealed. Our decision-optimal PFE results are stated in this protocol and include matching lower bounds; we further prove a tight separation showing that explicit base-kernel recovery is strictly harder (in its unavoidable $|\mathcal{S}|$ dependence) than uniform decision-optimal query answering, making the gap information-theoretic (M3).

Hypervolume is a standard Pareto-quality indicator (Zitzler et al., 2007) and is widely used in MORL benchmarks and toolkits (Felten et al., 2023). Prior work studies hypervolume optimization and hypervolume-driven learning objectives; our focus is complementary: in RL, episode-start randomization yields a deployable convex hull of return vectors, motivating evaluation on $\mathrm{HV}(\mathrm{conv}(\cdot))$ and stability certificates that translate uniform support-function error into hypervolume error (M2).

## 3. Model and Protocols

We study episodic finite-horizon MORL with an $m$-dimensional reward vector and unknown dynamics, and we fix two interaction protocols that make preference uncertainty explicit. In both cases we treat preference as a *query-time object*: interaction learns preference-agnostic quantities (reward coordinates or transition structure), and a preference enters only when forming a scalarized planning/evaluation problem.

### 3.1. Episodic MORL and scalarization

We consider an episodic MDP with horizon $H$, finite state space $\mathcal{S}$, finite action space $\mathcal{A}$, and an (unknown but fixed) initial state distribution $\rho$. At stage $h$, after taking action $a$ in state $s$, the next state is drawn from $P_h(\cdot \mid s, a)$ and a reward vector $r_h(s, a) \in [0, 1]^m$ is sampled. We denote the conditional mean by $\bar{r}_h(s, a) := \mathbb{E}[r_h(s, a)] \in [0, 1]^m$ and write $\bar{r}_{h,i}$ for objective $i \in [m]$. A policy $\pi$ induces the return vector

$$v(\pi) = \mathbb{E}_\pi \Big[ \sum_{h=1}^H r_h(s_h, a_h) \Big] \in [0, H]^m,$$

so for a preference $w \in \Delta_m$ the scalarized value is $V^\pi(w) = \langle w, v(\pi) \rangle$ and $V^*(w) = \sup_\pi V^\pi(w)$.

**Definition 3.1** (Simplex of preferences). The preference simplex is $\Delta_m := \{ w \in \mathbb{R}^m : w_i \geq 0 \; \forall i, \; \sum_{i=1}^m w_i = 1 \}$.

**Definition 3.2** (Return scale). We write $U_{\mathrm{ret}} \geq \sup_\pi \|v(\pi)\|_\infty$ for a known return-scale bound; under $r_h(s, a) \in [0, 1]^m$ one may take $U_{\mathrm{ret}} = H$.

### 3.2. Protocols and objectives

*Online vector feedback (M1).* At episode $k$, a preference $w_k \in \Delta_m$ is revealed *before* the episode begins and remains fixed within the episode. The learner observes realized reward vectors along the trajectory:

**Assumption 3.3** (Vector feedback). In episode $k$ and step $h$, after choosing $a_{k,h}$, the learner observes the next state $s_{k,h+1}$ and the full reward vector $r_{k,h} \in [0, 1]^m$.

We assume bounded martingale noise for each coordinate:

**Assumption 3.4** (Bounded reward noise). For each stage $h \in [H]$ and objective $i \in [m]$, the observed reward coordinate satisfies

$$r_{k,h,i} = \bar{r}_{h,i}(s_{k,h}, a_{k,h}) + \xi_{k,h,i},$$

where $(\xi_{k,h,i})$ is a bounded martingale difference sequence w.r.t. the episode–step filtration $\{\mathcal{F}_{k,h}\}$: $|\xi_{k,h,i}| \leq 1$ almost surely and $\mathbb{E}[\xi_{k,h,i} \mid \mathcal{F}_{k,h}] = 0$.

Predictability captures the protocol constraint that the preference sequence cannot depend on within-episode randomness.

**Definition 3.5** (Predictable preference sequence). Let $\mathcal{F}_{k,0}$ be the $\sigma$-field generated by the interaction history up to the end of episode $k-1$. A sequence $(w_k)_{k=1}^K$ with $w_k \in \Delta_m$ is *predictable* if each $w_k$ is $\mathcal{F}_{k,0}$-measurable. Equivalently, $w_k$ may adapt to past trajectories/returns but cannot depend on episode-$k$ randomness (including $s_{k,1}$).

For predictable $(w_k)$, the online regret is

$$\mathrm{Reg}(K) := \sum_{k=1}^K \big( V^*(w_k) - V^{\pi_k}(w_k) \big).$$

*Reward-free preference-free exploration (M3).* Exploration runs without observing rewards; after exploration, queries reveal the *mean* scalar reward corresponding to the queried preference.

**Definition 3.6** (Reward-free feedback model for PFE). In the PFE protocol, the learner interacts with the MDP for $N$ exploration episodes *without observing rewards*: after choosing $a_{k,h}$ it observes only the next state $s_{k,h+1}$. After exploration, an arbitrary (possibly adaptive) sequence of preference queries $w \in \Delta_m$ may be issued; upon each query, the corresponding mean scalar reward function

$$\bar{r}_h^w(s, a) := \langle w, \bar{r}_h(s, a) \rangle$$

is revealed and the learner must output a policy $\hat{\pi}(w)$ without further interaction.

Thus "reward-free" and "preference-free" refer only to the exploration stage of M3; they do not describe the online vector-feedback protocol M1, where preferences and reward vectors are observed during learning. We will contrast decision-optimal query answering with the stronger requirement of explicit base-kernel recovery (defined in Section 6.3).

### 3.3. Linear MDPs

We work with the finite-horizon *linear MDP* model (Jin et al., 2020b). For each stage $h \in [H]$, a known feature map $\phi_h : \mathcal{S} \times \mathcal{A} \to \mathbb{R}^d$ satisfies $\|\phi_h(s, a)\|_2 \leq 1$. There exist unknown parameters $(\theta_{h,i})_{i \in [m]}$ and $(\mu_h(s'))_{s' \in \mathcal{S}}$ such that, for all $(s, a) \in \mathcal{S} \times \mathcal{A}$,

$$\bar{r}_{h,i}(s, a) = \langle \phi_h(s, a), \theta_{h,i} \rangle \in [0, 1],$$
$$P_h(s' \mid s, a) = \langle \phi_h(s, a), \mu_h(s') \rangle,$$

and $P_h(\cdot \mid s, a)$ is a valid distribution. Equivalently, for any bounded $V : \mathcal{S} \to \mathbb{R}$,

$$[P_h V](s, a) = \langle \phi_h(s, a), \mu_h(V) \rangle,$$
$$\mu_h(V) := \sum_{s' \in \mathcal{S}} \mu_h(s') V(s'),$$

with the standard boundedness condition $\|\mu_h(V)\|_2 \leq \sqrt{d} \|V\|_\infty$. This state–action feature model differs from the triplet-feature linear mixture MDP model of Zhou et al. (2021).

*Remark* 3.7 (Anchored simplex-mixture special case). For the explicit-model requirement in Section 6.3, we additionally assume an anchored simplex structure: $\phi_h(s, a) \in \Delta_d$ and known anchor pairs $(s_{h,j}, a_{h,j})$ with $\phi_h(s_{h,j}, a_{h,j}) = e_j$. Defining $P_h^j(\cdot) := P_h(\cdot \mid s_{h,j}, a_{h,j})$ yields

$$P_h(\cdot \mid s, a) = \sum_{j=1}^d \phi_h(s, a)_j P_h^j(\cdot).$$

# 4. Deployable Hypervolume Semantics

This section provides geometric tools for turning *uniform scalarization/supported-value errors* into *deployable hypervolume* certificates. In RL, deployment can randomize policies at episode start, so the appropriate evaluation object is $\mathrm{HV}(\mathrm{conv}(\cdot))$; the stability chain below is used later to propagate query-time planning errors to HV error bounds. This hypervolume layer is used for deployment evaluation and certification, not as an additional objective optimized by the online algorithm.

## 4.1. Deployable convex hull

Let $\mathcal{V}_{\mathrm{det}}$ be the set of return vectors of deterministic Markov policies.

**Definition 4.1** (Deployable convexified return set). The deployable return set under episode-start randomization is

$$\mathcal{C}^* := \mathrm{conv}(\mathcal{V}_{\mathrm{det}}).$$

Such convexification under linear scalarization is standard in MORL (cf. convex coverage sets; (Roijers et al., 2013)).

**Lemma 4.2** (Episode-start randomization realizes the convex hull). *If, at the start of each episode, the learner samples a deterministic Markov policy from an arbitrary distribution over such policies and then executes it for the whole episode, the resulting expected return vector lies in $\mathcal{C}^* = \mathrm{conv}(\mathcal{V}_{\mathrm{det}})$. Conversely, every point in $\mathcal{C}^*$ can be realized as the expected return of such an episode-start randomization.*

The proof is a direct linearity-of-expectation argument and is deferred to Appendix C.2.

## 4.2. Hypervolume and why convexification matters

**Definition 4.3** (Dominated region and hypervolume (reference point 0)). For $X \subset \mathbb{R}^m_+$, define its dominated region

$$D_0(X) = \{ y \in \mathbb{R}^m_+ : \exists x \in X \text{ with } y \leq x \},$$

where $\leq$ is coordinate-wise. The hypervolume is $\mathrm{HV}(X) := \mathrm{Vol}(D_0(X))$ (Zitzler et al., 2007).

**Proposition 4.4** (Counterexample: convexification can change hypervolume). *Let $X := \{(1,0),(0,1)\}$, a subset of $\mathbb{R}^2_+$. With reference point 0, $\mathrm{HV}(X) = 0$, whereas $\mathrm{HV}(\mathrm{conv}(X)) = 1/2$.*

This example highlights that $\mathrm{HV}(\cdot)$ is not deployment-invariant under convexification; our evaluation and certificates therefore target $\mathrm{HV}(\mathcal{C}^*)$.

## 4.3. Support functions and the stability chain

**Definition 4.5** (Support function). For a compact set $C \subset \mathbb{R}^m$, its support function is $h_C(w) = \sup_{x \in C} \langle w, x \rangle$.

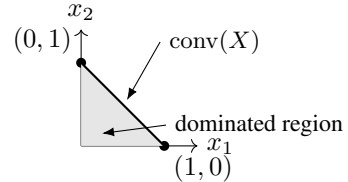

*Figure 2.* In 2D, $X = \{(1,0),(0,1)\}$ dominates only a measure-zero set, but $\mathrm{conv}(X)$ dominates a triangle of area $1/2$, hence $\mathrm{HV}(\mathrm{conv}(X)) = 1/2$ w.r.t. reference point 0.

**Lemma 4.6** (Support values equal optimal scalarized values). *For the deployable set $\mathcal{C}^* = \mathrm{conv}(\mathcal{V}_{\mathrm{det}})$ and any $w \in \Delta_m$,*

$$h_{\mathcal{C}^*}(w) = \max_{v \in \mathcal{V}_{\mathrm{det}}} \langle w, v \rangle = V^*(w).$$

The proof is standard (linearity over convex hull and existence of optimal deterministic Markov policies) and is given in Appendix C.2.

We use the stability chain *support error on $\Delta_m$ $\Rightarrow$ dominated-region Hausdorff-$\infty$ error $\Rightarrow$ hypervolume error* (Figure 3).

**Definition 4.7** ($\ell_\infty$ Hausdorff distance). For sets $A, B \subset \mathbb{R}^m$, let $\mathrm{dist}_\infty(x, B) := \inf_{y \in B} \|x - y\|_\infty$ and

$$d_{H,\infty}(A, B) := \max \Big\{ \sup_{x \in A} \mathrm{dist}_\infty(x, B),$$
$$\sup_{y \in B} \mathrm{dist}_\infty(y, A) \Big\}.$$

**Definition 4.8** (Downward-closed set). A set $A \subset \mathbb{R}^m_+$ is *downward-closed* if $x \in A$ and $0 \leq y \leq x$ imply $y \in A$; in particular, dominated regions $D_0(\cdot)$ from Definition 4.3 are downward-closed.

**Theorem 4.9** (Support error implies dominated-region Hausdorff-$\infty$ error). *Let $K, K' \subset \mathbb{R}^m_+$ be compact and convex. If*

$$\sup_{w \in \Delta_m} |h_K(w) - h_{K'}(w)| \leq \varepsilon,$$

*then $d_{H,\infty}(D_0(K), D_0(K')) \leq \varepsilon$.*

**Theorem 4.10** (Hypervolume is Lipschitz in Hausdorff-$\infty$ for downward-closed sets). *Let $A, B \subset [0, U]^m$ be compact, convex, and downward-closed. Then*

$$|\mathrm{HV}(A) - \mathrm{HV}(B)| \leq m\, U^{m-1}\, d_{H,\infty}(A, B).$$

# 5. Protocol-safe Algorithms

Sections 3–4 fixed the protocol (predictable preferences + linear MDPs) and the deployable evaluation target $\mathrm{HV}(\mathcal{C}^*)$

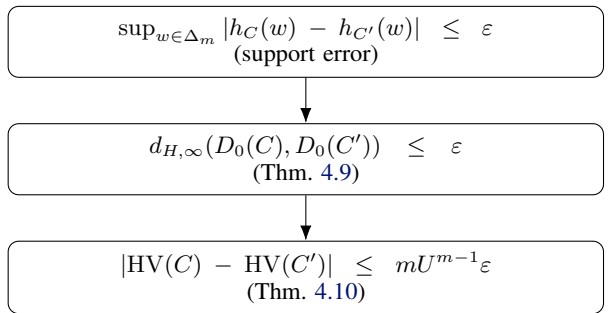

$$\sup_{w \in \Delta_m} |h_C(w) - h_{C'}(w)| \leq \varepsilon$$
(support error)

$$\downarrow$$

$$d_{H,\infty}(D_0(C), D_0(C')) \leq \varepsilon$$
(Thm. 4.9)

$$\downarrow$$

$$|\mathrm{HV}(C) - \mathrm{HV}(C')| \leq mU^{m-1}\varepsilon$$
(Thm. 4.10)

*Figure 3.* The hypervolume stability chain used for error propagation. Proofs of Theorems 4.9 and 4.10 are in Appendix C.

together with the stability chain in Figure 3. We now describe the algorithms that realize these objects. Algorithm 1 is the sole online interaction algorithm for M1; the rest of the paper analyzes it and uses the stability chain to turn scalar planning errors into deployable hypervolume certificates. We also summarize the reward-free PFE subroutines used in M3.

### 5.1. Online: a protocol-safe wrapper around BASEPLANNER

Our online algorithm (Algorithm 1) is a MORL wrapper around any *episodic* optimistic planner for linear MDPs (e.g., LSVI-UCB; (Jin et al., 2020b)). The wrapper isolates preference handling from exploration: it learns each reward coordinate from vector feedback, then uses the revealed $w_k$ only at episode start to synthesize an optimistic scalar reward oracle for BASEPLANNER. This avoids post-hoc scalarization of past noise (pitfall P1) and yields only a $\log m$ overhead in the confidence transfer. (Here "++" refers to our preference-safe reward interface, not the low-switching LSVI-UCB++ of He et al. (2023), which is not directly compatible with arbitrary preference changes.)

**Coordinate-wise ridge regression and query-time synthesis.** For each stage $h$ and objective $i$, the linear reward model in Section 3.3 implies $r_{\tau,h,i} \approx \langle \phi_h(s_{\tau,h}, a_{\tau,h}), \theta_{h,i} \rangle$. We therefore compute at the start of episode $k$ the ridge estimate

$$\hat{\theta}_{k,h,i} := \underset{\theta \in \mathbb{R}^d}{\arg\min} \sum_{\tau < k} \big(\langle x_{\tau,h}, \theta \rangle - r_{\tau,h,i}\big)^2 + \lambda\|\theta\|_2^2,$$

where $x_{\tau,h} := \phi_h(s_{\tau,h}, a_{\tau,h})$. Its closed form is exactly the update in Algorithm 1: $\hat{\theta}_{k,h,i} = \Lambda_{k,h}^{-1} b_{k,h,i}$ with $\Lambda_{k,h} = \lambda I + \sum_{\tau < k} x_{\tau,h} x_{\tau,h}^\top$ and $b_{k,h,i} = \sum_{\tau < k} x_{\tau,h} r_{\tau,h,i}$. Given the predictable preference $w_k$, we then combine *parameters*

---

**Algorithm 1** HV-LinMDP-UCB++ (protocol-safe MORL wrapper; vector feedback)

1: **Inputs:** features $\phi_h$, regularization $\lambda$, confidence $\delta$, base optimistic planner BASEPLANNER
2: Initialize $\Lambda_{1,h} = \lambda I_d$ and $b_{1,h,i} = 0 \in \mathbb{R}^d$ for all $h \in [H], i \in [m]$
3: Initialize transition dataset $\mathcal{D}_0 \leftarrow \emptyset$
4: **for** episode $k = 1, 2, \ldots, K$ **do**
5:     $\mathcal{D}_k \leftarrow \mathcal{D}_{k-1}$
6:     Observe predictable preference $w_k \in \Delta_m$
7:     **for** each stage $h$ and objective $i$ **do**
8:         $\hat{\theta}_{k,h,i} \leftarrow \Lambda_{k,h}^{-1} b_{k,h,i}$
9:         Define $\hat{r}_{k,h,i}(s,a) = \langle \phi_h(s,a), \hat{\theta}_{k,h,i} \rangle$ and a radius $\beta_{k,h,i}^r(s,a)$
10:     **end for**
11:     Query-time combine: $\hat{\theta}_{k,h}(w_k) \leftarrow \sum_i w_{k,i} \hat{\theta}_{k,h,i}$
12:     Define scalarized reward model $\hat{r}_{k,h}^w(s,a) = \langle \phi_h(s,a), \hat{\theta}_{k,h}(w_k) \rangle$
13:     Define scalar reward radius $\beta_{k,h}^r(s,a) = \sum_i w_{k,i} \beta_{k,h,i}^r(s,a)$
14:     $\pi_k \leftarrow$ BASEPLANNER$(\mathcal{D}_{k-1}, \hat{r}_k^w, \beta_k^r, \delta)$
15:     **for** $h = 1$ to $H$ **do**
16:         Execute $a_{k,h} \sim \pi_k(\cdot \mid s_{k,h}, h)$; observe $s_{k,h+1}$ and full $r_{k,h} \in [0,1]^m$
17:         $\mathcal{D}_k \leftarrow \mathcal{D}_k \cup \{(s_{k,h}, a_{k,h}, s_{k,h+1})\}$
18:         $x_{k,h} \leftarrow \phi_h(s_{k,h}, a_{k,h})$
19:         $\Lambda_{k+1,h} \leftarrow \Lambda_{k,h} + x_{k,h} x_{k,h}^\top$
20:         **for** $i = 1$ to $m$ **do**
21:             $b_{k+1,h,i} \leftarrow b_{k,h,i} + x_{k,h} r_{k,h,i}$
22:         **end for**
23:     **end for**
24: **end for**

---

at query time:

$$\hat{\theta}_{k,h}(w_k) = \sum_{i=1}^m w_{k,i} \hat{\theta}_{k,h,i},$$

$$\hat{r}_{k,h}^w(s,a) = \langle \phi_h(s,a), \hat{\theta}_{k,h}(w_k) \rangle.$$

Since each $\hat{\theta}_{k,h,i}$ depends only on past vector feedback, it is valid for any future predictable preference. If $\beta_{k,h,i}^r(s,a)$ are coordinate-wise radii, then by nonnegativity of $w_k$ the scalar radius

$$\beta_{k,h}^r(s,a) := \sum_{i=1}^m w_{k,i}\, \beta_{k,h,i}^r(s,a)$$

provides a valid confidence bound for the scalarized reward (Lemma C.6). We feed $(\hat{r}_k^w, \beta_k^r)$ into BASEPLANNER.

**Reward-modular BASEPLANNER interface.** We require an episodic interface: at episode start, BASEPLANNER receives the transition dataset $\mathcal{D}_{k-1}$ and an optimistic reward

specification $(\hat{r}_k^w, \beta_k^r)$, and returns a policy that is optimistic under standard confidence events. Appendix B.6 states the interface formally and verifies it for standard episodic optimistic planners.

## 5.2. PFE subroutines

For decision-optimal PFE under Definition 3.6, we treat reward-free exploration in linear MDPs as a black box (Jin et al., 2020a; Wagenmaker et al., 2022): exploration collects transition data sufficient to plan for any subsequently revealed scalar reward $\bar{r}^w$. At query time, given $w$ we run a planner with the revealed $\bar{r}^w$.

For explicit-model PFE, we estimate each base kernel $P_h^j$ and output $\{\hat{P}_h^j\}$. To isolate the information-theoretic difficulty of explicit kernel recovery (used for the separation in Section 6.3), we adopt the following standard identifiability condition and a strong anchor-sampling oracle (which only makes explicit-model guarantees easier).

**Assumption 5.1** (Anchor identifiability (anchored simplex mixture))**.** For each stage $h \in [H]$ and each index $j \in [d]$, there exists a known anchor pair $(s_{h,j}, a_{h,j})$ such that $\phi_h(s_{h,j}, a_{h,j}) = e_j$. We define the associated *base kernels* $P_h^j(\cdot) := P_h(\cdot \mid s_{h,j}, a_{h,j})$. Moreover, we assume $\phi_h(s, a) \in \Delta_d$ for all $(s, a)$ so that, for every $(s, a)$,

$$P_h(\cdot \mid s, a) = \sum_{j=1}^{d} \phi_h(s, a)_j \, P_h^j(\cdot)$$

is a convex mixture of these base kernels.

**Assumption 5.2** (Anchor sampling access)**.** For each $(h, j)$, the exploration protocol can query i.i.d. samples $s_{h+1} \sim P_h^j(\cdot)$ (e.g., via simulator resets). One exploration episode consists of $H$ such anchor transition queries (one per stage).

We estimate each $P_h^j$ via multinomial concentration at its anchor and extend to all $(s, a)$ by linearity of the mixture (Theorem 6.14). The next section states the theoretical guarantees.

## 6. Main Results

We state the main guarantees for the three main lines fixed in Section 1: (M1) filtration-safe online learning under predictable preferences, (M2) deployable hypervolume semantics and calibrated certificates, and (M3) reward-free preference-free exploration (PFE) with a decision-optimal vs. explicit-model separation. Proofs are deferred to Appendix C, with standard tools collected in Appendix B.

### 6.1. M1: Filtration-safe online regret under predictable preferences

Pitfall (P1) is that preferences can adapt to the past, so post-hoc scalarization of earlier stochastic rewards using future weights can break the martingale structure needed by optimistic RL. Our wrapper avoids this by learning coordinate rewards from vector feedback and introducing the predictable weight only at episode start via query-time parameter synthesis (Section 5).

**Theorem 6.1** (Predictable preferences are filtration-safe (time-uniform))**.** *Let $\{\xi_t\}$ be an $m$-dimensional martingale difference sequence (MDS) adapted to $\{\mathcal{F}_t\}$ with each coordinate in $[-1, 1]$. Let $\{w_t\}$ be predictable with $w_t \in \Delta_m$ and $w_t$ measurable w.r.t. $\mathcal{F}_{t-1}$. Then the scalarized noise $\eta_t := \langle w_t, \xi_t \rangle$ is a scalar MDS and is conditionally 1-subgaussian. Consequently, for any predictable regressors $\{x_t\} \subset \mathbb{R}^d$ with $\|x_t\|_2 \le 1$, the standard time-uniform self-normalized inequality applies to $\{(x_t, \eta_t)\}$ (Appendix B.3).*

Theorem 6.1 yields a confidence transfer from coordinate-wise regression to the scalarized reward oracle used by Algorithm 1, with only a $\log m$ overhead and without discretizing $\Delta_m$.

**Theorem 6.2** (Online regret; no discretization of $\Delta_m$)**.** *Under Assumption 3.3 and the linear MDP model, Algorithm 1 instantiated with the episodic* `LSVI-UCB` *planner for linear MDPs (Jin et al., 2020b) satisfies, with probability at least $1 - \delta$,*

$$\mathrm{Reg}(K) \le \widetilde{\mathcal{O}}\big(d^{3/2} H^2 \sqrt{K}\big),$$

*where dependence on $m$ enters only logarithmically via the $Hm$ coordinate regressions. More generally, the wrapper is compatible with any episodic optimistic* BASEPLANNER *that recomputes its optimistic value functions each episode under the provided reward oracle (Appendix B.6).*

**Corollary 6.3** (Near-minimax instantiation via an episodic linear-MDP planner)**.** *If* BASEPLANNER *is instantiated with an episodic near-minimax optimistic planner for* linear MDPs *that recomputes an optimistic policy every episode (e.g.,* `LSVI-UCB` *of Hu et al. (2022)), then Algorithm 1 achieves*

$$\mathrm{Reg}(K) \le \widetilde{\mathcal{O}}\big(d\sqrt{H^3 K}\big),$$

*up to logarithmic factors and lower-order terms (see Appendix B.7).*

**Theorem 6.4** (Online regret lower bound; single-objective reduction)**.** *Fix any $d \ge 4$ and $H \ge 3$. There exist constants $c_0, c_1 > 0$ such that for any $K \ge c_0 d^2 H$, there exists a linear MDP instance in the sense of Section 3.3 (embedded as an MORL instance with $m \ge 2$ objectives by setting only the first objective nonzero) and predictable preferences $w_k \equiv e_1$ for which every algorithm satisfies*

$$\mathbb{E}[\mathrm{Reg}(K)] \ge c_1 \, d \sqrt{H^3 K}.$$

## 6.2. M2: Deployable HV(conv) certificates and the $w$-net curse

Pitfall (P2) is that deployment can convexify the return set via episode-start randomization, so evaluation and certification must target $\mathrm{HV}(\mathcal{C}^*)$ (Section 4). Combining the stability chain in Theorems 4.9 and 4.10 with a finite weight set yields an explicit propagation of per-weight planning error to deployable hypervolume error.

**Theorem 6.5** (Tightness of the HV Lipschitz constant). *For $m \geq 2$ and any $U > 0$, there exist downward-closed sets $K, K' \subset [0, U]^m$ such that $d_{H,\infty}(K, K') \leq \varepsilon$ but $|\mathrm{HV}(K) - \mathrm{HV}(K')| \geq (m/2)U^{m-1}\varepsilon$ for all $\varepsilon \in (0, U/m]$.*

**Definition 6.6** (Weight covering radius). For a finite set $\mathcal{W} \subset \Delta_m$, define its $\ell_1$ covering radius

$$\eta(\mathcal{W}) = \sup_{w \in \Delta_m} \min_{\bar{w} \in \mathcal{W}} \|w - \bar{w}\|_1.$$

**Theorem 6.7** (Discrete weights + per-weight planning error imply HV gap). *Let $\mathcal{C}^*$ be the true deployable return set and $\mathcal{C}_{\mathrm{out}}$ be an output set obtained by solving scalarized problems at weights $\mathcal{W}_L \subset \Delta_m$ up to scalarization error $\varepsilon_{\mathrm{alg}}$. Let $U = \sup_{x \in \mathcal{C}^*} \|x\|_\infty$. Then*

$$\mathrm{HV}(\mathcal{C}^*) - \mathrm{HV}(\mathcal{C}_{\mathrm{out}}) \leq mU^{m-1}\big(\varepsilon_{\mathrm{alg}} + 2U\eta(\mathcal{W}_L)\big).$$

**Theorem 6.8** (Optimal simplex covering rate under $\|\cdot\|_1$). *Assume $m \geq 2$. For any $L \geq 1$, the minimax covering radius satisfies $\inf_{|\mathcal{W}|=L} \eta(\mathcal{W}) = \Theta\big(L^{-1/(m-1)}\big)$, where the hidden constants may depend on $m$.*

The $\Theta(L^{-1/(m-1)})$ rate quantifies the $w$-net curse: achieving a target covering radius $\eta$ requires $L = \Theta(\eta^{-(m-1)})$ weights, i.e., exponential in $m$.

**Proposition 6.9** (Online regret does not imply deployable hypervolume coverage). *There exists an episodic MORL instance with predictable preferences $\{w_k\}_{k \geq 1}$ and an algorithm achieving $\mathrm{Reg}(K) = 0$ for all $K$, yet the deployable hypervolume gap remains bounded away from 0. Let $\{\pi_k\}_{k=1}^{K}$ be the executed policies and define $\hat{\mathcal{C}}_K := \mathrm{conv}(\{v(\pi_k)\}_{k=1}^{K})$. Then*

$$\mathrm{HV}(\mathcal{C}^*) - \mathrm{HV}(\hat{\mathcal{C}}_K) \geq c_0 > 0 \quad \text{for all } K.$$

The proof is in Appendix C.

## 6.3. M3: Reward-free PFE and the decision-optimal vs explicit-model separation

In the reward-free PFE protocol, exploration observes only transitions and must later answer preference queries. Pitfall (P3) is that *learning enough to decide* can be strictly easier than recovering an explicit model: decision-optimal query answering admits a near-minimax complexity that is $|\mathcal{S}|$-free, while explicit recovery of the base kernels necessarily scales with $d(|\mathcal{S}| - 1)$.

**Theorem 6.10** (PFE decision-optimal sample complexity; minimax-rate optimal up to polylog). *Fix $\delta \in (0, 1)$. Under the reward-free PFE model (Definition 3.6) in a linear MDP with return scale $U_{\mathrm{ret}}$ and dimension $d$, there exists an algorithm such that after*

$$N = \widetilde{\mathcal{O}}\big(d^2 U_{\mathrm{ret}}^2/\varepsilon^2\big)$$

*exploration episodes, with probability at least $1 - \delta$ the exploration output supports uniform query answering:*

$$\forall w \in \Delta_m : \qquad V^*(w) - V^{\hat{\pi}(w)}(w) \leq \varepsilon.$$

*A matching lower bound holds up to logarithmic factors for the reward-free feedback model.*

*Remark* 6.11 (Uniformity over queries in PFE). The guarantee in Theorem 6.10 is uniform over all future queries because the scalar reward is revealed only after exploration (Definition 3.6); this uniformity is not implied by vector-feedback online learning without a post-exploration reward reveal (Jin et al., 2020a).

**Definition 6.12** (Explicit-model PFE requirement). Fix a target total-variation accuracy $\varepsilon_P > 0$. Under the anchored simplex-mixture assumptions (Assumptions 5.1 and 5.2), an explicit-model PFE algorithm outputs estimates $\{\hat{P}_h^j\}_{h \in [H], j \in [d]}$ such that, with probability at least $1 - \delta$,

$$\max_{h \in [H], j \in [d]} \|\hat{P}_h^j - P_h^j\|_1 \leq \varepsilon_P.$$

**Theorem 6.13** (Explicit-model lower bound under anchor identifiability). *In explicit-model PFE (Definition 6.12), even under anchor identifiability, any algorithm that outputs $\{\hat{P}_h^j\}$ with $\max_{h,j} \|\hat{P}_h^j - P_h^j\|_1 \leq \varepsilon_P$ with probability at least $3/4$ requires*

$$\Omega\big(d(|\mathcal{S}| - 1)/\varepsilon_P^2\big)$$

*exploration episodes for some anchored simplex-mixture instances satisfying Assumptions 5.1 and 5.2.*

**Theorem 6.14** (Explicit-model upper bound). *Under Assumptions 5.1 and 5.2, there exists an explicit-model PFE algorithm that achieves Definition 6.12 with accuracy $\varepsilon_P$ and confidence $1 - \delta$ using*

$$N = \widetilde{\mathcal{O}}\left(\frac{d(|\mathcal{S}| - 1)}{\varepsilon_P^2}\right)$$

*exploration episodes. Moreover, choosing $\varepsilon_P = \Theta(\varepsilon/(HU_{\mathrm{ret}}))$ yields an $\varepsilon$-decision-optimal policy for any query $w$.*

**Corollary 6.15** (Deployable hypervolume certificate via PFE queries). *Assume a PFE algorithm outputs a query oracle $w \mapsto \hat{\pi}(w)$ such that, on an event of probability at least $1 - \delta$,*

$$\forall w \in \Delta_m : \qquad V^*(w) - V^{\hat{\pi}(w)}(w) \leq \varepsilon. \qquad (1)$$

*For any finite $\mathcal{W}_L \subset \Delta_m$, query the oracle on $\mathcal{W}_L$ and let*

$$\mathcal{C}_{\text{out}} := \text{conv}\Big(\{v(\hat{\pi}(\bar{w})) : \bar{w} \in \mathcal{W}_L\}\Big),$$

$$U := \sup_{x \in \mathcal{C}^*} \|x\|_\infty.$$

*Then on the same event as* (1),

$$\text{HV}(\mathcal{C}^*) - \text{HV}(\mathcal{C}_{\text{out}}) \le mU^{m-1}\big(\varepsilon + 2U\,\eta(\mathcal{W}_L)\big).$$

The proof and a discussion of how to estimate $\mathcal{C}_{\text{out}}$ under reward-free feedback are deferred to Appendix C.

# 7. Experiments

We validate the three main lines (M1–M3) on a synthetic anchored simplex-mixture benchmark (Remark 3.7), evaluating regret and value gaps by exact dynamic programming in the simulator. All comparisons use paired seeds: for each seed, all methods are evaluated on the same generated MDP instance and preference schedule. Full protocols, ablations, and additional diagnostics are deferred to Appendix A.

**E1 (M1): predictable-preference online regret.** We run the protocol-safe wrapper (Algorithm 1) with AnchorUCB under i.i.d., cyclic, and predictable-adversary preferences. We compare against (i) a $w$-net discretization baseline and (ii) a feature-aware scalar baseline that uses the same feature map and optimistic planner but learns only the scalarized reward $\langle w_k, r \rangle$ online. Figure 4 shows that our protocol-safe interface consistently reduces cumulative regret, with a larger gap under predictable adversaries; the $w$-net baseline quickly becomes throughput-limited as $m$ grows (Appendix A.4). A scaling-collapse diagnostic across $(d, H)$ is provided in Appendix A.3. In the two-policy instance of Proposition 6.9, regret can be zero while the deployable HV(conv) gap stays $1/2$ for all $K$. *Takeaway:* Coordinate-wise estimation with query-time synthesis avoids the $w$-net curse and remains stable under predictable preferences.

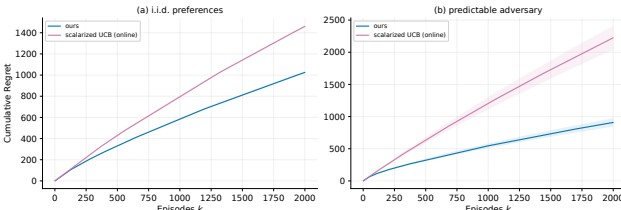

*Figure 4.* Feature-aware scalar baseline (same features and optimistic planner): our protocol-safe coordinate-wise regression + query-time scalarization vs. scalar-only online UCB regression on $\langle w_k, r \rangle$. Both methods use identical MDP instances and preference sequences for each seed; mean $\pm 95\%$ CI over 20 seeds; regret is computed by exact DP.

**E2 (M2): HV(conv) vs. HV(raw).** Consistent with Proposition 4.4, learned policy sets can exhibit ranking flips

between $\text{HV}(\cdot)$ and deployable $\text{HV}(\text{conv}(\cdot))$; see Appendix A.4. *Takeaway:* Under episode-start randomization, HV(conv) is the deployable metric; reporting only HV(raw) can be misleading.

**E3 (M3): PFE scaling and separation.** We verify the $N^{-1/2}$ scaling for decision-optimal PFE and the $|\mathcal{S}|$-dependence for explicit-model recovery; see Appendix A.7. *Takeaway:* The separation predicted by Theorems 6.10 and 6.13 is visible empirically.

**Environment-instance sweep.** To highlight performance across independently generated MDP instances from the same synthetic family ($m=10$), we repeat E1 over a fixed set of environment seeds. Figure 5 reports the distribution of final regret and deployable $\text{HV}(\text{conv}(\cdot))$; estimator details are in Appendix A.13.

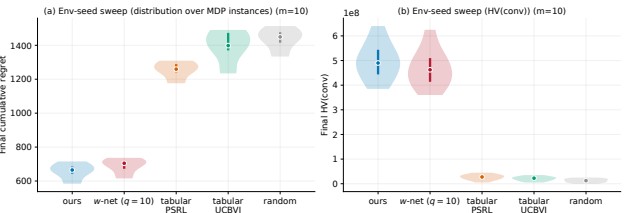

*Figure 5.* Env-instance sweep ($m=10$): distribution over independently generated MDP instances of final cumulative regret and deployable $\text{HV}(\text{conv}(\cdot))$ under i.i.d. preferences (HV(conv) via Monte Carlo; Appendix A.13).

# 8. Discussion and Limitations

**Summary.** We treat preference as a query-time object: interaction learns preference-agnostic quantities, and preferences enter only when answering a planning/evaluation query. This yields (M1) filtration-safe online regret under predictable preferences via coordinate-wise reward regression and query-time aggregation (only a $\log m$ overhead), (M2) a deployment-consistent hypervolume semantics based on $\text{HV}(\text{conv}(\cdot))$ together with stability certificates, and (M3) reward-free PFE query answering with near-minimax complexity and an information-theoretic separation from explicit-model recovery.

**Limitations.** Our strongest online guarantee in (M1) assumes vector feedback; extending filtration-safe learning to scalar-only feedback while retaining the $\log m$ dependence is open. The theory targets linear MDPs with known features and deployment that permits episode-start randomization; otherwise a nonconvex operational metric is preferable. HV(conv) computation/certification as $m$ grows and explicit-model recovery without anchors remain challenging. The explicit base-kernel result is scoped to the anchored identifiable branch; anchor-free recovery should target the full transition map, not a noncanonical basis.

## Impact Statement

This paper presents work whose goal is to advance the field of machine learning. There are many potential societal consequences of our work, none of which we feel must be specifically highlighted here.

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

# Appendix Roadmap

This supplement is written to make the paper easy to audit. It (i) records reproducible experimental protocols and additional diagnostics, (ii) collects imported technical tools in a checkable "assumptions → conclusion" form, and (iii) provides full proofs for the new results. For reviewer convenience, we summarize where each ingredient lives, what it contains, and which part of the main paper it supports.

| Appendix | Contents | How it supports the main paper |
|---|---|---|
| A | **Supplementary experiments and protocol.** Key items: (i) full reproducible protocol and metrics; (ii) E1 scaling-collapse diagnostic (Appendix A.3); (iii) E10 $w$-net runtime/complexity and HV(conv) vs. HV(raw) diagnostics (Appendix A.4); (iv) preference-space / hyperparameter diagnostic maps (Figure 9); (v) full PFE separation dashboard (Appendix A.7); (vi) compact supplementary diagnostics (Figure 11); (vii) richer baseline visualizations, ablations, scale-up, and environment-instance sweeps. | Complements Section 7 (E1–E3) and the "stability chain" discussion: adds the plots and reproducibility notes that would otherwise be space-prohibitive. |
| B | **External tools and theorem blocks.** Centralized statements for: concentration/martingale tools, predictable-sequence online learning, linear-bandit/linear-MDP optimism machinery, reward-free exploration primitives, and convex-analysis / hypervolume facts. Each block is stated as explicit assumptions → conclusion. | Used across the paper (online regret, HV stability, and PFE proofs); Appendix C cites these blocks by label to keep proofs modular and auditable. |
| C | **Proofs for new claims.** Full proofs for the paper's new results, including: filtration-safe online regret under predictable preferences (M1), the HV(conv) semantics and support-gap → HV stability chain (M2), and the decision-optimal PFE guarantee plus the explicit-model separation (M3). | Provides the complete derivations that are only sketched in the main paper; each proof points to the exact external tool used for every nontrivial step. |
| D | **Additional discussion, counterexamples, and notation.** Sanity checks and pitfalls (e.g., regret $\not\Rightarrow$ HV coverage; HV(raw) vs. HV(conv) ranking flips; measurability issues), plus auxiliary notation that would interrupt the main narrative. | Supports the "what the guarantees do/do not imply" clarifications in Sections 4–8. |

Each appendix section begins with a short *reading note* that states its purpose and points to the main statements it supports.

## A. Supplementary Experiments and Reproducible Protocol

**Purpose and reading note.** Appendix A complements Section 7 by documenting the full experimental protocol and by reporting additional diagnostics/ablations that support the empirical claims. Unless stated otherwise, experiments use the synthetic anchored simplex-mixture simulator (Remark 3.7) and evaluate (i) *predictable-preference online regret* and (ii) *deployable hypervolume* computed on the convex hull of collected return vectors (HV(conv)). In particular, Appendix A.3 provides the scaling-collapse plot used in (E1), and Appendix A.4 quantifies why $w$-*net discretization* is computationally prohibitive at large objective dimension.

### A.1. Experimental protocol (reproducible)

We describe the protocol and evaluation used in all experiments.

**Key evaluation metrics (for non-specialists).** We report two complementary signals. First, for predictable-preference online learning we evaluate the scalarized regret

$$\text{Reg}(K) := \sum_{k=1}^{K} \big( V^*(w_k) - V^{\pi_k}(w_k) \big), \qquad V^\pi(w) := \langle w, v(\pi) \rangle,$$

where $V^*(w) := \max_\pi V^\pi(w)$ is computed by dynamic programming in the known simulator (so the plotted curves isolate *learning* error from Monte Carlo evaluation noise). Second, to summarize the quality of the *learned policy set* we report the *deployable hypervolume* of the convex hull of collected return vectors, $\text{HV}(\text{conv}(\Pi_K))$. Convexification reflects that a finite set of policies can be deployed via randomization, and it avoids ranking artifacts that can occur when reporting $\text{HV}(\cdot)$ on a non-convex set (Proposition 4.4 and Figure 7, panels a–b).

**Synthetic anchored simplex-mixture generator (linear-MDP special case).**

- Choose $|\mathcal{S}|$, $|\mathcal{A}|$, $H$, $d$, $m$.

- Fix feature maps $\phi_h(s, a) \in \Delta_d$ (e.g., random sparse simplex vectors).

- Sample base kernels $P_h^j$ and reward parameters $\theta_{h,i}$ to match the simplex-mixture structure in Remark 3.7 and the reward normalization in the main text.

**Online evaluation under predictable preferences.**

- Generate $w_k$ by (a) i.i.d. Dirichlet(1), (b) cyclic adversarial schedule, and (c) a predictable adversary that depends on past realized returns but not current-episode randomness (Definition 3.5).

- Run Algorithm 1 with a chosen base planner implementation.

- Report regret and the hypervolume of the convex hull of collected return vectors.

A.1.1. CYCLIC SCHEDULE FOR PREDICTABLE PREFERENCES

In the *cyclic* predictable-adversary experiments, we fix in advance a finite list of weight vectors $\{w^{(1)}, \ldots, w^{(M)}\} \subset \Delta_m$ and then set $w_k := w^{\big((k-1) \bmod M\big)+1}$ for episode $k$. This sequence is deterministic (hence predictable) and repeats with period $M$.

**PFE evaluation.**

- Run exploration for $N$ episodes.

- Sample test preferences $w$ from a dense set or random Dirichlet.

- For each $w$, compute suboptimality $V^*(w) - V^{\hat{\pi}(w)}(w)$ using either exact planning in the known simulator (for evaluation) or Monte Carlo rollouts.

- Report worst-case over test weights, plus hypervolume of the convex hull of policies induced by a weight grid (Theorem 6.7).

**Statistics.**

- Use at least 20 random seeds.

- Report mean and 95% confidence intervals for regret and suboptimality.

- Hypervolume: report median and interquartile range (HV often has heavy tails).

**A.2. Guide to the supplement**

The remainder of Appendix A provides additional experiments, diagnostics, and ablations that support the main results. Appendix A.4 quantifies the computational overhead of $w$-net discretization (E10). Appendices A.5 and A.6 provide diagnostic maps over preference space and hyperparameters. Appendix A.7 expands the PFE separation results with the full dashboard and compact diagnostics. Appendices A.9–A.13 provide richer baseline visualizations, component ablations, scale-up results, and robustness across independently generated MDP instances.

### A.3. Scaling-collapse diagnostic for normalized regret (E1)

**What this plot is testing.** For near-minimax optimistic planning in linear MDPs, the leading-order online regret is predicted to scale as $\mathrm{Reg}(K) = \tilde{\mathcal{O}}\big(dH^{3/2}\sqrt{K}\big)$ (up to logarithmic factors). To check whether our empirical curves are consistent with this dependence on dimension and horizon, we plot the normalized quantity $\mathrm{Reg}(K)/(dH^{3/2}\sqrt{K})$ across multiple $(d, H)$ settings. If the scaling law captures the dominant dependence, curves from different $(d, H)$ should approximately *collapse* after normalization.

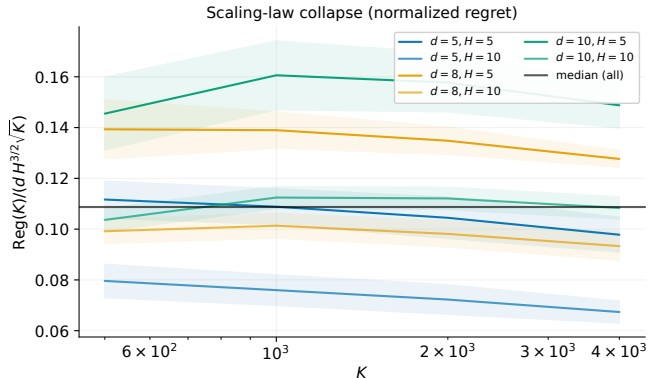

*Figure 6.* Scaling-collapse diagnostic for normalized regret (E1). Each curve corresponds to a $(d, H)$ pair and reports $\mathrm{Reg}(K)/(dH^{3/2}\sqrt{K})$ versus episodes $K$; shaded regions denote 95% confidence intervals across seeds and the thick black curve is the median across all settings.

**Takeaway.** After normalization, curves for different $(d, H)$ cluster around a common scale, which is consistent with the predicted leading-order $dH^{3/2}\sqrt{K}$ dependence; the remaining spread is expected from logarithmic terms and finite-$K$ effects.

### A.4. $w$-net dimension curse (grid size and runtime; E10)

**Context.** A common heuristic for handling arbitrary preferences is to discretize the simplex $\Delta_m$ into a finite $w$-net (e.g., a simplex lattice of resolution $q$) and to answer a query $w$ by its nearest grid point $\bar{w}$. This can work in low dimension, but it introduces a combinatorial dependence on the objective dimension: the grid size is $L = \binom{q+m-1}{m-1}$. In an online setting, $w$-net methods also pay a per-episode overhead for nearest-grid-point search (and, if one naively maintains separate learners per grid point, an even larger overhead).

**How to read Figure 7.** Panels (a,b) report method ranks computed from the same learned return sets but under two hypervolume definitions. *HV(raw)* applies hypervolume directly to the empirical set of return vectors, while *HV(conv)* first takes the convex hull (deployable under policy randomization) and then computes hypervolume. The rank stripes highlight that HV(raw) can change method ordering relative to HV(conv), motivating our use of HV(conv) for all reported hypervolume numbers and certificates. Panel (c) visualizes the simplex-lattice size explosion $L$ as a function of $(m, q)$; values above roughly $10^6$ are already infeasible in our implementation.

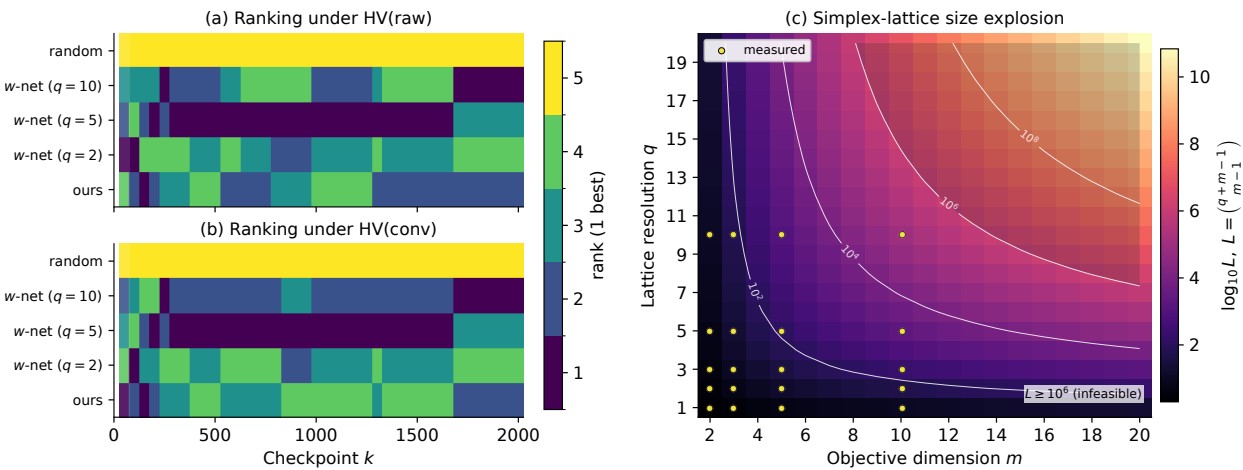

**Figure 7.** (Left) HV(raw) vs HV(conv) ranking stripes on learned policy sets (E2). (Right) $w$-net curse (E10): simplex-lattice size $L = \binom{q+m-1}{m-1}$ grows combinatorially in $(m, q)$.

Figure 7 (right) visualizes the simplex-lattice grid-size explosion $L = \binom{q+m-1}{m-1}$ as a function of $(m, q)$. To quantify the wall-clock impact, Figures 8a and 8b and Table 1 report fixed time-budget comparisons (for $m = 10$ and $m = 200$) and measured overhead statistics in our implementation.

**Connection to our method.** Our protocol-safe wrapper avoids enumerating a $w$-net: it learns objective-wise reward (and transition) models once, and then synthesizes the scalarized reward/values on demand via *query-time* combinations (Section 5, "coordinate-wise regression and query-time synthesis"). Consequently, the additional bookkeeping cost grows roughly linearly in $m$ rather than combinatorially in $(m, q)$, which is exactly what the fixed time-budget comparisons in Figure 8 illustrate.

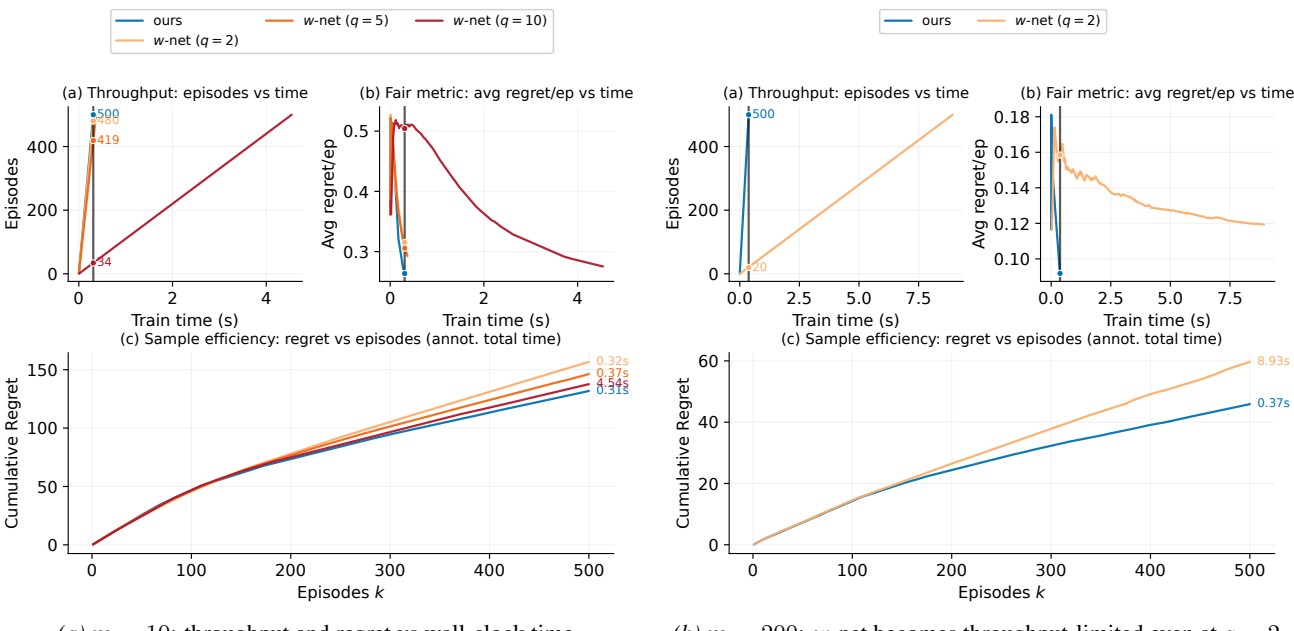

*(a) $m = 10$: throughput and regret vs wall-clock time.*          *(b) $m = 200$: $w$-net becomes throughput-limited even at $q = 2$.*

**Figure 8.** Fixed wall-clock time-budget comparisons (E10). Side-by-side, these plots highlight how $w$-net overhead quickly dominates the training budget as objective dimension grows.

*Table 1.* $w$-net deployability numbers (E10; $K = 500$): lattice size $L = \binom{q+m-1}{m-1}$, cached lattice memory, and measured per-episode train time / slowdown vs no-$w$-net.

| $m$ | $q$ | grid size $L$ | grid mem (MB) | train/ep (ms) | slowdown |
|---|---|---|---|---|---|
| 2 | 2 | 3 | $< 0.01$ | 0.62 | 1.02× |
| 2 | 5 | 6 | $< 0.01$ | 0.62 | 1.02× |
| 2 | 10 | 11 | $< 0.01$ | 0.61 | 1.02× |
| 10 | 2 | 55 | $< 0.01$ | 0.63 | 1.03× |
| 10 | 5 | 2 002 | 0.15 | 0.71 | 1.17× |
| 10 | 10 | 92 378 | 7.05 | 9.35 | 15.41× |
| 20 | 2 | 210 | 0.03 | 0.64 | 1.05× |
| 20 | 5 | 42 504 | 6.49 | 3.64 | 5.92× |
| 20 | 10 | 20 030 010 | 3 056 | – | *infeasible* |
| 50 | 2 | 1 275 | 0.49 | 0.77 | 1.20× |
| 50 | 5 | 3 162 510 | 1 206 | – | *infeasible* |
| 50 | 10 | 62 828 356 305 | $2.4 \times 10^7$ | – | *infeasible* |
| 100 | 2 | 5 050 | 3.85 | 1.75 | 2.62× |
| 100 | 5 | 91 962 520 | 70 162 | – | *infeasible* |
| 100 | 10 | 42 634 215 112 710 | $3.3 \times 10^{10}$ | – | *infeasible* |

## A.5. Preference-space diagnostics ($m = 3$)

Figure 9a visualizes a *preference-space diagnostic* that is hard to see from a single scalar regret curve: for each weight $w \in \Delta_3$, we plot the instantaneous support gap $V^*(w) - \max_{\pi \in \Pi_k} V^\pi(w)$ at a representative checkpoint. Darker regions indicate preferences for which the current policy set remains far from optimal, helping localize where learning is difficult and how this difficulty evolves over training.

**How to read the ternary plot.** Each point in the triangle corresponds to a weight vector $w \in \Delta_3$ (here evaluated on a simplex-lattice grid). The three corners put all weight on a single objective. The colormap reports the support gap: larger values mean larger suboptimality for that preference.

**Definition and reproducibility.** For each $w \in \Delta_3$, the plotted value is the (nonnegative) support-function gap $\text{gap}(w) = \left[V^*(w) - \max_{\pi \in \Pi_K} V^\pi(w)\right]_+$. We compute $V^*(w)$ by exact DP in the known simulator and evaluate $\max_{\pi \in \Pi_K} V^\pi(w)$ over the prefix policy set at $K{=}2000$ episodes under i.i.d. Dirichlet(1) preferences. To make the ternary heatmap stable, we fix the underlying MDP instance (env seed 0) and average the gap pointwise over 5 algorithm seeds. Weights are evaluated on a simplex-lattice grid with resolution $q{=}35$ (so $|W| = \binom{q+m-1}{m-1} = 666$ for $m{=}3$).

## A.6. Hyperparameter robustness

To ensure the empirical conclusions are not an artifact of a narrow hyperparameter choice, we sweep two influential knobs: the reward bonus scale (which controls optimism in the reward model) and a Dirichlet smoothing parameter used in transition estimation. Figure 9b reports the final cumulative regret across this grid; the broad regions of comparable performance indicate that the method is not overly brittle to these tunings.

**How to read the heatmap.** The horizontal axis controls the reward optimism bonus level and the vertical axis controls the transition pseudo-count smoothing. Lower values correspond to lower final regret; the broad low-regret region indicates robustness rather than fragile tuning.

**Sweep details.** We run a 2D grid over reward bonus scale $\beta_r \in [0.07, 1.2]$ (51 points) and Dirichlet smoothing $\alpha_0 \in \{10^{-4}, 3 \cdot 10^{-4}, 10^{-3}, 3 \cdot 10^{-3}, 10^{-2}\}$, with a reduced budget $K{=}500$ for tractability (3 seeds; i.i.d. preferences). The plotted heatmap reports final cumulative regret (mean over seeds); grey cells correspond to failed runs. The default configuration used elsewhere is $(\beta_r, \alpha_0) = (0.3, 10^{-3})$ (marked on the plot).

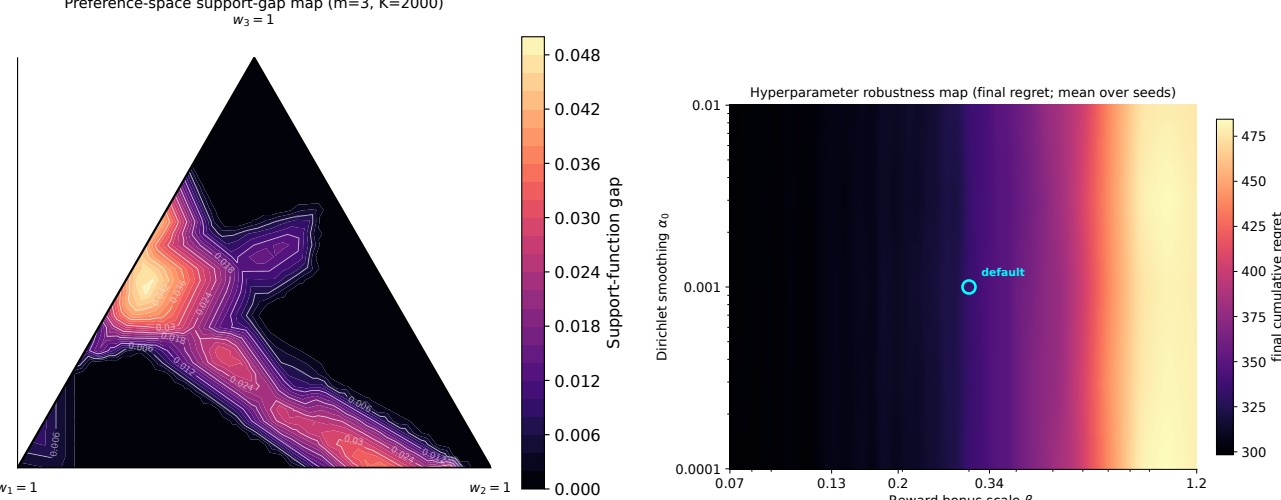

*(a)* Preference-space support-gap map on $\Delta_3$ ($K{=}2000$; mean over 5 algo seeds; env seed fixed).

*(b)* Hyperparameter robustness sweep (final cumulative regret; $K{=}500$).

*Figure 9.* Diagnostic maps that complement scalar regret curves. Left: where the current policy set is far from optimal as a function of preference. Right: robustness over $(\beta_r, \alpha_0)$ sweeps.

### A.7. PFE separation dashboard

The main text summarizes the two headline trends for PFE scaling (Section 7, E3): decision-optimal PFE follows the $N^{-1/2}$ scaling, while explicit-model transition estimation exhibits an unavoidable linear $|\mathcal{S}|$ dependence. Here we provide the full 4-panel dashboard (including gap distributions and the scaling-collapse normalization) and additional diagnostics.

**How to interpret the four panels.** This dashboard intentionally juxtaposes two *different* PFE objectives. Panels (a,b) summarize decision-optimal PFE in the anchored simplex-mixture MDP ($S{=}20, H{=}5, d{=}5, m{=}2$; 20 seeds; $N \in \{50, 100, 200, 500, 1000\}$; evaluated on a dense weight grid), while panels (c,d) probe explicit-model estimation via an Anchor Bandit ($d{=}10$) across $S \in \{20, 50, 100, 200\}$ (20 seeds). Panel (b) uses a representative seed only to show the distributional shape (ECDF) over test weights. Thus, the left and right columns should not be read as the same run—they are a side-by-side comparison of objectives.

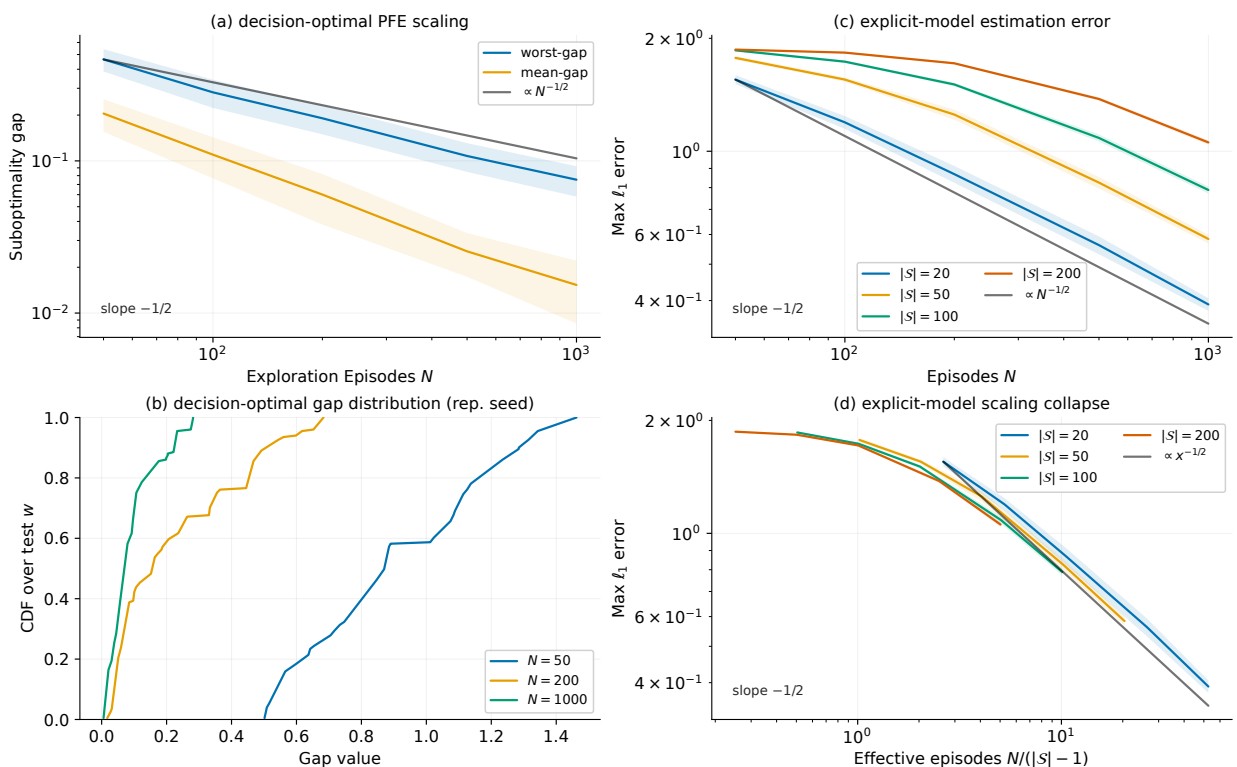

*Figure 10.* Full PFE separation dashboard. Panels (a,c) highlight the headline trends discussed in the main text (E3); panels (b,d) report a representative-seed gap distribution and the explicit-model scaling collapse after normalization by $N/(|\mathcal{S}|-1)$.

We additionally sweep horizon $H$ for explicit-model transition estimation; Figure 11a reports the distribution over seeds of base-kernel $\ell_1$ estimation errors at a fixed episode budget as a function of $H$.

### A.8. Supplementary diagnostics (compact layout)

Figure 11 reports two supplementary diagnostics in a compact layout. Panel (a) studies horizon sensitivity for explicit-model transition estimation in PFE, and panel (b) calibrates the support-gap $\to$ HV stability chain across methods, seeds, and checkpoints.

**Reproducibility notes.** For panel (a), we sweep horizon $H \in \{3, 5, 10, 20\}$ at a fixed exploration budget $N{=}1000$ episodes and report the median/IQR over 20 seeds of base-kernel $\ell_1$ estimation error (mean and max over $(h, j)$). For panel (b), each dot corresponds to a triple (method, seed, checkpoint $k$) with checkpoints 50:50:2000 (5 seeds; 4 methods), x-axis $\varepsilon_{\sup}(k)$ computed on a dense weight grid and y-axis the observed gap-to-final in deployable $\mathrm{HV}(\mathrm{conv}(\Pi_k))$. The plotted line uses the proxy constant $c{=}2H$ (here $H{=}5$ so $c{=}10$). This is an *empirical calibration* sanity check rather than a theorem claim.

**What panel (a) is showing (and why it can look "flat").** Panel (a) is not a learning-curve over time; it is a *horizon ablation* at a fixed sample budget. Because each episode provides one transition sample at *each* stage, holding $N$ fixed means every stage still sees $N$ samples. Consequently, the *mean* base-kernel error is nearly invariant to $H$ in this synthetic family. The *maximum* error (blue curve) increases mildly with $H$ because we maximize over more $(h, j)$ pairs, so the "worst" stage is more likely to be an outlier (a multiple-comparisons/union-bound effect). The intended takeaway is therefore qualitative: at this budget, horizon alone does not create a large explicit-model estimation gap in the mean, but worst-case errors become harder to uniformly control as $H$ grows. This aligns with the main-text separation message in (E3): the dominant explicit-model difficulty is driven by *model size* (e.g., $|\mathcal{S}|$ and the number of kernels to estimate), rather than by decision-optimality requirements.

**What panel (b) is checking and how it connects to the main text.** Panel (b) is a compact empirical "sanity check" for the stability chain discussed in Section 4: support-function (scalarization) accuracy controls geometric set error, which in turn controls the HV(conv) error. Here $\varepsilon_{\sup}(k)$ is the worst-case support-gap of the current policy set (a certificate-style diagnostic that is cheap to compute given a preference grid), and the y-axis measures the *observed* hypervolume deficit to the final checkpoint. The log–log trend and the fitted slope indicate that, across methods/seeds/checkpoints, reducing the support-gap is strongly correlated with closing the HV(conv) gap. The solid proxy line $c\,\varepsilon_{\sup}(k)$ illustrates that a simple constant choice (here $c=2H$) is conservative for this instance: most points lie below it.

**Reading guide for panel (b).** The x-axis measures the worst-case scalarization error (support-function gap) of the current policy set, while the y-axis measures how far the current deployable hypervolume still is from its final value. A roughly linear trend on the log–log plot is consistent with the certificate-style implication "small support error ⇒ small HV deficit". Points below the proxy line indicate that the chosen constant is conservative.

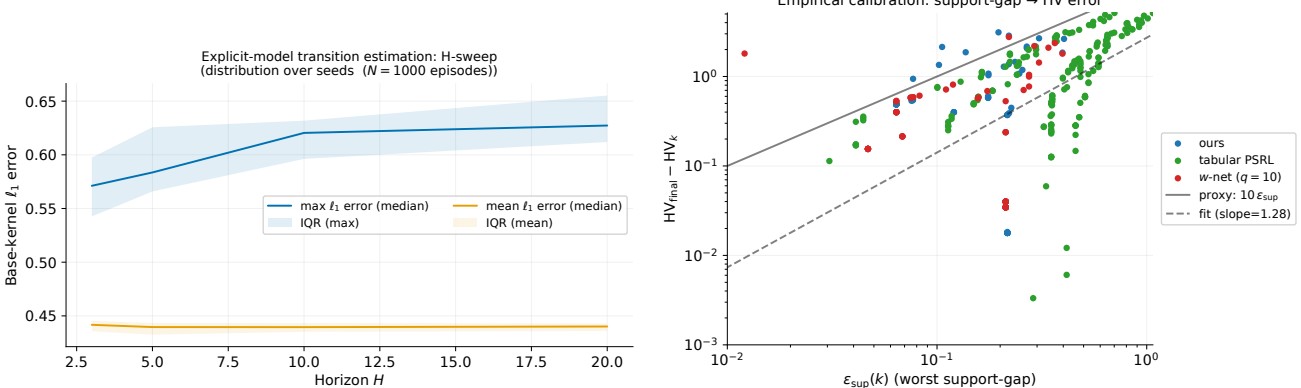

*(a)* Explicit-model transition estimation: horizon $H$ sweep (seeds=20).

*(b)* Empirical calibration of support-gap $\varepsilon_{\sup}(k)$ vs HV gap-to-final (log–log).

*Figure 11.* Compact supplementary diagnostics. (a) Explicit-model transition estimation under a fixed budget: the *mean* base-kernel $\ell_1$ error is nearly invariant to horizon $H$, while the *max* error increases mildly with $H$ (a multiple-comparisons effect over more stages/kernels). (b) Empirical calibration of the support-gap $\varepsilon_{\sup}(k)$ vs. the observed HV(conv) gap-to-final across methods, seeds, and checkpoints (log–log): smaller support-gap is strongly correlated with smaller hypervolume deficit, consistent with the support-gap $\rightarrow$ HV stability chain discussed in Section 4.

### A.9. Richer baseline comparisons and Pareto-set visualization

We provide two qualitative views that complement scalar regret/HV summaries.

**Pareto-set visualization.** Figure 12 visualizes the return vectors collected along training for a representative seed, together with deployable convex-hull snapshots (the boundary used by HV(conv)). This helps interpret whether improvements come from discovering new extremes or from filling in the hull.

**Plotting details.** Each panel shows the density of return vectors along the training trajectory (hexbin background) and overlays a few convex-hull snapshots computed from the prefix set $\{v_{\pi_t}\}_{t\leq k}$ at $k \in \{100, 500, 1000\}$ (markers indicated in the guide panel). This visualization is run on a representative seed for readability; statistical comparisons remain in the main regret/HV curves.

**Qualitative pattern.** Across methods, higher deployable HV(conv) corresponds to a larger and more stable hull boundary. In the representative-seed view, our method and $w$-net rapidly expand coverage toward extreme trade-offs and then densify near the boundary, whereas tabular baselines tend to concentrate in a narrower region of return space. (The panel titles report the seed's HV(conv) for quick reference; quantitative comparisons are in the multi-seed curves reported in the main text and earlier appendix sections.)

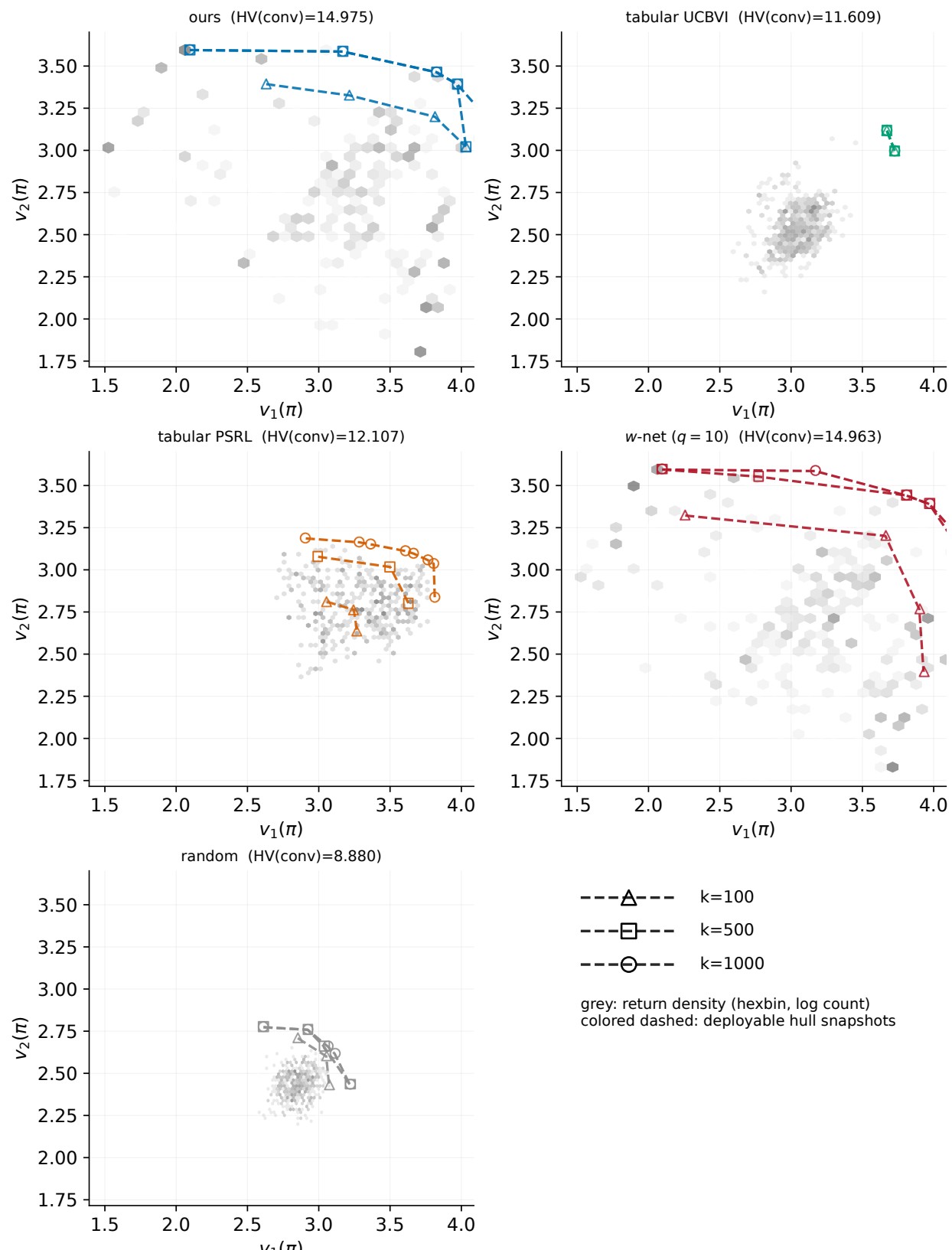

*Figure 12.* Pareto visualization with reduced clutter (representative seed=0): density-based small multiples (one panel per method) with deployable hull snapshots, improving readability when point clouds are dense.

**HV stability chain dashboard.** To connect directly to our support-function → Hausdorff → HV stability chain, Figure 13 juxtaposes a support-gap heatmap, the deployable hypervolume $\text{HV}(\text{conv}(\Pi_k))$, the worst-case support-gap $\varepsilon_{\text{sup}}(k)$, and a simple proxy upper bound on the HV error.

**Implementation details.** Because this dashboard requires exact $\text{HV}(\text{conv}(\Pi_k))$ and exact $V^*(w)$ on a dense preference grid, it is generated for $m{=}2$ (where 2D convex-hull HV is exact in our code). Panel (a) uses $w = (w_1, 1 - w_1)$ with a 1D grid, and panel (c) takes the worst-case over $w$ to obtain $\varepsilon_{\text{sup}}(k)$.

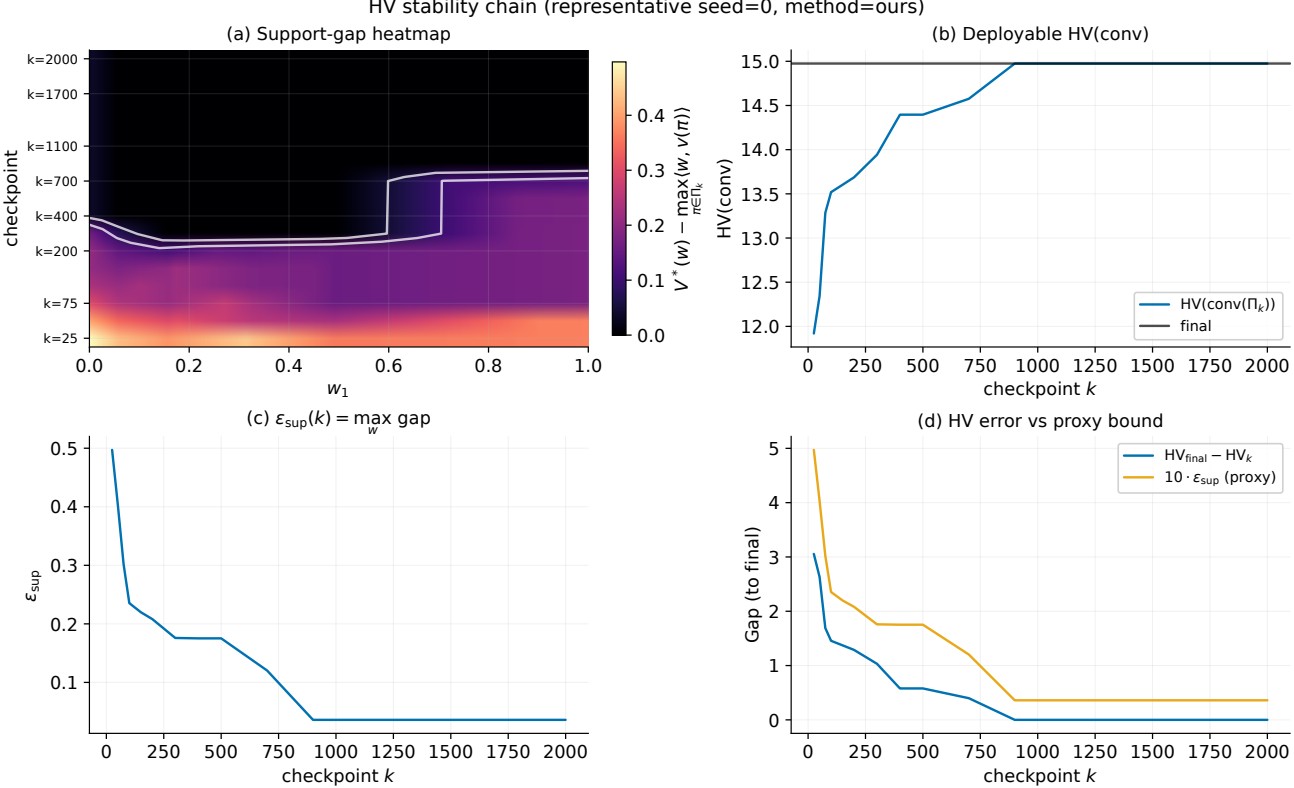

*Figure 13.* HV stability chain dashboard (representative seed=0): (a) support-gap heatmap over checkpoints and preferences; (b) deployable HV(conv) along training; (c) worst-case support-gap $\varepsilon_{\text{sup}}(k)$; (d) HV error-to-final compared with a proxy bound $c \cdot \varepsilon_{\text{sup}}(k)$ (with $c$ set using the return upper bound $U = H$).

**Empirical calibration.** As an empirical calibration test of the support-gap → HV error chain, Figure 11b collects points across methods, seeds, and checkpoints and compares the observed HV gap-to-final against the support-gap proxy (log–log).

### A.10. Upper-bound component ablations

To probe the empirical upper bound of our approach, we separate reward learning and transition learning. We compare (i) our full method (learn reward + learn transition), (ii) an oracle-transition variant (KnownP; learn reward only), and (iii) an oracle-reward variant (learn transition only). Figure 14 (right panel) summarizes the cumulative regret under i.i.d. preferences; Figure 14 (left panel) reports the scale-up sanity check discussed in Section A.12.

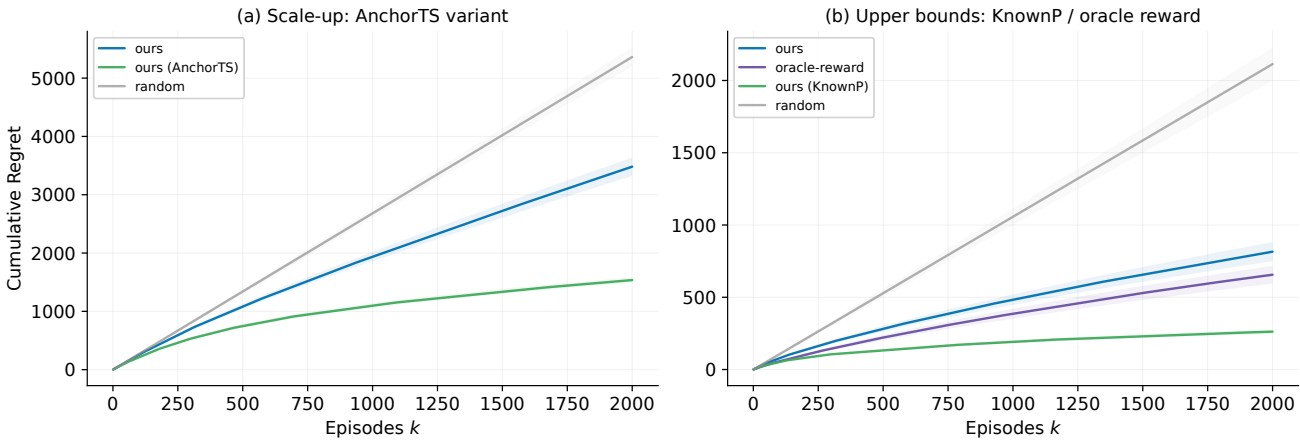

*Figure 14.* Left: scale-up sanity check on a larger instance ($|\mathcal{S}| = 50$, $H = 10$, $d = 10$), comparing AnchorUCB vs posterior-sampling AnchorTS (seeds=20). Right: upper-bound component ablations under i.i.d. preferences (seeds=20), separating reward-learning and transition-learning effects via oracle components (KnownP / oracle reward).

To directly connect the regret gap to transition learning, Figure 15 reports a representative-seed diagnostic: we track the $\ell_1$ error of the estimated base kernels and relate it to the regret gap between AnchorUCB (unknown $P$) and KnownP (oracle transition).

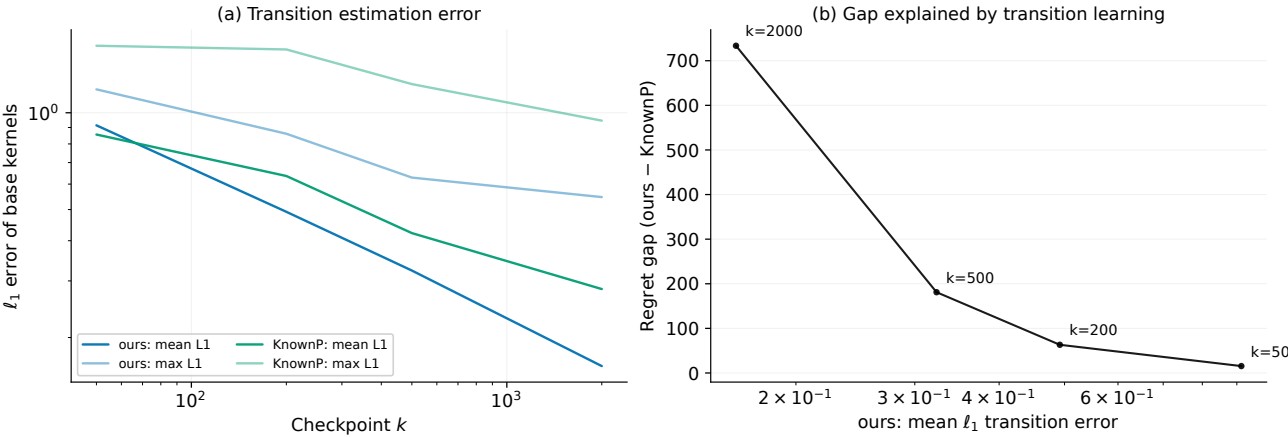

*Figure 15.* Transition-learning diagnostic (representative seed=0): (a) base-kernel $\ell_1$ estimation error along training for AnchorUCB vs KnownP; (b) regret gap vs transition estimation error, illustrating that the remaining gap is largely dominated by transition exploration.

## A.11. Posterior-sampling base planning (AnchorTS)

The upper-bound component ablation suggests that the remaining gap to the oracle-transition variant (KnownP) is largely dominated by how the base planner explores unknown transitions. As an empirical upper-limit, we also consider a posterior-sampling base planner (AnchorTS): at the beginning of each episode, we sample each base kernel from its Dirichlet posterior and then plan by DP.

We additionally report a scale-up instance in Section A.12 (Figure 14), which shows a substantial regret reduction compared to AnchorUCB and supports the "base-planner modularity" discussion in the main text.

## A.12. Scale-up sanity check

To validate robustness beyond our main synthetic setting, we additionally run the same online protocol on a larger anchored simplex-mixture instance ($|\mathcal{S}| = 50$, $H = 10$, $d = 10$, $|\mathcal{A}| = 15$) with unknown transitions. We report a posterior-sampling base-planning variant (AnchorTS) on this instance; Figure 14 shows a substantial regret reduction compared to AnchorUCB,

supporting the "base-planner modularity" discussion in the main text.

## A.13. Environment-instance sweep (multiple MDP instances)

We evaluate performance across independently generated MDP instances from the same synthetic family. Each "env seed" corresponds to a different sampled MDP (different base kernels/reward parameters), and we repeat training on each instance. Figure 5 (main text) summarizes the distribution of final cumulative regret and (approx.) deployable HV(conv) under i.i.d. preferences ($m{=}10$). This appendix records the hypervolume estimator used for $m > 2$.

**Note on hypervolume for $m > 2$.** For $m{=}10$, exact computation of deployable HV(conv$(\cdot)$) is expensive in high dimensions (Beume et al., 2009; Guerreiro et al., 2021). We therefore use a Monte Carlo estimator and report HV(conv) only as a qualitative robustness signal (regret remains the primary quantitative metric in high dimensions). Concretely, let $V$ be the set of return vectors collected during training and $C := \mathrm{conv}(V)$. We estimate the dominated-region volume $\mathrm{HV}(C) = \mathrm{Vol}\big(D_0(C) \cap [0, H]^m\big)$ by sampling $N_{\mathrm{mc}}{=}2000$ points $z \sim \mathrm{Unif}([0, H]^m)$ and estimating the fraction classified as dominated. To test whether $z \in D_0(C)$ efficiently, we use the convex-analytic characterization

$$z \in D_0(C) \iff \langle w, z \rangle \le h_C(w) := \max_{v \in V} \langle w, v \rangle \quad \text{for all } w \in \Delta_m,$$

which holds because $D_0(C) = C - \mathbb{R}_+^m$ is convex and satisfies $h_{D_0(C)}(w) = h_C(w)$ for all $w \in \mathbb{R}_+^m$. We approximate the universal quantifier by a finite direction set of size 512 with $w \sim \mathrm{Dirichlet}(1)$. This direction-based test can be optimistic (it may over-classify some points as dominated), so the resulting HV(conv) values should be interpreted cautiously.

# B. External Tools and Theorem Blocks

**Purpose and how to navigate.** We collect standard lemmas and external theorem statements used throughout the paper. Each result is listed in a checkable "assumptions → conclusion" format, together with brief notes on how it matches our setting. In Appendix C we reference these blocks by label, allowing the remaining proofs to focus on the paper-specific arguments.

## B.1. Quick reference tables (comparison, notation, and main results)

To make the proof appendices easier to navigate, we include three compact tables at the start of the proof-related material: a positioning/comparison table, a key-notation table, and a one-page overview of the main results with pointers.

*Table 2.* Positioning relative to representative paradigms. "$w$-net" refers to preference discretization / convex coverage set approaches that solve (approximately) many scalarizations on a finite grid of weights. "Reward-free" refers to scalar reward-free exploration followed by planning for a revealed reward. A dagger † indicates the feature holds only under restrictions (e.g., fixed weights announced in advance, scalar rewards, or evaluation without an explicit uniform certificate).

|  | Single-objective RL | $w$-net MORL | Reward-free (scalar) | This work |
|---|---|---|---|---|
| Handles predictable/adversarial episode-wise preferences | × | × | × | ✓ |
| No discretization of $\Delta_m$ in online learning | ✓ | × | ✓ | ✓ |
| Filtration-safe w.r.t. adaptive weights (no post-hoc scalarization) | ✓ | † | ✓ | ✓ |
| Deployable evaluation target HV(conv$(\cdot)$) + calibrated certificate | × | † | × | ✓ |
| Reward-free preference-free query answering | × | × | † | ✓ |
| Decision-optimal vs. explicit-model separation for PFE | × | × | × | ✓ |

*Table 3.* Key notation (abridged; the paper defines additional symbols as needed).

| Symbol | Meaning |
|---|---|
| $H, K$ | horizon; number of episodes (online) or exploration episodes (PFE) |
| $m, d$ | number of objectives; feature dimension in linear MDPs |
| $\mathcal{S}, \mathcal{A}$ | state and action spaces |
| $w \in \Delta_m$ | nonnegative preference/weight vector (simplex) |
| $\phi_h(s, a)$ | known stage-dependent state–action feature vector |
| $\theta_{h,i}$ | unknown linear reward parameter for objective $i$ at stage $h$ |
| $\Lambda_{k,h}, \ b_{k,h,i}$ | ridge (design) matrix and response vector at stage $h$ after $k-1$ episodes |
| $\hat{\theta}_{k,h,i}$ | ridge estimate $\Lambda_{k,h}^{-1} b_{k,h,i}$; coordinate-wise reward estimator |
| $\beta_{k,h}^r$ | optimistic reward confidence radius (after scalarization by $w_k$) |
| $\mathcal{C}^*$ | deployable return set $\mathrm{conv}(\mathcal{V}_{\mathrm{det}})$ under episode-start randomization |
| $h_C(w)$ | support function $\sup_{x \in C} \langle w, x \rangle$ |
| $\mathrm{HV}(\cdot)$ | hypervolume w.r.t. reference point 0 |

*Table 4.* Main results overview (paper statements; proofs are in Appendix C).

| Task | Guarantee (informal) | Where |
|---|---|---|
| M1: predictable preferences (filtration-safety) | Predictable scalarization preserves MDS/subgaussianity; enables time-uniform self-normalized bounds | Thm. 6.1 |
| M1: online regret in linear MDPs | No discretization of $\Delta_m$; regret inherits scalar linear-MDP planner rate (plus lower-order reward-learning term) | Thm. 6.2 |
| M1: lower bound | Reduction to single-objective linear MDP yields $\Omega(d\sqrt{H^3 K})$ | Thm. 6.4 |
| M2: deployable HV(conv) stability | Uniform support-function error $\Rightarrow$ HV error via Hausdorff-$\infty$ stability | Thms. 4.9, 4.10 |
| M2: $w$-net curse | Optimal covering rate on $\Delta_m$ is $\Theta(L^{-1/(m-1)})$; exponential weights in $m$ | Thm. 6.8 |
| M3: decision-optimal PFE | Reward-free exploration sample complexity matches minimax (up to polylog) for query-time planning | Thm. 6.10 |
| M3: explicit-model separation | Explicit base-kernel recovery can require polynomially more samples; upper/lower bounds | Thms. 6.13, 6.14 |

**Bibliographic notes.** Predictable sequences in online learning are formalized by Rakhlin & Sridharan (2013). Our martingale and self-normalized concentration tools build on classical results on martingale tails and self-normalized processes (Freedman, 1975; de la Peña et al., 2009; Howard et al., 2020) and standard linear-bandit analyses (Abbasi-Yadkori et al., 2011). Reward-free (task-agnostic) exploration originates in Jin et al. (2020a) and subsequent refinements include adaptive and linear-function-approximation settings (Kaufmann et al., 2021; Wang et al., 2020). Hypervolume foundations and computation are discussed in (Zitzler et al., 2007; Beume et al., 2009; Guerreiro et al., 2021), with recent regret perspectives in Zhang (2024). For MORL surveys and close theoretical variants under lexicographic structure, see (Roijers et al., 2013; Xue et al., 2025); scalarization background is covered in Miettinen (1999). Convex-analytic background for support functions and convex bodies can be found in (Schneider, 2013).

## B.2. Conditional Hoeffding lemma (bounded implies subgaussian)

**Theorem B.1** (conditional Hoeffding lemma). *Let $X$ be a random variable and let $\mathcal{H}$ be a sigma-field. Assume:*

- $\mathbb{E}[X \mid \mathcal{H}] = 0$ *almost surely,*

- $X \in [a, b]$ *almost surely.*

*Then for any real $\alpha$,*

$$\mathbb{E}[\exp(\alpha X) \mid \mathcal{H}] \leq \exp\left(\frac{\alpha^2 (b-a)^2}{8}\right).$$

**Reference:** Hoeffding's lemma; see, e.g., Hoeffding (1963).

## B.3. Time-uniform self-normalized inequality (Abbasi-Yadkori style)

**Theorem B.2** (time-uniform self-normalized inequality; Abbasi-Yadkori et al. 2011)**.** *Let $(\mathcal{G}_t)$ be a filtration. For each $t \geq 1$:*

- $x_t \in \mathbb{R}^d$ *is $\mathcal{G}_{t-1}$-measurable with $\|x_t\|_2 \leq 1$,*

- $\eta_t$ *is $\mathcal{G}_t$-measurable with $\mathbb{E}[\eta_t \mid \mathcal{G}_{t-1}] = 0$,*

- $\eta_t$ *is conditionally 1-subgaussian: $\mathbb{E}[\exp(\alpha\eta_t) \mid \mathcal{G}_{t-1}] \leq \exp(\alpha^2/2)$ for all $\alpha$.*

*Let $\Lambda_0 = \lambda I_d$ for $\lambda \geq 1$, and $\Lambda_t = \Lambda_0 + \sum_{s=1}^{t} x_s x_s^{\top}$.*

*Then for any $\delta \in (0, 1)$, with probability at least $1 - \delta$, for all integers $t \geq 1$ simultaneously:*

$$\Big\| \sum_{s=1}^{t} x_s \eta_s \Big\|_{\Lambda_t^{-1}} \leq \sqrt{2 \log\Big( \frac{\det(\Lambda_t)^{1/2} \cdot \det(\Lambda_0)^{-1/2}}{\delta} \Big)}.$$

**Reference:** Abbasi-Yadkori et al. (2011), Theorem 1 (time-uniform form).

## B.4. Elliptical potential lemma

**Theorem B.3** (elliptical potential)**.** *Let $\Lambda_0 = \lambda I_d$ with $\lambda \geq 1$, and $\Lambda_t = \Lambda_{t-1} + x_t x_t^{\top}$ with $\|x_t\|_2 \leq 1$. Then:*

$$\sum_{t=1}^{T} \min(1, \|x_t\|_{\Lambda_{t-1}^{-1}}^2) \leq 2 \log\Big( \frac{\det(\Lambda_T)}{\det(\Lambda_0)} \Big) \leq 2d \log\Big( 1 + \frac{T}{\lambda d} \Big).$$

**Reference:** Abbasi-Yadkori et al. (2011), Lemma 11.

## B.5. External near-minimax online regret theorem for linear MDPs

This is the single-objective theorem we treat as a black box in Theorem 6.2.

**Theorem B.4** (near-minimax online regret for linear MDPs; stated from He et al. 2023)**.** *Consider a scalar-reward episodic MDP with horizon $H$, finite state space $\mathcal{S}$, finite action space $\mathcal{A}$, and a* fixed *transition kernel. Assume the model is a (finite)* ***linear MDP*** *with known feature map in the following sense:*

- *There exists a known feature map $\phi_h : \mathcal{S} \times \mathcal{A} \to \mathbb{R}^d$ satisfying $\|\phi_h(s, a)\|_2 \leq 1$ for all $(h, s, a)$.*

- *(**Linear rewards**) There exist unknown vectors $\theta_h \in \mathbb{R}^d$ such that for all $(s, a, h)$, $\bar{r}_h(s, a) = \langle \phi_h(s, a), \theta_h \rangle$, and rewards are bounded in $[0, 1]$.*

- *(**Linear transitions**) There exist unknown vectors $\mu_h(s') \in \mathbb{R}^d$ for each $h$ and $s' \in \mathcal{S}$ such that for all $(s, a, h, s')$, $P_h(s' \mid s, a) = \langle \phi_h(s, a), \mu_h(s') \rangle$.*

- *(**Bounded parameters**) The parameters satisfy $\|\theta_h\|_2 \leq B_\theta$ and $\|\mu_h(s')\|_2 \leq B_\mu$ for known constants (the theorem's regret bound depends on these only through logarithmic factors/absolute constants as in He et al. 2023).*

- *The reward noise is a martingale difference sequence and is conditionally subgaussian with respect to the learner's filtration (a standard condition ensured in our setting by Theorem 6.1).*

*Then the algorithm* `LSVI-UCB++` *(He et al., 2023) with suitable tuning satisfies: for any $\delta \in (0, 1)$, with probability at least $1 - \delta$ (absorbing the constant factor 7 into $\delta$),*

$$\text{Regret}(K) \leq \widetilde{O}\big( d\sqrt{H^3 K} + d^7 H^8 \big),$$

*where $\widetilde{O}$ hides logarithmic factors in $(d, H, K, 1/\delta)$. For transparency, He et al. (2023) state a more detailed polynomial lower-order term; throughout we upper bound it by $d^7 H^8$ for a cleaner summary.*

**Reference:** He et al. (2023), Theorem 5.1 and Remark 5.2.

**Assumption match to our paper:**

- Our definition of linear MDPs in Section 3.3 matches the standard linear MDP model assumed by He et al. (2023). In particular, the transition kernels satisfy $P_h(s' \mid s, a) = \langle \phi_h(s, a), \mu_h(s') \rangle$ with known features $\|\phi_h(s, a)\|_2 \leq 1$ and bounded linear-transition parameters as required. (Remark 3.7 only concerns an additional anchored simplex special case used for explicit-model PFE, and is not needed for the online-regret results.)

- Our scalarized rewards are bounded in $[0, 1]$ by construction, and their noise is a martingale difference sequence that is conditionally 1-subgaussian under predictable preferences (Theorem 6.1).

- The modular interface theorem below (Section B.6) formalizes how our wrapper supplies an *episode-varying* optimistic reward oracle to a scalar base planner. It clarifies why low-switching planners that reuse value functions across episodes (such as `LSVI-UCB++` (He et al., 2023)) are not directly compatible with arbitrary preference changes, and provides a safe instantiation with standard episodic planners (e.g., `LSVI-UCB` (Jin et al., 2020b)).

### B.6. Modular reward interface for episode-varying scalar rewards

Here we formalize a simple but important point: our MORL wrapper can be combined with any *episodic optimistic* scalar planner via a reward-modular interface, even when the scalar reward mean changes from episode to episode due to predictable preferences. The key is that the optimism-based regret decomposition is episode-wise: conditioned on (i) a reward confidence event for the supplied scalar reward and (ii) whatever transition-confidence event is used by the planner, the per-episode regret is bounded by the planner's bonus terms, regardless of how the episode-wise reward function varies. This allows us to plug our reward-UCB oracle into standard planners that recompute their optimistic value functions each episode (e.g., `LSVI-UCB` (Jin et al., 2020b)) and inherit their regret guarantees without discretizing $\Delta_m$. We also explain below why low-switching planners that reuse past value functions across episodes (e.g., `LSVI-UCB++` (He et al., 2023)) are not directly compatible with arbitrary preference changes.

**Setting and reward confidence event.**    Fix an episodic MDP with horizon $H$ and fixed transition kernels $\{P_h\}_{h=1}^H$. For each episode $k \in [K]$, let $\bar{r}_h^{(k)}(s, a) \in [0, 1]$ denote the conditional mean scalar reward at stage $h$ when taking $(s, a)$ in episode $k$. We allow $\bar{r}^{(k)}$ to be *history-dependent across episodes*: for every $(h, s, a)$, $\bar{r}_h^{(k)}(s, a)$ is measurable w.r.t. the pre-episode history $\mathcal{F}_{k,0}$.

Let $\hat{r}_h^{(k)}(s, a)$ and $\beta_{k,h}^r(s, a) \geq 0$ be any (possibly history-dependent) reward estimator and radius supplied to the planner at the beginning of episode $k$. Define the (simultaneous) reward confidence event

$$\mathcal{E}_r := \left\{ \forall k, h, s, a : |\hat{r}_h^{(k)}(s, a) - \bar{r}_h^{(k)}(s, a)| \leq \beta_{k,h}^r(s, a) \right\}. \tag{B.1}$$

In our paper, $\mathcal{E}_r$ is guaranteed by coordinate-wise ridge regression combined with predictable preferences (Appendix C.11).

**Planner update with reward UCB.**    Consider an episodic optimistic planner that performs an `LSVI-UCB`-style optimistic Bellman backup using the supplied reward UCB $\hat{r}_h^{(k)} + \beta_{k,h}^r$. Concretely, in episode $k$ and stage $h$, define

$$Q_{k,h}(s, a) := \min\left\{ \hat{r}_h^{(k)}(s, a) + \beta_{k,h}^r(s, a) + \widehat{\mathcal{P}}_{k,h} V_{k,h+1}(s, a) + \beta_{k,h}^P(s, a), \ H \right\}, \tag{B.2}$$

with $V_{k,h}(s) := \max_a Q_{k,h}(s, a)$ and $V_{k,H+1} \equiv 0$. Here $\widehat{\mathcal{P}}_{k,h}$ and $\beta_{k,h}^P$ denote the transition estimate and transition bonus supplied by the chosen base planner; we denote by $\mathcal{E}_P$ the corresponding transition-confidence event used in its analysis.

**Episode-wise optimal values.**    Let $V_h^{*,(k)}$ and $Q_h^{*,(k)}$ denote the optimal value and action-value functions for the MDP with transition $\{P_h\}$ and *episode-$k$* mean reward $\bar{r}^{(k)}$. Let $\pi_k$ be the greedy policy induced by $Q_{k,h}$ in episode $k$.

**Lemma B.5** (Episode-wise optimism under changing rewards)**.**    *On $\mathcal{E}_r \cap \mathcal{E}_P$, for every episode $k$, stage $h \in [H]$, and state $s \in \mathcal{S}$,*

$$V_{k,h}(s) \ \geq \ V_h^{*,(k)}(s).$$

**Proof.** Fix an episode $k$. We prove by backward induction on $h$. For $h = H + 1$ the claim holds since both sides are 0. Assume the claim holds at $h + 1$. For any $(s, a)$, on $\mathcal{E}_r$ we have $\bar{r}_h^{(k)}(s, a) \leq \hat{r}_h^{(k)}(s, a) + \beta_{k,h}^r(s, a)$. On $\mathcal{E}_P$, the optimistic transition backup satisfies $[P_h V_{k,h+1}](s, a) \leq \widehat{\mathcal{P}}_{k,h} V_{k,h+1}(s, a) + \beta_{k,h}^P(s, a)$ (the same inequality as used in He et al. (2023)). Combining these and using the induction hypothesis $V_{k,h+1} \geq V_{h+1}^{*,(k)}$ gives

$$\bar{r}_h^{(k)}(s, a) + [P_h V_{h+1}^{*,(k)}](s, a) \leq \hat{r}_h^{(k)}(s, a) + \beta_{k,h}^r(s, a) + \widehat{\mathcal{P}}_{k,h} V_{k,h+1}(s, a) + \beta_{k,h}^P(s, a).$$

The left-hand side equals $Q_h^{*,(k)}(s, a)$ by the Bellman optimality equation, and the right-hand side equals the argument of (B.2); the outer min with $H$ preserves the inequality because values are in $[0, H]$. Taking a maximum over $a$ yields $V_{k,h}(s) \geq V_h^{*,(k)}(s)$. $\qquad\square$

**Lemma B.6** (Regret decomposition with episode-wise rewards). *On $\mathcal{E}_r \cap \mathcal{E}_P$, the instantaneous regret in episode $k$,*

$$\Delta_k := V_1^{*,(k)}(s_{k,1}) - V_1^{\pi_k,(k)}(s_{k,1}),$$

*admits the bound*

$$\Delta_k \leq \sum_{h=1}^H \Big( 2\beta_{k,h}^r(s_{k,h}, a_{k,h}) + 2\beta_{k,h}^P(s_{k,h}, a_{k,h}) \Big) + \sum_{h=1}^H \zeta_{k,h}, \tag{B.3}$$

*where $(\zeta_{k,h})_{k,h}$ is a martingale difference sequence (coming from within-episode transition randomness) bounded by $|\zeta_{k,h}| \leq 2H$.*

**Proof (standard; included for completeness).** Fix episode $k$ and abbreviate $\bar{r}_h := \bar{r}_h^{(k)}$, $\hat{r}_h := \hat{r}_h^{(k)}$, $\beta_h^r := \beta_{k,h}^r$. By Lemma B.5 and the definition of $\Delta_k$,
$$\Delta_k \leq V_{k,1}(s_{k,1}) - V_1^{\pi_k,(k)}(s_{k,1}).$$

Let $a_{k,h} \sim \pi_{k,h}(\cdot \mid s_{k,h})$ and denote $V_h^{\pi_k} := V_h^{\pi_k,(k)}$ for brevity. Using $V_{k,h}(s_{k,h}) = Q_{k,h}(s_{k,h}, a_{k,h})$ for the greedy action and expanding $Q_{k,h}$ via (B.2) (dropping the outer min with $H$, which can only increase the RHS) yields

$$V_{k,h}(s_{k,h}) \leq \hat{r}_h(s_{k,h}, a_{k,h}) + \beta_h^r(s_{k,h}, a_{k,h}) + \widehat{\mathcal{P}}_{k,h} V_{k,h+1}(s_{k,h}, a_{k,h}) + \beta_{k,h}^P(s_{k,h}, a_{k,h}).$$

Subtract the true Bellman equation along $\pi_k$, $V_h^{\pi_k}(s_{k,h}) = \bar{r}_h(s_{k,h}, a_{k,h}) + [P_h V_{h+1}^{\pi_k}](s_{k,h}, a_{k,h})$, to obtain

$$V_{k,h}(s_{k,h}) - V_h^{\pi_k}(s_{k,h}) \leq \underbrace{\hat{r}_h(s_{k,h}, a_{k,h}) - \bar{r}_h(s_{k,h}, a_{k,h})}_{(\star)} + \beta_h^r(s_{k,h}, a_{k,h})$$
$$+ \underbrace{\widehat{\mathcal{P}}_{k,h} V_{k,h+1}(s_{k,h}, a_{k,h}) - [P_h V_{k,h+1}](s_{k,h}, a_{k,h})}_{(\diamond)} + \beta_{k,h}^P(s_{k,h}, a_{k,h})$$
$$+ [P_h(V_{k,h+1} - V_{h+1}^{\pi_k})](s_{k,h}, a_{k,h}).$$

On $\mathcal{E}_r$, $|(\star)| \leq \beta_h^r(s_{k,h}, a_{k,h})$; similarly on $\mathcal{E}_P$, $|(\diamond)| \leq \beta_{k,h}^P(s_{k,h}, a_{k,h})$. Therefore,

$$V_{k,h}(s_{k,h}) - V_h^{\pi_k}(s_{k,h}) \leq 2\beta_h^r(s_{k,h}, a_{k,h}) + 2\beta_{k,h}^P(s_{k,h}, a_{k,h}) + [P_h(V_{k,h+1} - V_{h+1}^{\pi_k})](s_{k,h}, a_{k,h}).$$

Add and subtract the realized next-state values to create a martingale difference:

$$\zeta_{k,h} := \Big( V_{k,h+1}(s_{k,h+1}) - V_{h+1}^{\pi_k}(s_{k,h+1}) \Big) - [P_h(V_{k,h+1} - V_{h+1}^{\pi_k})](s_{k,h}, a_{k,h}).$$

Then $\mathbb{E}[\zeta_{k,h} \mid \mathcal{F}_{k,h}] = 0$ and $|\zeta_{k,h}| \leq 2H$ since values lie in $[0, H]$. Rearranging gives

$$V_{k,h}(s_{k,h}) - V_h^{\pi_k}(s_{k,h}) \leq 2\beta_h^r(s_{k,h}, a_{k,h}) + 2\beta_{k,h}^P(s_{k,h}, a_{k,h}) + \zeta_{k,h} + \Big( V_{k,h+1}(s_{k,h+1}) - V_{h+1}^{\pi_k}(s_{k,h+1}) \Big).$$

Telescoping over $h = 1, \ldots, H$ yields (B.3). $\qquad\square$

**Consequence for reward-modularity.** Lemmas B.5 and B.6 are *episode-wise*: they only require the reward confidence bound (B.1) for the *current* episode $k$. Thus, any base planner whose transition analysis controls the sum of transition bonuses $\sum_{k,h} \beta_{k,h}^P(s_{k,h}, a_{k,h})$ and the associated martingale terms can be plugged into our wrapper even when the scalar reward mean varies across episodes (as long as the planner recomputes its optimistic value functions each episode using the supplied reward oracle). *Importantly*, low-switching variants that reuse previously-computed value functions across episodes (e.g., the rare-switching mechanism in LSVI-UCB++ (He et al., 2023)) are not directly compatible with arbitrary preference changes, because reusing an old $Q$ computed under a different scalar reward can break per-episode optimism.

**Reward-estimation bonuses are lower order.** When rewards are unknown, the regret decomposition (B.3) includes the additional term $\sum_{k,h} \beta_{k,h}^r(s_{k,h}, a_{k,h})$ accounting for reward uncertainty. Under our coordinate-wise ridge regression choice, this term admits a standard elliptical-potential bound.

**Lemma B.7** (Bounding the cumulative reward bonus). *Suppose $\beta_{k,h}^r(s,a) = \alpha_r \|\phi_h(s,a)\|_{\Lambda_{k,h}^{-1}}$ for ridge matrices $\Lambda_{k,h} := \lambda I + \sum_{\tau < k} \phi_h(s_{\tau,h}, a_{\tau,h}) \phi_h(s_{\tau,h}, a_{\tau,h})^\top$ and some $\alpha_r > 0$. Then for any trajectory,*

$$\sum_{k=1}^{K} \sum_{h=1}^{H} \beta_{k,h}^r(s_{k,h}, a_{k,h}) \leq \tilde{O}\left(\alpha_r H \sqrt{dK}\right).$$

**Proof sketch.** For a fixed stage $h$, apply Cauchy–Schwarz: $\sum_{k=1}^{K} \|\phi_h(s_{k,h}, a_{k,h})\|_{\Lambda_{k,h}^{-1}} \leq \sqrt{K \sum_{k=1}^{K} \|\phi_h(s_{k,h}, a_{k,h})\|_{\Lambda_{k,h}^{-1}}^2}$. The sum of squared norms is bounded by the log-determinant (elliptical potential lemma; see, e.g., Abbasi-Yadkori et al. (2011)), giving $\sum_{k=1}^{K} \|\phi_h(s_{k,h}, a_{k,h})\|_{\Lambda_{k,h}^{-1}}^2 \leq \tilde{O}(d)$. Summing over $h \in [H]$ yields the claim. $\square$

**Summary.** Combining (B.3) with (i) the reward-bonus control in Lemma B.7 and (ii) the transition-bonus/martingale bounds of a chosen optimistic base planner yields an overall online regret bound for our wrapper. For example, instantiating the base planner with the standard episodic LSVI-UCB planner (Jin et al., 2020b) (which recomputes optimistic value functions each episode and thus accommodates episode-varying scalar rewards) gives $\mathrm{Reg}(K) \leq \widetilde{\mathcal{O}}(d^{3/2} H^2 \sqrt{K})$, matching Theorem 6.2 in the main text up to log factors and lower-order reward-estimation terms.

### B.7. Near-minimax base planner that recomputes each episode: LSVI-UCB (Hu et al., 2022)

The "recompute every episode" requirement is satisfied by several episodic optimistic algorithms for *linear MDPs*. In particular, Hu et al. (2022) propose an episodic variant of least-squares value iteration, denoted LSVI-UCB, that achieves the near-minimax regret rate for linear MDPs and (crucially for our M1 protocol) performs a full backward dynamic program in *every* episode.

We highlight that this is different from UCRL-VTR$^+$ in Zhou et al. (2021), which is designed for the *linear mixture MDP* model parameterized by features $\phi(s' \mid s, a)$ over state–action–next-state triplets (Definition 1 in that paper) and is not directly applicable to our linear MDP / (state–action)-feature setting.

**Theorem B.8** (Regret of LSVI-UCB). *Reference: Hu et al. (2022), Theorem 6.1.*

*Consider an episodic linear MDP with horizon $H$ and feature dimension $d$ (as in Theorem B.4). Then, with high probability, LSVI-UCB achieves*

$$\mathrm{Reg}(K) \leq \widetilde{\mathcal{O}}\left(Hd\sqrt{HK} + H^3 d^6 + \sqrt{H^7 d^7}\right).$$

*In the regime where the leading term dominates (e.g., for $HK$ sufficiently large as specified in Hu et al. (2022)), this simplifies to*

$$\mathrm{Reg}(K) = \widetilde{\mathcal{O}}\left(d\sqrt{H^3 K}\right).$$

**Plugging LSVI-UCB into our wrapper.** To use LSVI-UCB as BasePlanner in M1, we instantiate it with the episode-dependent optimistic scalar reward oracle $(\hat{r}^{(k)}, \beta_{k,h}^r)$ produced by our reward-learning module. Because the preference $w_k$ is predictable, the scalar mean-reward function $\bar{r}^{(k)}$ is $\mathcal{F}_{k,0}$-measurable; therefore the optimistic value targets in the backward recursion remain measurable w.r.t. the pre-episode filtration. Conditioned on the reward confidence event $\mathcal{E}_r$ in (B.1) and the transition confidence event used by LSVI-UCB, the episode-wise optimism and regret decomposition in

Lemmas B.5 and B.6 continue to hold. Consequently, combining Theorem B.8 with Lemma B.7 yields a near-minimax regret bound for our predictable-preference MORL protocol (up to the additional, lower-order reward-estimation term).

## B.8. External linear MDP regret lower bound

**Theorem B.9** (Linear MDP regret lower bound). *Fix integers $H \geq 1$ (horizon) and $d \geq 1$ (feature dimension). Consider the standard episodic interaction protocol over $K$ episodes: in episode $k$, the learner selects a (possibly randomized) policy $\pi_k$, executes it for $H$ steps, and observes the next state and scalar reward at each step.*

*Assume the MDP belongs to the* linear MDP *class (e.g., Jin et al. (2020b)), meaning that there exists a known feature map $\phi_h : \mathcal{S} \times \mathcal{A} \to \mathbb{R}^d$ with $\|\phi_h(s,a)\|_2 \leq 1$ such that, for each stage $h$, the transition kernel admits a linear representation*

$$P_h(s' \mid s, a) = \langle \phi_h(s, a), \mu_h(s') \rangle, \qquad \forall (s, a, s'),$$

*for some (unknown) vector-valued signed measures $\mu_h(\cdot) \in \mathbb{R}^d$ that induce a valid probability kernel on $\mathcal{S}$. Assume additionally that rewards are scalar and bounded: $r_h(s, a) \in [0, 1]$ almost surely.*

*Let $V^*$ denote the optimal value (under the optimal policy) from the start-state distribution, and let $V^{\pi_k}$ denote the value of $\pi_k$ on episode $k$ from the same start distribution. Define the (expected) episodic regret after $K$ episodes by*

$$\mathrm{Reg}(K) := \sum_{k=1}^{K} (V^* - V^{\pi_k}), \qquad \mathbb{E}[\mathrm{Reg}(K)] := \mathbb{E}[\mathrm{Reg}(K)].$$

*Then there exists an absolute constant $c > 0$ such that for every learning algorithm (possibly randomized), there exists a linear MDP instance satisfying the above assumptions for which*

$$\mathbb{E}[\mathrm{Reg}(K)] \geq c \cdot d\sqrt{H^3 K}.$$

**Reference.** A linear-MDP lower bound of the form $\mathbb{E}[\mathrm{Reg}(K)] \geq c \, d \, \sqrt{H^3 K}$ is given explicitly in Hu et al. (2022, Appendix E, Lemma E.1) (building on the hard instance originally described for the linear-mixture model by Zhou et al. (2021)). The statement above matches the $Hd\sqrt{T}$ scaling in that construction under the standard identification $T = HK$.

**Compatibility with our Assumption 3.3 (vector reward feedback).** Assumption 3.3 in the main text gives the learner more information (the full $m$-vector reward each step). The above lower bound still applies to our MORL protocol because we can restrict to a subclass of instances where only one objective is nonzero. Concretely, fix $m \geq 2$ and define a MORL instance by

- letting objective 1 have the scalar reward $r_{h,1}(s, a) := r_h(s, a)$ from the hard single-objective instance,

- setting all other objectives to zero: $r_{h,i}(s, a) := 0$ for every $i \in \{2, \ldots, m\}$ and all $(h, s, a)$,

- keeping the same transition kernel.

Under a constant preference $w = (1, 0, \ldots, 0)$, the scalarized return equals the single-objective return, and the additional observed zero coordinates provide no extra information about the unknown transition dynamics. Therefore any algorithm for the MORL protocol induces an algorithm for the single-objective protocol with the same interaction information about transitions, and the minimax regret lower bound remains valid.

## B.9. External reward-free exploration / PFE theorem for linear MDPs

**Theorem B.10** (Reward-free exploration; stated from Wagenmaker et al. 2022). *Consider episodic linear MDPs with horizon $H$ and feature dimension $d$. Suppose the scalar reward function is not revealed during exploration; only transitions are observed. Assume the total return is bounded by 1 for any policy (this can be enforced by reward scaling).*

*Then there exists a reward-free exploration algorithm such that, after*

$$N = \widetilde{O}\big(d^2/\varepsilon^2\big)$$

*episodes, with probability at least $1 - \delta$, the algorithm outputs a model or policy class sufficient such that for any reward function revealed after exploration, the resulting planned policy is $\varepsilon$-optimal.*

*Moreover, any algorithm requires*

$$N \geq \Omega(d^2/\varepsilon^2)$$

*in the worst case.*

**Reference:** Wagenmaker et al. (2022). We only use the order of the sample-complexity dependence in $(d, \varepsilon, \delta)$.

**Scaling interface:** If the total return is bounded by $U_{\text{ret}}$ instead of 1, rescaling rewards by $1/U_{\text{ret}}$ changes $\varepsilon$ to $\varepsilon/U_{\text{ret}}$, so the required $N$ scales as $U_{\text{ret}}^2/\varepsilon^2$.

**Lemma B.11** (reward scaling and $\varepsilon$-sample complexity mapping). *Fix $U_0 > 0$ and let $N(\varepsilon, \delta)$ denote the sample complexity of an algorithm on a class of MDPs whose total episodic return satisfies $0 \leq V^\pi \leq U_0$ for all policies $\pi$. Suppose that after $N(\varepsilon, \delta)$ episodes the algorithm outputs a policy $\hat{\pi}$ such that, with probability at least $1 - \delta$,*

$$V^* - V^{\hat{\pi}} \leq \varepsilon.$$

*Then on the same MDP class but with return bound $0 \leq V^\pi \leq U_{\text{ret}}$, the same algorithm attains $V^* - V^{\hat{\pi}} \leq \varepsilon$ (with probability at least $1 - \delta$) after*

$$N\left(\varepsilon \cdot \frac{U_0}{U_{\text{ret}}}, \delta\right)$$

*episodes.*

**Proof.** Consider an instance in the same class except that the total episodic return is bounded by $U_{\text{ret}} > 0$. Define the rescaled reward

$$\tilde{r} := \frac{U_0}{U_{\text{ret}}} r.$$

Let $\tilde{V}^\pi$ denote values in the rescaled MDP. Then for every policy $\pi$,

$$\tilde{V}^\pi = \frac{U_0}{U_{\text{ret}}} V^\pi, \qquad 0 \leq \tilde{V}^\pi \leq U_0.$$

Applying the baseline guarantee to the rescaled instance yields that after $N(\tilde{\varepsilon}, \delta)$ episodes,

$$\tilde{V}^* - \tilde{V}^{\hat{\pi}} \leq \tilde{\varepsilon}$$

with probability at least $1 - \delta$. Converting back to the original scale gives

$$V^* - V^{\hat{\pi}} = \frac{U_{\text{ret}}}{U_0}\left(\tilde{V}^* - \tilde{V}^{\hat{\pi}}\right) \leq \frac{U_{\text{ret}}}{U_0} \tilde{\varepsilon}.$$

Choosing $\tilde{\varepsilon} = \varepsilon \cdot \frac{U_0}{U_{\text{ret}}}$ proves the claim. $\qquad\square$

## B.10. Information-theoretic lower bound for multinomial $\ell_1$ estimation

This appendix proves a high-probability $\ell_1$ estimation lower bound for a categorical (multinomial) distribution. The lemma is later used as a tool in the proof of Theorem 6.13 (explicit model recovery requires estimating $d$ categorical transition kernels).

**Lemma B.12** (Multinomial $\ell_1$ estimation lower bound). *Fix $N \in \mathbb{N}$ and a finite alphabet size $S := |\mathcal{S}| \geq 2$. Let $\hat{p}$ be any (possibly randomized) estimator that, after observing $N$ i.i.d. samples from an unknown categorical distribution $p \in \Delta_S$, returns an estimate $\hat{p} \in \Delta_S$.*

*Suppose that for some $(\varepsilon, \delta) \in (0, 1) \times (0, 1/2)$,*

$$\Pr\left(\|\hat{p} - p\|_1 \leq \varepsilon\right) \geq 1 - \delta \qquad \text{for all } p \in \Delta_S.$$

*Then necessarily*

$$N \geq c_1 \frac{S - 1}{\varepsilon^2} \log\left(\frac{1}{2\delta}\right),$$

*for some absolute constant $c_1 > 0$.*

**Proof.**

**Hard hypercube family.** Let $S := |\mathcal{S}| \geq 3$ (the case $S = 2$ can be handled by a standard two-point argument). For completeness, when $S = 2$, writing $P = (p, 1 - p)$ reduces $\|\hat{P} - P\|_1 \leq \varepsilon_P$ to $|\hat{p} - p| \leq \varepsilon_P/2$, which requires $\Omega(1/\varepsilon_P^2)$ samples by a standard two-point (Le Cam) argument; this matches the statement since $S - 1 = 1$. Define

$$M := \left\lfloor \frac{S - 1}{2} \right\rfloor.$$

Then $2M \leq S - 1$, so the $2M$ states $\{1, 2, \ldots, 2M\}$ form $M$ disjoint pairs:

$$(1, 2), \ (3, 4), \ \ldots, \ (2M - 1, 2M).$$

The remaining states $\{2M + 1, \ldots, S\}$ (there are either 1 or 2 such states) will be left unperturbed.

Fix a perturbation magnitude $\alpha > 0$ satisfying

$$\alpha \leq \frac{3}{4S}.$$

For each sign vector $v = (v_1, \ldots, v_M) \in \{-1, +1\}^M$, define a categorical distribution $P_v \in \Delta(\mathcal{S})$ by

$$P_v(2r - 1) := \frac{1}{S} + v_r\alpha, \qquad P_v(2r) := \frac{1}{S} - v_r\alpha, \qquad r = 1, \ldots, M,$$

and for the remaining states $s \in \{2M + 1, \ldots, S\}$,

$$P_v(s) := \frac{1}{S}.$$

We verify $P_v$ is a probability distribution. First, nonnegativity holds because for $r \leq M$,

$$P_v(2r - 1) \geq \frac{1}{S} - \alpha \geq \frac{1}{S} - \frac{3}{4S} = \frac{1}{4S} \geq 0.$$

Moreover, for the even state $2r$ we have $P_v(2r) = \frac{1}{S} - v_r\alpha \geq \frac{1}{S} - \alpha \geq \frac{1}{4S} \geq 0$, and the remaining states have probability $1/S \geq 0$.

Second, the total mass is

$$\sum_{s=1}^{S} P_v(s) = \sum_{r=1}^{M} \left( \frac{1}{S} + v_r\alpha + \frac{1}{S} - v_r\alpha \right) + \sum_{s=2M+1}^{S} \frac{1}{S} = \sum_{r=1}^{M} \frac{2}{S} + (S - 2M) \cdot \frac{1}{S} = \frac{2M}{S} + \frac{S - 2M}{S} = 1.$$

**Neighbor KL/TV bound.** Fix $r \in [M]$. Let $v^{(r)}$ be the sign vector obtained by flipping the $r$-th coordinate:

$$v_r^{(r)} := -v_r, \qquad v_\ell^{(r)} := v_\ell \text{ for } \ell \neq r.$$

Then $P_v$ and $P_{v^{(r)}}$ differ only on the pair of states $(2r - 1, 2r)$:

$$P_v(2r - 1) = \frac{1}{S} + v_r\alpha, \quad P_v(2r) = \frac{1}{S} - v_r\alpha,$$

$$P_{v^{(r)}}(2r - 1) = \frac{1}{S} - v_r\alpha, \quad P_{v^{(r)}}(2r) = \frac{1}{S} + v_r\alpha.$$

Define

$$p := \frac{1}{S} + \alpha, \qquad q := \frac{1}{S} - \alpha.$$

Then the KL divergence is

$$\mathrm{KL}(P_v \,\|\, P_{v^{(r)}}) = p \log\left( \frac{p}{q} \right) + q \log\left( \frac{q}{p} \right).$$

We simplify the second term:

$$q \log \left( \frac{q}{p} \right) = -q \log \left( \frac{p}{q} \right).$$

Therefore,

$$\mathrm{KL}(P_v \,\|\, P_{v^{(r)}}) = (p - q) \log \left( \frac{p}{q} \right) = 2\alpha \log \left( \frac{p}{q} \right).$$

Next, compute the ratio:

$$\frac{p}{q} = \frac{\frac{1}{S} + \alpha}{\frac{1}{S} - \alpha} = \frac{1 + \alpha S}{1 - \alpha S}.$$

Let $t := \alpha S$. Since $\alpha \leq 3/(4S)$, we have $t \leq 3/4 < 1$. Then

$$\log \left( \frac{p}{q} \right) = \log(1 + t) - \log(1 - t).$$

We bound each term. First, for $t \geq 0$, $\log(1 + t) \leq t$. Second, for $t \in [0, 1)$,

$$-\log(1 - t) = \int_0^t \frac{1}{1 - u} \, du \leq \int_0^t \frac{1}{1 - t} \, du = \frac{t}{1 - t}.$$

Combining,

$$\log(1 + t) - \log(1 - t) \leq t + \frac{t}{1 - t} = \frac{t(2 - t)}{1 - t} \leq \frac{2t}{1 - t}.$$

Therefore,

$$\mathrm{KL}(P_v \,\|\, P_{v^{(r)}}) \leq 2\alpha \cdot \frac{2t}{1 - t} = 4\alpha \cdot \frac{\alpha S}{1 - \alpha S} = \frac{4\alpha^2 S}{1 - \alpha S}.$$

Using $\alpha S \leq 3/4$, we have $1 - \alpha S \geq 1/4$, hence

$$\mathrm{KL}(P_v \,\|\, P_{v^{(r)}}) \leq \frac{4\alpha^2 S}{1/4} = 16\,\alpha^2 S$$

(We will only use that this is at most a numerical constant times $\alpha^2 S$.)

For $n$ i.i.d. samples, the KL divergence between product measures is

$$\mathrm{KL}\big((P_v)^{\otimes n} \,\|\, (P_{v^{(r)}})^{\otimes n}\big) = n \cdot \mathrm{KL}(P_v \,\|\, P_{v^{(r)}}) \leq 16n\alpha^2 S.$$

Let $\mathrm{TV}(\cdot, \cdot)$ denote total variation distance. Pinsker's inequality states that for any two distributions $\mathbb{P}, \mathbb{Q}$,

$$\mathrm{TV}(\mathbb{P}, \mathbb{Q}) \leq \sqrt{\frac{1}{2} \mathrm{KL}(\mathbb{P} \,\|\, \mathbb{Q})}.$$

Applying this to the $n$-sample product distributions yields

$$\mathrm{TV}\big((P_v)^{\otimes n}, (P_{v^{(r)}})^{\otimes n}\big) \leq \sqrt{\frac{1}{2} \cdot 16n\alpha^2 S} = \sqrt{8n\alpha^2 S}.$$

In particular, if

$$n \leq \frac{1}{512\alpha^2 S},$$

then $8n\alpha^2 S \leq 1/64$ and hence

$$\mathrm{TV}\big((P_v)^{\otimes n}, (P_{v^{(r)}})^{\otimes n}\big) \leq \frac{1}{8}.$$

**Assouad argument and $\ell_1$ error.** The following is a standard Assouad-lemma reduction (Assouad, 1983; Pollard et al., 1997); the resulting $\ell_1$ scaling matches classical multinomial minimax rates (e.g., (Kamath et al., 2015)).

Assume that for every $r \in [M]$ and every $v \in \{-1, +1\}^M$, $\mathrm{TV}((P_v)^{\otimes n}, (P_{v^{(r)}})^{\otimes n}) \leq 1/8$. Consider any estimator $\hat{P}$ (hence any induced sign estimator $\hat{v} \in \{-1, +1\}^M$ obtained from $\hat{P}$). Let

$$Y(v) := \sum_{r=1}^{M} \mathbf{1}\{\hat{v}_r \neq v_r\}$$

be the Hamming error.

Fix $r \in [M]$ and define the index set $\mathcal{V}_r^+ := \{v \in \{-1, +1\}^M : v_r = +1\}$. For each $v \in \mathcal{V}_r^+$, let $A_v := \{\hat{v}_r = +1\}$. Then under $P_v$ the coordinate-$r$ error event is $A_v^c$, whereas under $P_{v^{(r)}}$ the coordinate-$r$ error event is $A_v$. By the definition of total variation distance,

$$(P_v)^{\otimes n}(A_v^c) + (P_{v^{(r)}})^{\otimes n}(A_v) \geq 1 - \mathrm{TV}((P_v)^{\otimes n}, (P_{v^{(r)}})^{\otimes n}) \geq \frac{7}{8}.$$

Therefore, averaging over the pair $(v, v^{(r)})$ gives

$$\frac{1}{2}(P_v)^{\otimes n}(\hat{v}_r \neq v_r) + \frac{1}{2}(P_{v^{(r)}})^{\otimes n}(\hat{v}_r \neq v_r^{(r)}) \geq \frac{7}{16}.$$

Since the map $v \mapsto v^{(r)}$ is a bijection between $\mathcal{V}_r^+$ and $\mathcal{V}_r^- := \{v : v_r = -1\}$, we can rewrite the uniform prior average as

$$\frac{1}{2^M} \sum_{v \in \{-1, +1\}^M} (P_v)^{\otimes n}(\hat{v}_r \neq v_r) = \frac{1}{2^{M-1}} \sum_{v \in \mathcal{V}_r^+} \frac{1}{2}\left[(P_v)^{\otimes n}(\hat{v}_r \neq v_r) + (P_{v^{(r)}})^{\otimes n}(\hat{v}_r \neq v_r^{(r)})\right] \geq \frac{7}{16}.$$

Summing over $r$ yields

$$\frac{1}{2^M} \sum_{v \in \{-1, +1\}^M} \mathbb{E}_{(P_v)^{\otimes n}}[Y(v)] \geq \sum_{r=1}^{M} \frac{7}{16} = \frac{7M}{16}.$$

Hence there exists some $v^\star$ such that $\mathbb{E}_{(P_{v^\star})^{\otimes n}}[Y(v^\star)] \geq 7M/16$. To convert this expectation bound into a probability bound, let $a := M/6$ and use

$$\mathbb{E}[Y] \leq a\,\mathbb{P}(Y < a) + M\,\mathbb{P}(Y \geq a) \leq a + (M - a)\,\mathbb{P}(Y \geq a),$$

which implies $\mathbb{P}(Y \geq a) \geq (\mathbb{E}[Y] - a)/(M - a)$. Plugging $\mathbb{E}[Y] \geq 7M/16$ and $a = M/6$ gives

$$(P_{v^\star})^{\otimes n}(Y(v^\star) \geq M/6) \geq \frac{7M/16 - M/6}{M - M/6} = \frac{13}{40}.$$

We show that each wrong bit forces a nontrivial $\ell_1$ error contribution.

Fix $r$ and suppose $v_r = +1$. Then the true pair probabilities are

$$P_v(2r - 1) = \frac{1}{S} + \alpha, \qquad P_v(2r) = \frac{1}{S} - \alpha.$$

If the estimator outputs $\hat{v}_r = -1$, then by definition of $\hat{v}_r$ we have

$$\hat{P}(2r - 1) \leq \hat{P}(2r).$$

Let $x := \hat{P}(2r - 1)$ and $y := \hat{P}(2r)$. Define $a := 1/S + \alpha$ and $b := 1/S - \alpha$, so $a - b = 2\alpha$ and $x - y \leq 0$. Using the triangle inequality in the form $|u| + |v| \geq |u - v|$ with $u = x - a$ and $v = y - b$, we obtain

$$|x - a| + |y - b| \geq |(x - a) - (y - b)| = |(x - y) - (a - b)| = |(x - y) - 2\alpha|.$$

Since $x - y \leq 0$, we have $(x - y) - 2\alpha \leq -2\alpha$, hence

$$|(x - y) - 2\alpha| \geq 2\alpha.$$

Therefore, whenever $\hat{v}_r \neq v_r$, the $\ell_1$ error on the pair $(2r-1, 2r)$ is at least $2\alpha$:

$$|\hat{P}(2r-1) - P_v(2r-1)| + |\hat{P}(2r) - P_v(2r)| \geq 2\alpha.$$

Summing this over all wrong coordinates gives

$$\|\hat{P} - P_v\|_1 \geq 2\alpha \cdot Y(v).$$

We now choose $\alpha$ as a function of the target accuracy $\varepsilon_P$. Set

$$\alpha := 3 \cdot \frac{\varepsilon_P}{M}.$$

Then for $a = M/6$,

$$2\alpha \cdot a = 2 \cdot 3 \cdot \frac{\varepsilon_P}{M} \cdot \frac{M}{6} = \varepsilon_P.$$

Therefore, on the event $\{Y(v) \geq M/6\}$ we have

$$\|\hat{P} - P_v\|_1 \geq 2\alpha \cdot Y(v) \geq 2\alpha \cdot \frac{M}{6} = \varepsilon_P.$$

Assume $n \leq 1/(512\alpha^2 S)$ so that neighbor TV distances are at most $1/8$. Then by a standard Assouad-type argument, there exists $v^\star$ such that

$$(P_{v^\star})^{\otimes n}\left(\|\hat{P} - P_{v^\star}\|_1 \geq \varepsilon_P\right) \geq (P_{v^\star})^{\otimes n}\left(Y(v^\star) \geq M/6\right) \geq \frac{13}{40}.$$

This proves the existence of a hard $P^\star := P_{v^\star}$ such that the failure probability is at least $13/40$ whenever $n \leq 1/(512\alpha^2 S)$.

It remains to rewrite the condition $n \leq 1/(512\alpha^2 S)$ in the desired form. We compute

$$\alpha^2 = (3)^2 \cdot \frac{\varepsilon_P^2}{M^2} = 9 \cdot \frac{\varepsilon_P^2}{M^2}.$$

Hence

$$\frac{1}{512\alpha^2 S} = \frac{1}{512 \cdot 9} \cdot \frac{M^2}{\varepsilon_P^2 S} = \frac{M^2}{4608\, \varepsilon_P^2\, S}.$$

Since $M = \lfloor (S-1)/2 \rfloor \geq (S-2)/2$, we have $M^2 \geq (S-2)^2/4$. For $S \geq 3$, $(S-2)^2/S \geq c'(S-1)$ for a numerical constant $c' > 0$ (one can take $c' = 1/8$ for all $S \geq 3$ by checking the inequality directly). Therefore,

$$\frac{M^2}{S} \geq c''(S-1) \quad \text{for a universal constant } c'' > 0,$$

and hence

$$\frac{1}{512\alpha^2 S} \geq \left(\frac{c''}{4608}\right) \cdot \frac{S-1}{\varepsilon_P^2}.$$

Let $c_0 := c''/4608$. Then if

$$n < c_0 \cdot \frac{S-1}{\varepsilon_P^2},$$

we have $n \leq 1/(512\alpha^2 S)$ and therefore the failure probability is at least $13/40$ for some $P^\star$. This proves Lemma B.12. $\quad\square$

Finally, Theorem 6.13 follows in Appendix C by reducing anchor MDP estimation to $d$ independent multinomial estimations.

## B.11. From explicit-model $\ell_1$ accuracy to $\varepsilon$-decision-optimality (and the resulting $H^2$ dependence)

This appendix proves the conversion used in Theorem 6.14: if we estimate every anchor/base kernel $P_h^j$ to $\ell_1$ accuracy $\varepsilon_P$, then any policy optimal in the estimated model is $\varepsilon$-decision-optimal in the true environment with $\varepsilon \leq 2HU_{\mathrm{ret}}\varepsilon_P$. Combining this with multinomial estimation yields the episode complexity scaling $H^2 U_{\mathrm{ret}}^2/\varepsilon^2$.

**Setup and value bounds.** Fix a preference $w \in \Delta_m$ and define the scalar reward $r^w = \langle w, r \rangle$. Let $\bar{r}_h^w(s, a)$ denote the mean scalar reward. Assume rewards are bounded so that the total return is uniformly bounded:

$$\forall \pi, \forall s : \quad 0 \leq V_1^{\pi,w}(s) \leq U_{\text{ret}}.$$

This implies that for all $h$ and all states $s$,

$$0 \leq V_h^{*,w}(s) \leq U_{\text{ret}}, \qquad 0 \leq \hat{V}_h^{*,w}(s) \leq U_{\text{ret}},$$

where $\hat{V}^{*,w}$ is the optimal value in the estimated model. (The inequality follows because any value—in either the true or the estimated model—is an expected sum of at most $H$ bounded rewards. In particular, under $r^w \in [0, 1]$ one may always take $U_{\text{ret}} := H$, and we keep the notation $U_{\text{ret}}$ for such a uniform bound.)

**From anchor accuracy to mixture accuracy.** Assume the anchored simplex-mixture transition model (Remark 3.7): for each stage $h$ and each $(s, a)$,

$$P_h(\cdot \mid s, a) = \sum_{j=1}^{d} \phi_j(s, a) \, P_h^j(\cdot), \qquad \phi(s, a) \in \Delta_d.$$

Let $\hat{P}_h^j$ be estimates of each base kernel and define the induced model transition estimate

$$\hat{P}_h(\cdot \mid s, a) := \sum_{j=1}^{d} \phi_j(s, a) \, \hat{P}_h^j(\cdot).$$

Assume the anchor $\ell_1$ accuracy event:

$$\forall h \in [H], \forall j \in [d] : \quad \|\hat{P}_h^j - P_h^j\|_1 \leq \varepsilon_P.$$

Then for any $(h, s, a)$,

$$\|\hat{P}_h(\cdot \mid s, a) - P_h(\cdot \mid s, a)\|_1 = \left\| \sum_{j=1}^{d} \phi_j(s, a) \, (\hat{P}_h^j - P_h^j) \right\|_1.$$

Using the triangle inequality for $\ell_1$ and $\phi_j(s, a) \geq 0$,

$$\left\| \sum_{j=1}^{d} \phi_j(s, a) \, (\hat{P}_h^j - P_h^j) \right\|_1 \leq \sum_{j=1}^{d} \phi_j(s, a) \, \|\hat{P}_h^j - P_h^j\|_1 \leq \sum_{j=1}^{d} \phi_j(s, a) \, \varepsilon_P = \varepsilon_P.$$

Therefore,

$$\forall h, s, a : \quad \|\hat{P}_h(\cdot \mid s, a) - P_h(\cdot \mid s, a)\|_1 \leq \varepsilon_P.$$

**Simulation bound.** Let $V_h^{*,w}$ and $\hat{V}_h^{*,w}$ be the optimal value functions in the true and estimated models, respectively, for the same scalar reward function $\bar{r}^w$. Define the uniform optimal-value deviation

$$\Delta_h := \sup_{s \in \mathcal{S}} \left| V_h^{*,w}(s) - \hat{V}_h^{*,w}(s) \right|.$$

We have $\Delta_{H+1} = 0$ because both value functions are 0 at the terminal step.

We now derive a recursion for $\Delta_h$.

For any fixed state $s$ at stage $h$, define:

$$V_h^{*,w}(s) = \max_{a \in \mathcal{A}} \left( \bar{r}_h^w(s, a) + P_h(\cdot \mid s, a)^\top V_{h+1}^{*,w} \right),$$

$$\hat{V}_h^{*,w}(s) = \max_{a \in \mathcal{A}} \left( \bar{r}_h^w(s, a) + \hat{P}_h(\cdot \mid s, a)^\top \hat{V}_{h+1}^{*,w} \right).$$

For each action $a$, define

$$f_a := \bar{r}_h^w(s, a) + P_h(\cdot \mid s, a)^\top V_{h+1}^{*,w}, \qquad g_a := \bar{r}_h^w(s, a) + \hat{P}_h(\cdot \mid s, a)^\top \hat{V}_{h+1}^{*,w}.$$

Then $V_h^{*,w}(s) = \max_a f_a$ and $\hat{V}_h^{*,w}(s) = \max_a g_a$.

We bound the difference between maxima:

$$V_h^{*,w}(s) - \hat{V}_h^{*,w}(s) = \max_a f_a - \max_a g_a \leq \max_a (f_a - g_a) \leq \max_a |f_a - g_a|.$$

For the reverse difference,

$$\hat{V}_h^{*,w}(s) - V_h^{*,w}(s) = \max_a g_a - \max_a f_a \leq \max_a (g_a - f_a) \leq \max_a |f_a - g_a|.$$

Therefore,

$$\left| V_h^{*,w}(s) - \hat{V}_h^{*,w}(s) \right| \leq \max_a |f_a - g_a|.$$

Now expand $f_a - g_a$:

$$f_a - g_a = P_h(\cdot \mid s, a)^\top V_{h+1}^{*,w} - \hat{P}_h(\cdot \mid s, a)^\top \hat{V}_{h+1}^{*,w}.$$

Add and subtract $P_h(\cdot \mid s, a)^\top \hat{V}_{h+1}^{*,w}$:

$$f_a - g_a = P_h(\cdot \mid s, a)^\top \left( V_{h+1}^{*,w} - \hat{V}_{h+1}^{*,w} \right) + \left( P_h(\cdot \mid s, a) - \hat{P}_h(\cdot \mid s, a) \right)^\top \hat{V}_{h+1}^{*,w}.$$

Take absolute values and use the triangle inequality:

$$|f_a - g_a| \leq \left| P_h(\cdot \mid s, a)^\top \left( V_{h+1}^{*,w} - \hat{V}_{h+1}^{*,w} \right) \right| + \left| \left( P_h(\cdot \mid s, a) - \hat{P}_h(\cdot \mid s, a) \right)^\top \hat{V}_{h+1}^{*,w} \right|.$$

For the first term, since $P_h(\cdot \mid s, a)$ is a probability distribution,

$$\left| P_h(\cdot \mid s, a)^\top \left( V_{h+1}^{*,w} - \hat{V}_{h+1}^{*,w} \right) \right| \leq \sup_{s'} \left| V_{h+1}^{*,w}(s') - \hat{V}_{h+1}^{*,w}(s') \right| = \Delta_{h+1}.$$

For the second term, expand and bound by $\ell_1$-$\ell_\infty$ duality:

$$\left| (P_h - \hat{P}_h)^\top \hat{V}_{h+1}^{*,w} \right| = \left| \sum_{s'} (P_h(s') - \hat{P}_h(s')) \hat{V}_{h+1}^{*,w}(s') \right| \leq \sum_{s'} |P_h(s') - \hat{P}_h(s')| \cdot \|\hat{V}_{h+1}^{*,w}\|_\infty = \|P_h - \hat{P}_h\|_1 \cdot \|\hat{V}_{h+1}^{*,w}\|_\infty.$$

Using $\|P_h - \hat{P}_h\|_1 \leq \varepsilon_P$ and $\|\hat{V}_{h+1}^{*,w}\|_\infty \leq U_{\text{ret}}$, we obtain

$$\left| (P_h - \hat{P}_h)^\top \hat{V}_{h+1}^{*,w} \right| \leq \varepsilon_P U_{\text{ret}}.$$

Combining these bounds gives

$$|f_a - g_a| \leq \Delta_{h+1} + \varepsilon_P U_{\text{ret}}.$$

Taking the maximum over $a$ and then the supremum over $s$ yields the recursion

$$\Delta_h \leq \Delta_{h+1} + \varepsilon_P U_{\text{ret}}.$$

Unrolling this recursion from $h = H$ down to $h = 1$ and using $\Delta_{H+1} = 0$ gives

$$\Delta_1 \leq \sum_{h=1}^{H} \varepsilon_P U_{\text{ret}} = H \varepsilon_P U_{\text{ret}}.$$

**Decision optimality.** Let $\hat{\pi}(w)$ be an optimal policy in the estimated model for reward $\bar{r}^w$. Then by definition,

$$\hat{V}_1^{*,w}(s) = \hat{V}_1^{\hat{\pi}(w),w}(s) \quad \text{for all } s.$$

We compare the true optimal value to the value of $\hat{\pi}(w)$ in the true model:

$$V_1^{*,w}(s) - V_1^{\hat{\pi}(w),w}(s) = \left(V_1^{*,w}(s) - \hat{V}_1^{*,w}(s)\right) + \left(\hat{V}_1^{*,w}(s) - \hat{V}_1^{\hat{\pi}(w),w}(s)\right) + \left(\hat{V}_1^{\hat{\pi}(w),w}(s) - V_1^{\hat{\pi}(w),w}(s)\right).$$

The middle term is zero because $\hat{\pi}(w)$ is optimal in the estimated model:

$$\hat{V}_1^{*,w}(s) - \hat{V}_1^{\hat{\pi}(w),w}(s) = 0.$$

Therefore,

$$V_1^{*,w}(s) - V_1^{\hat{\pi}(w),w}(s) = \left(V_1^{*,w}(s) - \hat{V}_1^{*,w}(s)\right) + \left(\hat{V}_1^{\hat{\pi}(w),w}(s) - V_1^{\hat{\pi}(w),w}(s)\right).$$

Taking absolute values, we obtain

$$V_1^{*,w}(s) - V_1^{\hat{\pi}(w),w}(s) \leq \left|V_1^{*,w}(s) - \hat{V}_1^{*,w}(s)\right| + \left|\hat{V}_1^{\hat{\pi}(w),w}(s) - V_1^{\hat{\pi}(w),w}(s)\right|.$$

The first term is bounded by $\Delta_1$ by definition. The second term is an evaluation error for a *fixed* policy, which is not controlled by $\Delta_1$ (since $\Delta_1$ only compares *optimal* value functions). To control it, define for any deterministic (possibly nonstationary) policy $\pi$,

$$\Delta_h^\pi := \sup_{s \in \mathcal{S}} \left|V_h^{\pi,w}(s) - \hat{V}_h^{\pi,w}(s)\right|.$$

Then $\Delta_{H+1}^\pi = 0$ and, for any stage $h$ and state $s$,

$$V_h^{\pi,w}(s) = \bar{r}_h^w(s, \pi_h(s)) + P_h(\cdot \mid s, \pi_h(s))^\top V_{h+1}^{\pi,w}, \qquad \hat{V}_h^{\pi,w}(s) = \bar{r}_h^w(s, \pi_h(s)) + \hat{P}_h(\cdot \mid s, \pi_h(s))^\top \hat{V}_{h+1}^{\pi,w}.$$

Repeating the same argument above (but without the $\max$) yields the recursion

$$\Delta_h^\pi \leq \Delta_{h+1}^\pi + \varepsilon_P \|\hat{V}_{h+1}^{\pi,w}\|_\infty \leq \Delta_{h+1}^\pi + \varepsilon_P U_{\text{ret}},$$

where we used $\|\hat{V}_{h+1}^{\pi,w}\|_\infty \leq U_{\text{ret}}$. Unrolling gives

$$\Delta_1^\pi \leq H\varepsilon_P U_{\text{ret}} \quad \text{for all policies } \pi.$$

Applying this with $\pi = \hat{\pi}(w)$ and using the bound $\Delta_1 \leq H\varepsilon_P U_{\text{ret}}$ established above, we conclude

$$V_1^{*,w}(s) - V_1^{\hat{\pi}(w),w}(s) \leq \Delta_1 + \Delta_1^{\hat{\pi}(w)} \leq H\varepsilon_P U_{\text{ret}} + H\varepsilon_P U_{\text{ret}} = 2H\varepsilon_P U_{\text{ret}}.$$

Thus, to guarantee $\varepsilon$-decision-optimality for all $w$, it suffices to choose

$$\varepsilon_P \leq \frac{\varepsilon}{2HU_{\text{ret}}}.$$

**Anchor sample complexity.** We now show how many samples are required to obtain $\|\hat{P}_h^j - P_h^j\|_1 \leq \varepsilon_P$ for all $(h, j)$ with high probability.

Fix a particular stage $h$ and anchor index $j$. Suppose we visit the anchor $(h, j)$ exactly $n_{\text{anc}}$ times and observe i.i.d. next states

$$X_1, \ldots, X_{n_{\text{anc}}} \sim P_h^j.$$

Let $\hat{P}_h^j$ be the empirical distribution:

$$\hat{P}_h^j(s) := \frac{1}{n_{\text{anc}}} \sum_{t=1}^{n_{\text{anc}}} \mathbf{1}\{X_t = s\}.$$

Let $S_{\text{size}} := |\mathcal{S}|$. For categorical distributions, a standard concentration inequality for empirical distributions implies that for all $\varepsilon_P \in (0, 2]$,

$$\mathbb{P}\left(\|\hat{P}_h^j - P_h^j\|_1 > \varepsilon_P\right) \leq 2^{S_{\text{size}}} \exp\left(-\frac{n_{\text{anc}}\varepsilon_P^2}{2}\right).$$

Equivalently, for a target failure probability $\delta_{hj}$, it suffices to take

$$n_{\mathrm{anc}} \geq 2 \cdot \frac{(S_{\mathrm{size}} \log 2) + \log(1/\delta_{hj})}{\varepsilon_P^2}$$

to ensure $\|\hat{P}_h^j - P_h^j\|_1 \leq \varepsilon_P$ with probability at least $1 - \delta_{hj}$.

We choose $\delta_{hj} := \delta/(Hd)$ and apply a union bound over all $Hd$ pairs:

$$\mathbb{P}\Big(\forall h \in [H], \forall j \in [d] : \|\hat{P}_h^j - P_h^j\|_1 \leq \varepsilon_P\Big) \geq 1 - \sum_{h=1}^{H} \sum_{j=1}^{d} \delta_{hj} = 1 - \delta.$$

Thus, it suffices to take

$$n_{\mathrm{anc}} = \widetilde{O}\Big(\frac{S_{\mathrm{size}} - 1}{\varepsilon_P^2}\Big).$$

Finally, we convert anchor visits to episodes. Each episode yields exactly one transition sample at each stage $h$, hence contributes at most one sample to any fixed anchor $(h, j)$. Thus, obtaining $n_{\mathrm{anc}}$ samples for each of the $d$ anchors at a given stage requires $d\, n_{\mathrm{anc}}$ episodes, which can be achieved by cycling through anchors across episodes (and the same episodes are reused across stages). Hence the total episode complexity for estimating all anchors at all stages is

$$N = \widetilde{O}(d \cdot n_{\mathrm{anc}}) = \widetilde{O}\Big(\frac{d(S_{\mathrm{size}} - 1)}{\varepsilon_P^2}\Big).$$

Substituting $\varepsilon_P = \varepsilon/(2HU_{\mathrm{ret}})$ yields

$$N = \widetilde{O}\Big(\frac{d(S_{\mathrm{size}} - 1)H^2 U_{\mathrm{ret}}^2}{\varepsilon^2}\Big),$$

which is the corrected $H^2$ dependence used in Theorem 6.14. $\qquad\square$

# C. Proofs for New Claims

**Purpose and how to navigate.** This appendix provides detailed proofs for the paper's new technical claims. Whenever a step is a direct application of a standard result (e.g., concentration, self-normalized bounds, basic convex-analytic facts), we invoke the corresponding theorem block from Appendix B rather than re-proving it.

**Quick reference.** Appendix B.1 (at the start of Appendix B) collects the comparison table, key notation, and the one-page main-results map; proofs begin below.

## C.1. Proof of Theorem 6.1 (predictable preference concentration)

Fix $t$. Let $\xi_t \in \mathbb{R}^m$ be the $m$-dimensional martingale difference in Theorem 6.1 and define the scalarized noise

$$\eta_t := \langle w_t, \xi_t \rangle.$$

Since $w_t$ is $\mathcal{F}_{t-1}$-measurable and $\mathbb{E}[\xi_t \mid \mathcal{F}_{t-1}] = 0$ by the martingale difference assumption in Theorem 6.1,

$$\mathbb{E}[\eta_t \mid \mathcal{F}_{t-1}] = \mathbb{E}[\langle w_t, \xi_t \rangle \mid \mathcal{F}_{t-1}] = \langle w_t, \mathbb{E}[\xi_t \mid \mathcal{F}_{t-1}] \rangle = 0.$$

Moreover, because $\xi_t \in [-1, 1]^m$ coordinate-wise and $w_t \in \Delta_m$, we have $\eta_t \in [-1, 1]$ almost surely. Applying the conditional Hoeffding lemma (Theorem B.1) with $(a, b) = (-1, 1)$ yields

$$\mathbb{E}[\exp(\alpha \eta_t) \mid \mathcal{F}_{t-1}] \leq \exp(\alpha^2/2) \quad \forall \alpha \in \mathbb{R},$$

so $\eta_t$ is conditionally 1-subgaussian.

Therefore, $(x_t, \eta_t)$ satisfies the assumptions of the time-uniform self-normalized inequality Theorem B.2: by Theorem 6.1, $x_t$ is $\mathcal{F}_{t-1}$-measurable with $\|x_t\|_2 \leq 1$, and we have established that $\eta_t$ is $\mathcal{F}_t$-measurable with $\mathbb{E}[\eta_t \mid \mathcal{F}_{t-1}] = 0$ and conditionally 1-subgaussian.

Let $\Lambda_0 = \lambda I_d$ for $\lambda \geq 1$ and $\Lambda_t = \Lambda_0 + \sum_{s=1}^t x_s x_s^\top$. Therefore, Theorem B.2 implies that with probability at least $1 - \delta$, for all $t \geq 1$ simultaneously,

$$\Big\| \sum_{s=1}^t x_s \eta_s \Big\|_{\Lambda_t^{-1}} \leq \sqrt{2 \log \Big( \frac{\det(\Lambda_t)^{1/2} \cdot \det(\Lambda_0)^{-1/2}}{\delta} \Big)}.$$

This yields the stated time-uniform self-normalized bound. $\qquad\square$

## C.2. Deferred proofs for Section 4

We provide short proofs for two basic lemmas used in the deployable hypervolume semantics.

**Proof of Lemma 4.2.** Let $\{\pi^j\}_{j \in J}$ be the (finite) support of the episode-start randomization distribution, and let $\alpha$ be the associated probability mass function. By linearity of expectation, the expected return vector is

$$\mathbb{E}[v(\pi)] = \sum_{j \in J} \alpha_j \, v(\pi^j) \in \mathrm{conv}(\mathcal{V}_{\mathrm{det}}) = \mathcal{C}^*.$$

Conversely, any $x \in \mathcal{C}^*$ admits a convex decomposition $x = \sum_{j \in J} \alpha_j v(\pi^j)$ with $\alpha_j \geq 0$ and $\sum_j \alpha_j = 1$. Sampling $\pi^j$ with probability $\alpha_j$ realizes $\mathbb{E}[v(\pi)] = x$. $\qquad\square$

**Proof of Lemma 4.6.** By definition, $h_{\mathcal{C}^*}(w) = \sup_{x \in \mathrm{conv}(\mathcal{V}_{\mathrm{det}})} \langle w, x \rangle$. Since $x \mapsto \langle w, x \rangle$ is linear, its maximum over a convex hull is attained at an extreme point, hence

$$h_{\mathcal{C}^*}(w) = \max_{v \in \mathcal{V}_{\mathrm{det}}} \langle w, v \rangle.$$

For the scalarized reward $\bar{r}_h^w(s, a) = \langle w, \bar{r}_h(s, a) \rangle$, standard finite-horizon dynamic programming yields an optimal deterministic Markov policy, so $V^*(w) = \max_{v \in \mathcal{V}_{\mathrm{det}}} \langle w, v \rangle$. Combining the two displays proves $h_{\mathcal{C}^*}(w) = V^*(w)$. $\qquad\square$

## C.3. Tools for Theorem 4.9: properties of dominated regions and a separation lemma

**Lemma C.1** (dominated regions preserve compactness and convexity). *Let $C \subset [0, U]^m$ be nonempty, compact, and convex. Then its dominated region $K := D_0(C)$ (Definition 4.3) is nonempty, compact, convex, and downward-closed.*

**Proof.** Nonemptiness holds since $0 \in K$ whenever $C \neq \emptyset$. Downward-closedness is immediate from the definition of $D_0(\cdot)$. For compactness, note $K \subset [0, U]^m$ and $K$ is closed: if $y_n \in K$ with $y_n \to y$, pick $x_n \in C$ with $0 \leq y_n \leq x_n$; by compactness of $C$ take a convergent subsequence $x_{n_j} \to x \in C$, and pass to the limit to get $0 \leq y \leq x$, hence $y \in K$. For convexity, if $y^i \in K$ with $0 \leq y^i \leq x^i \in C$, then for any $\alpha \in [0, 1]$ we have $0 \leq \alpha y^1 + (1 - \alpha) y^2 \leq \alpha x^1 + (1 - \alpha) x^2 \in C$ by convexity of $C$, so $\alpha y^1 + (1 - \alpha) y^2 \in K$. $\qquad\square$

**Lemma C.2** (separation using only nonnegative directions for downward-closed convex sets). *Let $K \subset \mathbb{R}_+^m$ be nonempty, closed, convex, and downward-closed. Let $y \in \mathbb{R}_+^m$ satisfy $y \notin K$. Then there exists $w \in \Delta_m$ such that*

$$\langle w, y \rangle > h_K(w),$$

*where $h_K$ is the support function (Definition 4.5).*

**Proof.**

Because $K$ is nonempty, closed, and convex, and $y \notin K$, the separating hyperplane theorem applies. Therefore there exist a vector $u \in \mathbb{R}^m$ with $u \neq 0$ and a real number $a$ such that

$$\langle u, y \rangle > a \geq \sup_{x \in K} \langle u, x \rangle. \tag{C.1}$$

Define the index set of negative coordinates

$$I_- = \{i \in \{1, \ldots, m\} : u_i < 0\}.$$

For any $x \in K$, define $\tilde{x} \in \mathbb{R}_+^m$ by

$$\tilde{x}_i = \begin{cases} 0 & \text{if } i \in I_-, \\ x_i & \text{if } i \notin I_-. \end{cases}$$

Then $\mathbf{0}_m \leq \tilde{x} \leq x$ coordinate-wise. Because $K$ is downward-closed, $\tilde{x}$ is in $K$.

Now compare $\langle u, \tilde{x} \rangle$ and $\langle u, x \rangle$:

$$\langle u, \tilde{x} \rangle = \sum_{i \notin I_-} u_i x_i + \sum_{i \in I_-} u_i \cdot 0 = \sum_{i \notin I_-} u_i x_i = \langle u, x \rangle - \sum_{i \in I_-} u_i x_i.$$

For $i \in I_-$ we have $u_i < 0$ and $x_i \geq 0$ (since $K \subset \mathbb{R}_+^m$), hence $-u_i x_i \geq 0$, so $-\sum_{i \in I_-} u_i x_i \geq 0$. Therefore

$$\langle u, \tilde{x} \rangle \geq \langle u, x \rangle. \tag{C.2}$$

Because for every $x \in K$ there exists $\tilde{x} \in K$ with (C.2), we have

$$\sup_{x \in K} \langle u, x \rangle = \sup_{x \in K} \langle u, \tilde{x} \rangle. \tag{C.3}$$

Define the nonnegative truncation $u_+$ by $(u_+)_i = \max(u_i, 0)$. For any $\tilde{x}$ constructed above, by definition of $\tilde{x}$ we have $\tilde{x}_i = 0$ whenever $u_i < 0$, hence for all $i$:

- if $u_i \geq 0$ then $(u_+)_i = u_i$ and contributes $u_i \tilde{x}_i$,

- if $u_i < 0$ then $(u_+)_i = 0$ and contributes $0 = u_i \tilde{x}_i$ because $\tilde{x}_i = 0$.

Therefore

$$\langle u_+, \tilde{x} \rangle = \langle u, \tilde{x} \rangle \quad \text{for all such } \tilde{x}. \tag{C.4}$$

Because $(u_+)_i = 0$ for $i \in I_-$ and $\tilde{x}_i = x_i$ for $i \notin I_-$ while $\tilde{x}_i = 0$ for $i \in I_-$, we have $\langle u_+, x \rangle = \langle u_+, \tilde{x} \rangle$ for every $x \in K$. Hence

$$\sup_{x \in K} \langle u_+, x \rangle = \sup_{x \in K} \langle u_+, \tilde{x} \rangle.$$

Using (C.4) and then (C.3),

$$\sup_{x \in K} \langle u_+, x \rangle = \sup_{x \in K} \langle u, \tilde{x} \rangle = \sup_{x \in K} \langle u, x \rangle. \tag{C.5}$$

Now use $y \in \mathbb{R}_+^m$. Since $u_+ \geq u$ coordinate-wise and $y \geq 0$ coordinate-wise,

$$\langle u_+, y \rangle \geq \langle u, y \rangle. \tag{C.6}$$

Combining (C.1), (C.5), and (C.6) yields

$$\langle u_+, y \rangle > a \geq \sup_{x \in K} \langle u_+, x \rangle. \tag{C.7}$$

Thus,

$$\langle u_+, y \rangle > \sup_{x \in K} \langle u_+, x \rangle. \tag{C.8}$$

If $u_+$ were the zero vector, then both sides would be 0 because $y$ and $x$ are nonnegative and $u_+$ is zero, contradicting strict inequality. Therefore $u_+ \neq 0$, so its $\ell_1$ norm is positive: $\|u_+\|_1 = \sum_i (u_+)_i > 0$.

Define

$$w = \frac{u_+}{\|u_+\|_1}.$$

Then $w$ has nonnegative coordinates and sums to 1, so $w \in \Delta_m$. Divide (C.8) by $\|u_+\|_1$ to obtain

$$\langle w, y \rangle > \sup_{x \in K} \langle w, x \rangle = h_K(w),$$

where the equality uses the definition of support function (Definition 4.5). This proves the lemma. $\qquad \square$

**Lemma C.3** (support function of dominated region only needs simplex directions). *Let $C \subset \mathbb{R}_+^m$ be nonempty and compact, and let $K = D_0(C)$ (Definition 4.3). Then for every $w \in \Delta_m$,*

$$h_K(w) = h_C(w).$$

**Proof.**

Fix $w \in \Delta_m$. By Definition 4.5 and Definition 4.3,

$$h_K(w) = \sup_{y \in K} \langle w, y \rangle = \sup_{x \in C} \sup_{\mathbf{0}_m \leq y \leq x} \langle w, y \rangle.$$

Because $w$ has nonnegative coordinates (Definition 3.1) and the constraint is coordinate-wise $0 \leq y \leq x$, the inner supremum is attained at $y = x$: for any feasible $y$, coordinate-wise $y_i \leq x_i$ and $w_i \geq 0$ implies $w_i y_i \leq w_i x_i$, summing gives $\langle w, y \rangle \leq \langle w, x \rangle$. Therefore, $\sup_{0 \leq y \leq x} \langle w, y \rangle = \langle w, x \rangle$. Hence

$$h_K(w) = \sup_{x \in C} \langle w, x \rangle = h_C(w),$$

where the last equality uses Definition 4.5. $\qquad\square$

### C.4. Proof of Theorem 4.9 (support error to Hausdorff-$\infty$)

Assume the conditions of Theorem 4.9. Let $K_0, K_0' \subseteq \mathbb{R}_+^m$ denote the compact convex sets in the theorem statement, and define their dominated regions $K := D_0(K_0)$ and $K' := D_0(K_0')$.

Since $K_0$ and $K_0'$ are compact subsets of $\mathbb{R}_+^m$, there exists $U > 0$ such that $K_0 \cup K_0' \subset [0, U]^m$; fix such a $U$.

By Lemma C.1, $K$ and $K'$ are nonempty compact convex subsets of $[0, U]^m$ and are downward-closed. By Lemma C.3, for every $w \in \Delta_m$ we have $h_K(w) = h_{K_0}(w)$ and $h_{K'}(w) = h_{K_0'}(w)$. Hence the theorem's assumption implies

$$\sup_{w \in \Delta_m} \left| h_K(w) - h_{K'}(w) \right| \leq \varepsilon. \tag{C.9}$$

The proof now uses a standard representation of Hausdorff distance between compact convex sets in terms of support functions. We include a self-contained argument specialized to the $\ell_\infty$ / $\ell_1$ dual pair.

**Extending from $\Delta_m$ to the $\ell_1$ unit ball.** Recall that support functions are positively homogeneous: $h_A(\alpha u) = \alpha h_A(u)$ for every $\alpha \geq 0$. Fix any $u \in \mathbb{R}^m$ with $\|u\|_1 \leq 1$.

Because $K \subseteq \mathbb{R}_+^m$ is downward-closed, we may restrict to nonnegative directions. Let $u_+ := u \vee 0$ denote the coordinate-wise positive part of $u$. For any $x \in K$, define $x' \in \mathbb{R}^m$ coordinate-wise by $x_i' := x_i$ if $u_i \geq 0$ and $x_i' := 0$ otherwise. Then $0 \leq x' \leq x$, hence $x' \in K$, and

$$\langle u, x \rangle \leq \langle u, x' \rangle = \langle u_+, x' \rangle \leq h_K(u_+).$$

Taking the supremum over $x \in K$ yields $h_K(u) \leq h_K(u_+)$. Conversely, for any $x \in K$ the same construction gives $\langle u_+, x \rangle = \langle u_+, x' \rangle = \langle u, x' \rangle \leq h_K(u)$, and taking the supremum yields $h_K(u_+) \leq h_K(u)$. The same argument applies to $K'$. Therefore,

$$h_K(u) = h_K(u_+), \qquad h_{K'}(u) = h_{K'}(u_+). \tag{C.10}$$

Now $u_+ \in \mathbb{R}_+^m$ and $\|u_+\|_1 \leq \|u\|_1 \leq 1$. If $u_+ = 0$ then $h_K(u) = h_{K'}(u) = 0$ and the desired bound is trivial. Otherwise let $w := u_+/\|u_+\|_1 \in \Delta_m$. By positive homogeneity and (C.9),

$$\left| h_K(u) - h_{K'}(u) \right| = \left| h_K(u_+) - h_{K'}(u_+) \right| = \|u_+\|_1 \left| h_K(w) - h_{K'}(w) \right| \leq \|u_+\|_1 \varepsilon \leq \varepsilon. \tag{C.11}$$

Thus we have shown

$$\sup_{\|u\|_1 \leq 1} \left| h_K(u) - h_{K'}(u) \right| \leq \varepsilon. \tag{C.12}$$

**Hausdorff–$\infty$ as a support-function discrepancy.** Let $B_\infty := \{z \in \mathbb{R}^m : \|z\|_\infty \leq 1\} = [-1, 1]^m$. For $\delta \geq 0$, the definition of Hausdorff distance implies

$$d_{H,\infty}(K, K') \leq \delta \quad \Longleftrightarrow \quad K \subseteq K' \oplus \delta B_\infty \text{ and } K' \subseteq K \oplus \delta B_\infty,$$

where $\oplus$ denotes Minkowski sum.

For any nonempty compact convex set $A$, its support function satisfies $h_{A \oplus B}(u) = h_A(u) + h_B(u)$. Moreover, the support function of $\delta B_\infty$ is

$$h_{\delta B_\infty}(u) = \delta \|u\|_1,$$

since $\max_{\|z\|_\infty \leq 1} \langle u, z \rangle = \sum_{i=1}^m |u_i|$.

We claim that for nonempty compact convex sets $A, B$,

$$A \subseteq B \oplus \delta B_\infty \quad \Longleftrightarrow \quad h_A(u) \leq h_B(u) + \delta \|u\|_1 \text{ for all } u \in \mathbb{R}^m. \tag{C.13}$$

The forward direction holds because inclusion implies $h_A \leq h_{B \oplus \delta B_\infty} = h_B + h_{\delta B_\infty}$. For the reverse direction, if $A \nsubseteq B \oplus \delta B_\infty$, pick $a \in A \setminus (B \oplus \delta B_\infty)$. Since $B \oplus \delta B_\infty$ is closed and convex, the separating hyperplane theorem yields some $u$ such that $\langle u, a \rangle > \sup_{x \in B \oplus \delta B_\infty} \langle u, x \rangle = h_B(u) + \delta \|u\|_1$, contradicting (C.13). This proves the equivalence.

Applying (C.13) to both inclusions in the definition of $d_{H,\infty}$, we obtain the standard identity

$$d_{H,\infty}(K, K') = \sup_{\|u\|_1 \leq 1} \left| h_K(u) - h_{K'}(u) \right|. \tag{C.14}$$

Combining (C.12) with (C.14) gives $d_{H,\infty}(K, K') \leq \varepsilon$, completing the proof. $\qquad \square$

## C.5. Proof of Theorem 4.10 (HV Lipschitz upper bound)

We prove Theorem 4.10. Let $U > 0$ and let $K, K' \subseteq [0, U]^m$ be compact, convex, and downward-closed. Assume that

$$d_{H,\infty}(K, K') \leq \varepsilon.$$

We write $e_i$ for the $i$-th standard basis vector in $\mathbb{R}^m$. For $\varepsilon \geq 0$, define the axis-aligned cube

$$Q_\varepsilon := [0, \varepsilon]^m \subseteq \mathbb{R}^m.$$

For any set $A \subseteq \mathbb{R}^m$, define its (one-sided) coordinate-wise $\varepsilon$-expansion

$$A \oplus Q_\varepsilon := \{a + z : a \in A, z \in Q_\varepsilon\}, \qquad A^{(+)} := (A \oplus Q_\varepsilon) \cap [0, U]^m.$$

**From Hausdorff–$\infty$ closeness to one-sided coordinate-wise expansions.** We first prove the inclusion

$$K \subseteq (K')^{(+)}.$$

Fix any $x \in K$. By Definition 4.7 (Hausdorff-$\infty$ distance), the condition $d_{H,\infty}(K, K') \leq \varepsilon$ implies

$$\sup_{x \in K} \operatorname{dist}_\infty(x, K') \leq \varepsilon.$$

Therefore, for this fixed $x$, we have $\operatorname{dist}_\infty(x, K') \leq \varepsilon$, and since $K'$ is compact, the infimum in $\operatorname{dist}_\infty(x, K')$ is attained; hence there exists $y \in K'$ such that

$$\|x - y\|_\infty \leq \varepsilon.$$

Define the coordinate-wise minimum vector $\tilde{y} \in \mathbb{R}^m$ by

$$\tilde{y}_j := \min(x_j, y_j) \quad \text{for each coordinate } j \in \{1, \ldots, m\}.$$

We verify two facts:

1. $\tilde{y} \in K'$. By construction, $\tilde{y}_j \leq y_j$ for every coordinate $j$, hence $\tilde{y} \leq y$ coordinate-wise. Since $y \in K'$ and $K'$ is downward-closed, the definition of downward-closedness implies $\tilde{y} \in K'$.

2. $x - \tilde{y} \in Q_\varepsilon$. Fix any coordinate $j$. We have $0 \leq x_j - \tilde{y}_j$ because $\tilde{y}_j \leq x_j$ by definition of the coordinate-wise minimum. Moreover,

   - if $y_j \leq x_j$, then $\tilde{y}_j = y_j$ and

   $$x_j - \tilde{y}_j = x_j - y_j \leq |x_j - y_j| \leq \|x - y\|_\infty \leq \varepsilon;$$

   - if $y_j > x_j$, then $\tilde{y}_j = x_j$ and $x_j - \tilde{y}_j = 0 \leq \varepsilon$.

   In both cases, $0 \leq x_j - \tilde{y}_j \leq \varepsilon$. Since this holds for every coordinate $j$, we obtain $x - \tilde{y} \in [0, \varepsilon]^m = Q_\varepsilon$.

Combining the two verified facts, we have written

$$x = \tilde{y} + (x - \tilde{y}), \quad \text{with } \tilde{y} \in K' \text{ and } (x - \tilde{y}) \in Q_\varepsilon.$$

Therefore $x \in K' \oplus Q_\varepsilon$. Since also $x \in K \subseteq [0, U]^m$, we conclude

$$x \in (K' \oplus Q_\varepsilon) \cap [0, U]^m = (K')^{(+)}.$$

Because $x \in K$ was arbitrary, this proves

$$K \subseteq (K')^{(+)}. \tag{C.15}$$

Since Lebesgue volume $\mathrm{Vol}_m(\cdot)$ is monotone under set inclusion, (C.15) implies

$$\mathrm{Vol}_m(K) \leq \mathrm{Vol}_m((K')^{(+)}).$$

Subtracting $\mathrm{Vol}_m(K')$ from both sides yields

$$\mathrm{Vol}_m(K) - \mathrm{Vol}_m(K') \leq \mathrm{Vol}_m((K')^{(+)}) - \mathrm{Vol}_m(K'). \tag{C.16}$$

We will later also need the reverse inclusion

$$K' \subseteq K^{(+)}.$$

We prove it now, using the other half of Definition 4.7. The assumption $d_{H,\infty}(K, K') \leq \varepsilon$ also implies

$$\sup_{y \in K'} \mathrm{dist}_\infty(y, K) \leq \varepsilon.$$

Fix any $y \in K'$. Then $\mathrm{dist}_\infty(y, K) \leq \varepsilon$. Since $K$ is compact, the infimum in $\mathrm{dist}_\infty(y, K)$ is attained, so there exists $x \in K$ with $\|y - x\|_\infty \leq \varepsilon$. Define the coordinate-wise minimum $\tilde{x}$ by

$$\tilde{x}_j := \min(y_j, x_j) \quad \text{for each } j \in \{1, \ldots, m\}.$$

Because $\tilde{x} \leq x$ coordinate-wise and $x \in K$, downward-closedness of $K$ implies $\tilde{x} \in K$. For each coordinate $j$, the same two-case argument as above gives

$$0 \leq y_j - \tilde{x}_j \leq \varepsilon,$$

hence $y - \tilde{x} \in Q_\varepsilon$. Therefore $y \in K \oplus Q_\varepsilon$, and since $y \in [0, U]^m$ we conclude $y \in K^{(+)} = (K \oplus Q_\varepsilon) \cap [0, U]^m$. As $y \in K'$ was arbitrary,

$$K' \subseteq K^{(+)}. \tag{C.17}$$

**Volume increase under the $\varepsilon$-expansion.** We next express $(K')^{(+)}$ as a sequence of $m$ one-dimensional expansions. Define

$$B_0 := K'.$$

For each $i \in \{1, \ldots, m\}$, define

$$B_i := \{x \in [0, U]^m : \exists t \in [0, \varepsilon] \text{ such that } x - te_i \in B_{i-1}\}.$$

Also define for each $i \in \{0, 1, \ldots, m\}$ the truncated cube

$$Q_\varepsilon^{(i)} := \{z \in \mathbb{R}^m : 0 \leq z_j \leq \varepsilon \text{ for } j \leq i, \text{ and } z_j = 0 \text{ for } j > i\}.$$

Equivalently, $Q_\varepsilon^{(i)} = [0, \varepsilon]^i \times \{0\}^{m-i}$ (with the convention $Q_\varepsilon^{(0)} = \{0\}^m$).

We claim that for every $i \in \{0, 1, \ldots, m\}$,

$$B_i = (K' \oplus Q_\varepsilon^{(i)}) \cap [0, U]^m. \tag{C.18}$$

**Proof of (C.18) by induction on $i$.**

**Base case ($i = 0$).** We have $B_0 = K'$ by definition. Also $Q_\varepsilon^{(0)} = \{0\}^m$, hence $K' \oplus Q_\varepsilon^{(0)} = K'$. Since $K' \subseteq [0, U]^m$, we get $(K' \oplus Q_\varepsilon^{(0)}) \cap [0, U]^m = K' = B_0$. Thus (C.18) holds for $i = 0$.

**Inductive step.** Fix an index $i \in \{1, \ldots, m\}$ and assume (C.18) holds for $i - 1$, i.e.

$$B_{i-1} = (K' \oplus Q_\varepsilon^{(i-1)}) \cap [0, U]^m.$$

We prove (C.18) for $i$ by proving two set inclusions.

**(a) Show $B_i \subseteq (K' \oplus Q_\varepsilon^{(i)}) \cap [0, U]^m$.** Take any $x \in B_i$. By definition of $B_i$, there exists $t \in [0, \varepsilon]$ such that $x - te_i \in B_{i-1}$. By the inductive hypothesis, $x - te_i$ belongs to $(K' \oplus Q_\varepsilon^{(i-1)}) \cap [0, U]^m$, so there exist $y \in K'$ and $z \in Q_\varepsilon^{(i-1)}$ such that

$$x - te_i = y + z.$$

Rearranging gives $x = y + (z + te_i)$. We check that $z + te_i \in Q_\varepsilon^{(i)}$:

- For any coordinate $j < i$, $(z + te_i)_j = z_j$ and $0 \leq z_j \leq \varepsilon$ because $z \in Q_\varepsilon^{(i-1)}$.
- For coordinate $j = i$, $(z + te_i)_i = t$ and $t \in [0, \varepsilon]$.
- For any coordinate $j > i$, $(z + te_i)_j = z_j = 0$ because $z \in Q_\varepsilon^{(i-1)}$ forces all coordinates $> i - 1$ to be zero.

Therefore $z + te_i \in Q_\varepsilon^{(i)}$ and hence $x \in K' \oplus Q_\varepsilon^{(i)}$. Also, by definition of $B_i$, we already have $x \in [0, U]^m$. Thus $x \in (K' \oplus Q_\varepsilon^{(i)}) \cap [0, U]^m$, proving inclusion (a).

**(b) Show $(K' \oplus Q_\varepsilon^{(i)}) \cap [0, U]^m \subseteq B_i$.** Take any $x \in (K' \oplus Q_\varepsilon^{(i)}) \cap [0, U]^m$. Then there exist $y \in K'$ and $z \in Q_\varepsilon^{(i)}$ such that $x = y + z$. Define $t := z_i$. By definition of $Q_\varepsilon^{(i)}$, we have $t \in [0, \varepsilon]$. Define $z' := z - te_i$. We check $z' \in Q_\varepsilon^{(i-1)}$:

- For $j < i$, $z'_j = z_j$ and $0 \leq z_j \leq \varepsilon$.
- For $j = i$, $z'_i = z_i - t = 0$.
- For $j > i$, $z'_j = z_j = 0$ because $z \in Q_\varepsilon^{(i)}$.

Therefore $z' \in Q_\varepsilon^{(i-1)}$. Now compute

$$x - te_i = (y + z) - te_i = y + (z - te_i) = y + z'.$$

Hence $x - te_i \in K' \oplus Q_\varepsilon^{(i-1)}$. We also have $x - te_i \in [0, U]^m$ because $x \in [0, U]^m$, and in the $i$-th coordinate we have

$$(x - te_i)_i = x_i - t = (y_i + z_i) - z_i = y_i,$$

and $y_i \in [0, U]$ since $y \in K' \subseteq [0, U]^m$. Thus $x - te_i \in (K' \oplus Q_\varepsilon^{(i-1)}) \cap [0, U]^m = B_{i-1}$ by the inductive hypothesis. With this $t \in [0, \varepsilon]$, the definition of $B_i$ gives $x \in B_i$. This proves inclusion (b).

Combining (a) and (b) establishes (C.18) for index $i$.

This completes the induction and proves (C.18) for all $i$.

Setting $i = m$ in (C.18) yields $Q_\varepsilon^{(m)} = Q_\varepsilon$ and therefore

$$B_m = (K' \oplus Q_\varepsilon) \cap [0, U]^m = (K')^{(+)}. \tag{C.19}$$

We prove that for each $i \in \{1, \dots, m\}$,

$$\text{Vol}_m(B_i) - \text{Vol}_m(B_{i-1}) \leq \varepsilon \cdot U^{m-1}. \tag{C.20}$$

We first show that each $B_i$ is downward-closed. The set $B_0 = K'$ is downward-closed by assumption. Fix $i \in \{1, \dots, m\}$ and assume $B_{i-1}$ is downward-closed. We prove $B_i$ is downward-closed.

Take any $x \in B_i$ and any $x' \in [0, U]^m$ such that $x' \leq x$ coordinate-wise. Because $x \in B_i$, by definition there exists $t \in [0, \varepsilon]$ such that

$$x - te_i \in B_{i-1}.$$

Define

$$t' := \min(t, x_i').$$

Then $t' \in [0, \varepsilon]$ because $t \in [0, \varepsilon]$ and $x_i' \in [0, U]$ implies $t' \leq t \leq \varepsilon$, and $t' \geq 0$. We now compare $x' - t'e_i$ and $x - te_i$ coordinate-wise.

- For any coordinate $j \neq i$, we have
$$(x' - t'e_i)_j = x_j' \leq x_j = (x - te_i)_j.$$

- For coordinate $j = i$, we consider two cases:
  - If $t' = t$, then $t \leq x_i'$, and
  $$(x' - t'e_i)_i = x_i' - t \leq x_i - t = (x - te_i)_i$$
  because $x_i' \leq x_i$.
  - If $t' = x_i'$, then $(x' - t'e_i)_i = x_i' - x_i' = 0$. Also, since $x - te_i \in [0, U]^m$, we have $x_i - t \geq 0$, hence
  $$0 = (x' - t'e_i)_i \leq x_i - t = (x - te_i)_i.$$

In both cases, we obtain

$$x' - t'e_i \leq x - te_i \quad \text{coordinate-wise.}$$

Since $x - te_i \in B_{i-1}$ and $B_{i-1}$ is downward-closed, it follows that $x' - t'e_i \in B_{i-1}$. With $t' \in [0, \varepsilon]$, the definition of $B_i$ implies $x' \in B_i$. This shows $B_i$ is downward-closed. By induction, every $B_i$ is downward-closed.

Now fix an index $i \in \{1, \dots, m\}$. We apply Tonelli's theorem (equivalently Fubini's theorem for nonnegative measurable functions) to the indicator functions of $B_i$ and $B_{i-1}$. By (C.18), each $B_i$ is compact (hence Lebesgue measurable) and contained in the bounded cube $[0, U]^m$, so its indicator function is integrable and Tonelli applies.

For $x \in [0, U]^m$, write $x = (x_{-i}, x_i)$ where $x_{-i} \in [0, U]^{m-1}$ denotes all coordinates except the $i$-th. Define the slice-length functions

$$\ell_i(x_{-i}) := \int_0^U \mathbf{1}_{B_i}(x_{-i}, t)\, dt, \qquad \ell_{i-1}(x_{-i}) := \int_0^U \mathbf{1}_{B_{i-1}}(x_{-i}, t)\, dt.$$

Tonelli's theorem gives

$$\mathrm{Vol}_m(B_i) = \int_{[0,U]^{m-1}} \ell_i(x_{-i})\, dx_{-i}, \qquad \mathrm{Vol}_m(B_{i-1}) = \int_{[0,U]^{m-1}} \ell_{i-1}(x_{-i})\, dx_{-i}.$$

Subtracting the second identity from the first yields

$$\mathrm{Vol}_m(B_i) - \mathrm{Vol}_m(B_{i-1}) = \int_{[0,U]^{m-1}} \big(\ell_i(x_{-i}) - \ell_{i-1}(x_{-i})\big)\, dx_{-i}. \tag{C.21}$$

We now upper bound the integrand pointwise. Fix $x_{-i} \in [0, U]^{m-1}$. Consider the one-dimensional slices

$$S_i(x_{-i}) := \{t \in [0, U] : (x_{-i}, t) \in B_i\}, \qquad S_{i-1}(x_{-i}) := \{t \in [0, U] : (x_{-i}, t) \in B_{i-1}\}.$$

Because $B_i$ and $B_{i-1}$ are downward-closed, each slice is downward-closed in $[0, U]$ in the following sense: if $t \in S_i(x_{-i})$ and $0 \le t' \le t$, then $(x_{-i}, t') \le (x_{-i}, t)$ coordinate-wise and therefore $(x_{-i}, t') \in B_i$, so $t' \in S_i(x_{-i})$. The same statement holds for $S_{i-1}(x_{-i})$.

Define the slice suprema

$$b_i(x_{-i}) := \sup S_i(x_{-i}), \qquad b_{i-1}(x_{-i}) := \sup S_{i-1}(x_{-i}),$$

with the convention $\sup \emptyset = 0$.

We claim that

$$\ell_i(x_{-i}) = b_i(x_{-i}), \qquad \ell_{i-1}(x_{-i}) = b_{i-1}(x_{-i}). \tag{C.22}$$

We prove the first equality; the second follows by the same reasoning with $B_{i-1}$.

If $S_i(x_{-i})$ is empty, then $b_i(x_{-i}) = 0$ by convention, and $\mathbf{1}_{B_i}(x_{-i}, t) = 0$ for all $t$, hence $\ell_i(x_{-i}) = 0 = b_i(x_{-i})$.

If $S_i(x_{-i})$ is nonempty, then $b_i(x_{-i}) \in (0, U]$. We show that

$$[0, b_i(x_{-i})) \subseteq S_i(x_{-i}) \subseteq [0, b_i(x_{-i})].$$

The inclusion $S_i(x_{-i}) \subseteq [0, b_i(x_{-i})]$ holds by definition of supremum. To prove $[0, b_i(x_{-i})) \subseteq S_i(x_{-i})$, fix any $t$ with $0 \le t < b_i(x_{-i})$. Since $t < \sup S_i(x_{-i})$, by the definition of supremum there exists $t' \in S_i(x_{-i})$ such that $t < t' \le b_i(x_{-i})$. Because the slice $S_i(x_{-i})$ is downward-closed in $[0, U]$ and $0 \le t \le t'$, we obtain $t \in S_i(x_{-i})$. This proves $[0, b_i(x_{-i})) \subseteq S_i(x_{-i})$. Therefore the one-dimensional Lebesgue measure of $S_i(x_{-i})$ equals $b_i(x_{-i})$ (both $[0, b)$ and $[0, b]$ have measure $b$). Since $\ell_i(x_{-i}) = \int_0^U \mathbf{1}_{S_i(x_{-i})}(t)\, dt$ equals that one-dimensional measure, we conclude $\ell_i(x_{-i}) = b_i(x_{-i})$. This proves (C.22).

Next we bound $b_i(x_{-i}) - b_{i-1}(x_{-i})$. Take any $t \in S_i(x_{-i})$. By definition of $B_i$, there exists $s \in [0, \varepsilon]$ such that $(x_{-i}, t) - se_i = (x_{-i}, t - s) \in B_{i-1}$. This means $t - s \in S_{i-1}(x_{-i})$. Therefore $t - s \le b_{i-1}(x_{-i})$, which implies

$$t \le b_{i-1}(x_{-i}) + s \le b_{i-1}(x_{-i}) + \varepsilon.$$

Also, by construction $t \in [0, U]$, so $t \le U$. Taking the supremum over all $t \in S_i(x_{-i})$ yields

$$b_i(x_{-i}) \le \min\big(U, b_{i-1}(x_{-i}) + \varepsilon\big).$$

Since $\min(U, b_{i-1}(x_{-i}) + \varepsilon) - b_{i-1}(x_{-i}) \le \varepsilon$, we obtain

$$b_i(x_{-i}) - b_{i-1}(x_{-i}) \le \varepsilon.$$

Using (C.22), this is exactly the pointwise bound

$$\ell_i(x_{-i}) - \ell_{i-1}(x_{-i}) \le \varepsilon \quad \text{for all } x_{-i} \in [0, U]^{m-1}. \tag{C.23}$$

Substituting (C.23) into (C.21) gives

$$\mathrm{Vol}_m(B_i) - \mathrm{Vol}_m(B_{i-1}) \le \int_{[0,U]^{m-1}} \varepsilon\, dx_{-i} = \varepsilon \cdot \mathrm{Vol}_{m-1}([0, U]^{m-1}) = \varepsilon \cdot U^{m-1}.$$

This proves (C.20).

Summing (C.20) over $i = 1, \ldots, m$ yields

$$\mathrm{Vol}_m(B_m) - \mathrm{Vol}_m(B_0) = \sum_{i=1}^{m} \big(\mathrm{Vol}_m(B_i) - \mathrm{Vol}_m(B_{i-1})\big) \leq \sum_{i=1}^{m} \varepsilon U^{m-1} = m\varepsilon U^{m-1}. \tag{C.24}$$

Since $B_0 = K'$ and $B_m = (K')^{(+)}$ by (C.19), (C.24) becomes

$$\mathrm{Vol}_m((K')^{(+)}) - \mathrm{Vol}_m(K') \leq m\varepsilon U^{m-1}. \tag{C.25}$$

Combining (C.16) and (C.25) gives

$$\mathrm{Vol}_m(K) - \mathrm{Vol}_m(K') \leq \mathrm{Vol}_m((K')^{(+)}) - \mathrm{Vol}_m(K') \leq m\varepsilon U^{m-1}. \tag{C.26}$$

We now derive the reverse inequality. By (C.17), we have $K' \subseteq K^{(+)}$, hence by monotonicity of volume,

$$\mathrm{Vol}_m(K') \leq \mathrm{Vol}_m(K^{(+)}).$$

Subtracting $\mathrm{Vol}_m(K)$ from both sides yields

$$\mathrm{Vol}_m(K') - \mathrm{Vol}_m(K) \leq \mathrm{Vol}_m(K^{(+)}) - \mathrm{Vol}_m(K). \tag{C.27}$$

We bound the right-hand side using the same coordinate-wise expansion argument, now starting from $K$. Define

$$C_0 := K, \quad C_i := \{x \in [0, U]^m : \exists t \in [0, \varepsilon] \text{ such that } x - te_i \in C_{i-1}\} \quad \text{for } i = 1, \ldots, m.$$

Repeating the proof of (C.20) with $C_i$ in place of $B_i$ yields, for each $i$,

$$\mathrm{Vol}_m(C_i) - \mathrm{Vol}_m(C_{i-1}) \leq \varepsilon U^{m-1}.$$

Summing over $i = 1, \ldots, m$ gives

$$\mathrm{Vol}_m(C_m) - \mathrm{Vol}_m(K) \leq m\varepsilon U^{m-1}. \tag{C.28}$$

The same induction proof as above yields $C_m = K^{(+)} = (K \oplus Q_\varepsilon) \cap [0, U]^m$. Therefore (C.28) becomes

$$\mathrm{Vol}_m(K^{(+)}) - \mathrm{Vol}_m(K) \leq m\varepsilon U^{m-1}. \tag{C.29}$$

Combining (C.27) and (C.29) gives

$$\mathrm{Vol}_m(K') - \mathrm{Vol}_m(K) \leq m\varepsilon U^{m-1}. \tag{C.30}$$

Finally, (C.26) and (C.30) together imply

$$|\mathrm{Vol}_m(K) - \mathrm{Vol}_m(K')| \leq m\varepsilon U^{m-1}.$$

This is exactly the claim of Theorem 4.10. $\qquad\square$

### C.6. Proof of Theorem 6.5 (HV Lipschitz lower bound; local $\varepsilon$ regime)

Fix $m \geq 2$ and $U > 0$. Let $\varepsilon$ satisfy $0 < \varepsilon \leq U/m$ (which in particular implies $\varepsilon \leq U/(m-1)$ for $m \geq 2$).

Define the downward-closed sets

$$K = [0, U]^m, \qquad K' = [0, U - \varepsilon]^m.$$

Both are subsets of $[0, U]^m$ and downward-closed.

**Hausdorff–$\infty$ distance.** We show $d_{H,\infty}(K, K') = \varepsilon$.

- First, $K' \subset K$, so $\sup_{y \in K'} \mathrm{dist}_\infty(y, K) = 0$.

- Next, consider the point $x = (U, \ldots, U) \in K$. Any point $y \in K'$ has all coordinates $\leq U - \varepsilon$, so $\|x - y\|_\infty \geq \varepsilon$. In particular, $\mathrm{dist}_\infty(x, K') \geq \varepsilon$.

Conversely, for any $y \in K$, define $y'$ by truncating coordinates: $y_i' = \min(y_i, U - \varepsilon)$. Then $y'$ is in $K'$ and $\|y - y'\|_\infty \leq \varepsilon$. Therefore $\mathrm{dist}_\infty(y, K') \leq \varepsilon$ for all $y \in K$, so $\sup_{y \in K} \mathrm{dist}_\infty(y, K') \leq \varepsilon$.

Combining, we get $\sup_{y \in K} \mathrm{dist}_\infty(y, K') = \varepsilon$, hence by Definition 4.7, $d_{H,\infty}(K, K') = \varepsilon$.

**Volume difference.** Because $K$ and $K'$ are axis-aligned boxes,

$$\text{Vol}_m(K) = U^m, \qquad \text{Vol}_m(K') = (U - \varepsilon)^m.$$

So

$$\text{Vol}_m(K) - \text{Vol}_m(K') = U^m - (U - \varepsilon)^m. \tag{C.31}$$

**Lower bounding the volume difference.** Let $t = \varepsilon/U$. Then $t \in (0, 1/(m-1)]$ by the $\varepsilon$ condition. We can rewrite $(U - \varepsilon)^m = U^m(1 - t)^m$, so (C.31) becomes $\text{Vol}_m(K) - \text{Vol}_m(K') = U^m[1 - (1 - t)^m]$.

We now justify the quadratic upper bound on $(1 - t)^m$ using Taylor's theorem. Define $f(u) := (1 - u)^m$. Since $t \leq 1/(m-1) \leq 1$, we have $t \in [0, 1]$. By Taylor's theorem with Lagrange remainder at $u = 0$, there exists $u^\star \in (0, t)$ such that

$$f(t) = f(0) + f'(0)t + \frac{f''(0)}{2}t^2 + \frac{f^{(3)}(u^\star)}{6}t^3.$$

Here $f(0) = 1$, $f'(0) = -m$, $f''(0) = m(m-1)$, and for any $u \in [0, 1]$,

$$f^{(3)}(u) = -m(m-1)(m-2)(1-u)^{m-3} \leq 0$$

(with equality when $m = 2$). Therefore the remainder term is non-positive, which yields

$$(1 - t)^m \leq 1 - mt + \frac{m(m-1)}{2}t^2. \tag{C.32}$$

Rearrange (C.32):

$$1 - (1 - t)^m \geq mt - \frac{m(m-1)}{2}t^2 = mt\left(1 - \frac{m-1}{2}t\right). \tag{C.33}$$

Since $t \leq 1/(m-1)$, we have $\frac{m-1}{2}t \leq \frac{1}{2}$, hence $1 - \frac{m-1}{2}t \geq \frac{1}{2}$. Plugging into (C.33) yields

$$1 - (1 - t)^m \geq \frac{m}{2}t. \tag{C.34}$$

Multiply (C.34) by $U^m$ and substitute $t = \varepsilon/U$:

$$U^m[1 - (1 - t)^m] \geq U^m \cdot \frac{m}{2} \cdot \frac{\varepsilon}{U} = \frac{m}{2}U^{m-1}\varepsilon.$$

Combine with (C.31) to get $\text{Vol}_m(K) - \text{Vol}_m(K') \geq \frac{m}{2}U^{m-1}\varepsilon$. This proves Theorem 6.5. $\qquad\square$

### C.7. Proof of Theorem 6.7 (discrete weights to HV gap)

Assume the conditions of Theorem 6.7. Fix any $w \in \Delta_m$ and let $\bar{w}$ be a nearest grid point in $\mathcal{W}_L$ in $\ell_1$ distance, i.e., $\|w - \bar{w}\|_1 \leq \eta(\mathcal{W}_L)$.

We first establish a Lipschitz property of linear functionals over $[0, U]^m$:

**Lemma C.4** (linear functional Lipschitz over a bounded cube). *For any $x \in [0, U]^m$ and any $w, w' \in \Delta_m$,*

$$|\langle w, x \rangle - \langle w', x \rangle| \leq U \cdot \|w - w'\|_1.$$

**Proof.**

Compute $\langle w, x \rangle - \langle w', x \rangle = \langle w - w', x \rangle$. By Hölder inequality for $\ell_1/\ell_\infty$ norms, $|\langle w - w', x \rangle| \leq \|w - w'\|_1 \cdot \|x\|_\infty$. Because $x \in [0, U]^m$, we have $\|x\|_\infty \leq U$. This yields the lemma. $\qquad\square$

Now proceed with Theorem 6.7. Let $\mathcal{C}^*$ be as in the theorem, and define $\mathcal{C}_{\text{out}}$ as in the theorem.

**Bounding the support-function gap.** By Lemma C.4 and the definition of support function (Definition 4.5), for any $x \in \mathcal{C}^* \subset [0, U]^m$, $\langle w, x \rangle \leq \langle \bar{w}, x \rangle + U \|w - \bar{w}\|_1$. Taking supremum over $x \in \mathcal{C}^*$ gives

$$h_{\mathcal{C}^*}(w) \leq h_{\mathcal{C}^*}(\bar{w}) + U \|w - \bar{w}\|_1. \tag{C.35}$$

By assumption of Theorem 6.7, for this grid point $\bar{w}$ there exists $\bar{\pi}$ such that

$$h_{\mathcal{C}^*}(\bar{w}) - \langle \bar{w}, v(\bar{\pi}) \rangle \leq \varepsilon_{\mathrm{alg}}. \tag{C.36}$$

Rearrange: $h_{\mathcal{C}^*}(\bar{w}) \leq \langle \bar{w}, v(\bar{\pi}) \rangle + \varepsilon_{\mathrm{alg}}$.

Because $v(\bar{\pi}) \in [0, U]^m$, Lemma C.4 gives $\langle \bar{w}, v(\bar{\pi}) \rangle \leq \langle w, v(\bar{\pi}) \rangle + U \|w - \bar{w}\|_1$. Therefore

$$h_{\mathcal{C}^*}(\bar{w}) \leq \langle w, v(\bar{\pi}) \rangle + \varepsilon_{\mathrm{alg}} + U \|w - \bar{w}\|_1. \tag{C.37}$$

Combining (C.35) and (C.37) yields

$$h_{\mathcal{C}^*}(w) \leq h_{\mathcal{C}^*}(\bar{w}) + U \|w - \bar{w}\|_1 \leq \langle w, v(\bar{\pi}) \rangle + \varepsilon_{\mathrm{alg}} + 2U \|w - \bar{w}\|_1.$$

Since $\|w - \bar{w}\|_1 \leq \eta(\mathcal{W}_L)$, we get

$$h_{\mathcal{C}^*}(w) \leq \langle w, v(\bar{\pi}) \rangle + \varepsilon_{\mathrm{alg}} + 2U \eta(\mathcal{W}_L). \tag{C.38}$$

By definition $\mathcal{C}_{\mathrm{out}} = \mathrm{conv}(\{v(\pi_\ell)\})$ and linearity of the inner product, the support function satisfies $h_{\mathcal{C}_{\mathrm{out}}}(w) = \max_\ell \langle w, v(\pi_\ell) \rangle \geq \langle w, v(\bar{\pi}) \rangle$. Therefore, from (C.38),

$$h_{\mathcal{C}^*}(w) - h_{\mathcal{C}_{\mathrm{out}}}(w) \leq \varepsilon_{\mathrm{alg}} + 2U \eta(\mathcal{W}_L). \tag{C.39}$$

**Converting support error to hypervolume.** Taking supremum over $w \in \Delta_m$ in (C.39) yields $\sup_w (h_{\mathcal{C}^*}(w) - h_{\mathcal{C}_{\mathrm{out}}}(w)) \leq \varepsilon_{\mathrm{alg}} + 2U \eta(\mathcal{W}_L)$. Since $\mathcal{C}_{\mathrm{out}} \subseteq \mathcal{C}^*$, we have $h_{\mathcal{C}_{\mathrm{out}}}(w) \leq h_{\mathcal{C}^*}(w)$ for all $w \in \Delta_m$, hence $\sup_{w \in \Delta_m} |h_{\mathcal{C}^*}(w) - h_{\mathcal{C}_{\mathrm{out}}}(w)| \leq \varepsilon_{\mathrm{alg}} + 2U \eta(\mathcal{W}_L)$. Because $\mathcal{C}^*$ and $\mathcal{C}_{\mathrm{out}}$ are compact convex subsets of $[0, U]^m$, Theorem 4.9 applies to obtain $d_{H,\infty}(D_0(\mathcal{C}^*), D_0(\mathcal{C}_{\mathrm{out}})) \leq \varepsilon_{\mathrm{alg}} + 2U \eta(\mathcal{W}_L)$. Then Theorem 4.10 applies (dominated regions are downward-closed, Lemma C.1) to obtain

$$\mathrm{HV}(\mathcal{C}^*) - \mathrm{HV}(\mathcal{C}_{\mathrm{out}}) = \mathrm{Vol}_m(D_0(\mathcal{C}^*)) - \mathrm{Vol}_m(D_0(\mathcal{C}_{\mathrm{out}})) \leq mU^{m-1}(\varepsilon_{\mathrm{alg}} + 2U \eta(\mathcal{W}_L)).$$

This is exactly the claim of Theorem 6.7. $\qquad\square$

## C.8. Proof of Theorem 6.8 (simplex covering rate)

We prove the two statements (upper and lower). The argument is explicit and self-contained.

**Coordinate chart for the simplex.** Define the chart mapping from $\mathbb{R}^{m-1}$ to $\Delta_m$: for $x \in \mathbb{R}^{m-1}$ with $x_i \geq 0$ and $\sum_{i=1}^{m-1} x_i \leq 1$, define

$$\Psi(x) = \left(x_1, \ldots, x_{m-1}, 1 - \sum_{i=1}^{m-1} x_i\right) \in \Delta_m.$$

Let

$$\Delta_{m-1}^{\mathrm{chart}} = \left\{x \in \mathbb{R}^{m-1} : x_i \geq 0 \text{ and } \sum_{i=1}^{m-1} x_i \leq 1\right\}.$$

Then $\Psi$ is a bijection between $\Delta_{m-1}^{\mathrm{chart}}$ and $\Delta_m$.

For $x, x' \in \Delta_{m-1}^{\mathrm{chart}}$, define $w = \Psi(x)$, $w' = \Psi(x')$. Then:

$$\|w - w'\|_1 = \sum_{i=1}^{m-1} |x_i - x_i'| + \left|\left(1 - \sum_{i=1}^{m-1} x_i\right) - \left(1 - \sum_{i=1}^{m-1} x_i'\right)\right| = \sum_{i=1}^{m-1} |x_i - x_i'| + \left|\sum_{i=1}^{m-1} (x_i' - x_i)\right| \leq 2\|x - x'\|_1. \tag{C.40}$$

Also, trivially $\|x - x'\|_1 \leq \|w - w'\|_1$ because the first $m - 1$ coordinates contribute directly. Therefore:

$$\|x - x'\|_1 \leq \|w - w'\|_1 \leq 2\|x - x'\|_1. \tag{C.41}$$

Thus, covering $\Delta_m$ under $\ell_1$ is equivalent up to a factor 2 to covering $\Delta_{m-1}^{\mathrm{chart}}$ under $\ell_1$.

**Upper bound.** Let $\eta > 0$. Consider the grid in $\mathbb{R}^{m-1}$:

$$G(\eta) = \{x \in \Delta_{m-1}^{\text{chart}} : \text{each } x_i \text{ is a multiple of } \eta\}.$$

For any $x \in \Delta_{m-1}^{\text{chart}}$, define $x_{\text{grid}}$ by rounding each coordinate $x_i$ down to the nearest multiple of $\eta$: $(x_{\text{grid}})_i = \eta \cdot \lfloor x_i/\eta \rfloor$. Then $0 \leq x_i - (x_{\text{grid}})_i < \eta$ for each $i$, hence $\|x - x_{\text{grid}}\|_1 < (m-1)\eta$. Also $x_{\text{grid}}$ remains in $\Delta_{m-1}^{\text{chart}}$ because rounding down cannot increase the sum of coordinates. Therefore $G(\eta)$ is an $(m-1)\eta$-cover of $\Delta_{m-1}^{\text{chart}}$ in $\ell_1$.

By (C.41), the set $\mathcal{W} = \Psi(G(\eta))$ is a cover of $\Delta_m$ with $\ell_1$ radius at most $2(m-1)\eta$. Therefore

$$\eta(\mathcal{W}) \leq 2(m-1)\eta. \tag{C.42}$$

We now bound the cardinality $|G(\eta)|$. Each grid point corresponds to a nonnegative integer vector $(k_1, \ldots, k_{m-1})$ with $x_i = k_i\eta$ and $\sum_i k_i\eta \leq 1$, i.e., $\sum_i k_i \leq \lfloor 1/\eta \rfloor$. The number of such integer vectors is at most $(\lfloor 1/\eta \rfloor + 1)^{m-1}$. Therefore,

$$|G(\eta)| \leq (1/\eta + 1)^{m-1} \leq (2/\eta)^{m-1} \quad \text{for } \eta \leq 1. \tag{C.43}$$

Given a target $L$, first note that if $L < 2^{m-1}$ then the desired upper bound is trivial: since the $\ell_1$ diameter of $\Delta_m$ is 2, any nonempty $\mathcal{W} \subseteq \Delta_m$ satisfies $\eta(\mathcal{W}) \leq 2$, and moreover $L^{-1/(m-1)} \geq 1/2$ when $L \leq 2^{m-1}$, so $2 \leq 4(m-1)L^{-1/(m-1)}$. Hence we may assume $L \geq 2^{m-1}$, so $\eta := 2 \cdot L^{-1/(m-1)} \leq 1$. Choose this $\eta$, for which (C.43) gives $|G(\eta)| \leq L$. Then there exists a set $\mathcal{W}_L$ of size at most $L$ with covering radius at most $2(m-1)\eta \leq 4(m-1)L^{-1/(m-1)}$ by (C.42). This proves the upper bound in Theorem 6.8 with an explicit dimension dependence: the constructed $\mathcal{W}_L$ satisfies $\eta(\mathcal{W}_L) \leq 4(m-1)L^{-1/(m-1)}$. Therefore Theorem 6.8 holds with $C_{\text{cov}} = 4$ in the form $\eta(\mathcal{W}_L) \leq C_{\text{cov}}(m-1)L^{-1/(m-1)}$.

**Lower bound.** We prove that any $\mathcal{W}_L$ must have radius at least on the order of $L^{-1/(m-1)}$.

Work in chart coordinates $\Delta_{m-1}^{\text{chart}}$ with $\ell_1$ metric. Let $B_1(r)$ denote the $\ell_1$ ball in $\mathbb{R}^{m-1}$ of radius $r$: $B_1(r) = \{z \in \mathbb{R}^{m-1} : \|z\|_1 \leq r\}$.

A key fact: the $(m-1)$-dimensional Lebesgue volume of $B_1(r)$ is

$$\text{Vol}_{m-1}(B_1(r)) = \frac{2^{m-1}}{(m-1)!}r^{m-1}. \tag{C.44}$$

This is the standard cross-polytope volume formula.

Also, the volume of $\Delta_{m-1}^{\text{chart}}$ is

$$\text{Vol}_{m-1}(\Delta_{m-1}^{\text{chart}}) = \frac{1}{(m-1)!}. \tag{C.45}$$

Now suppose $\mathcal{W}_L$ is any set of $L$ points in $\Delta_m$ with $\ell_1$ covering radius $\eta(\mathcal{W}_L) = r$. Let $X_L$ be the corresponding set in chart coordinates: $X_L = \Psi^{-1}(\mathcal{W}_L) \subset \Delta_{m-1}^{\text{chart}}$. By (C.41), $X_L$ has covering radius at most $r$ in $\ell_1$ on the chart domain: for any $x \in \Delta_{m-1}^{\text{chart}}$, there exists $x_\ell \in X_L$ with $\|x - x_\ell\|_1 \leq r$ (because otherwise $\Psi(x)$ would be farther than $r$ from all $w_\ell$, contradicting coverage).

Therefore,

$$\Delta_{m-1}^{\text{chart}} \subset \bigcup_{\ell=1}^{L}(x_\ell + B_1(r)). \tag{C.46}$$

Taking volumes and using subadditivity,

$$\text{Vol}_{m-1}(\Delta_{m-1}^{\text{chart}}) \leq \sum_{\ell=1}^{L} \text{Vol}_{m-1}\big((x_\ell + B_1(r)) \cap \Delta_{m-1}^{\text{chart}}\big) \leq \sum_{\ell=1}^{L} \text{Vol}_{m-1}(B_1(r)) = L \cdot \text{Vol}_{m-1}(B_1(r)).$$

The second inequality uses that intersection with $\Delta_{m-1}^{\text{chart}}$ cannot increase volume.

Using (C.44) and (C.45), we obtain

$$\frac{1}{(m-1)!} \leq L \cdot \frac{2^{m-1}}{(m-1)!} \cdot r^{m-1}.$$

Multiply both sides by $(m-1)!$ and rearrange:

$$r^{m-1} \geq 2^{-(m-1)} \cdot \frac{1}{L}, \quad \text{so} \quad r \geq \frac{1}{2} L^{-1/(m-1)}. \tag{C.47}$$

Finally, by (C.41) we have $\|x - x'\|_1 \leq \|w - w'\|_1$ for chart coordinates $x = \Psi^{-1}(w)$. Hence any $\ell_1$–cover of $\Delta_m$ with radius $\eta(\mathcal{W}_L)$ induces an $\ell_1$–cover of $\Delta_{m-1}^{\text{chart}}$ with radius at most $\eta(\mathcal{W}_L)$, implying $\eta(\mathcal{W}_L) \geq r$. Combining with (C.47) yields $\eta(\mathcal{W}_L) \geq \frac{1}{2} L^{-1/(m-1)}$. $\qquad\square$

### C.9. Proof of Proposition 6.9 (regret does not imply HV coverage)

We give a simple two-objective instance in which the online regret can be identically zero under a predictable preference sequence while the deployable set remains far from the Pareto frontier.

Consider an episodic MDP with $H = 1$, a single state $s$, and two deterministic actions $a_1, a_2$. Let $m = 2$ and define deterministic vector rewards $r(s, a_1) = (1, 0)$ and $r(s, a_2) = (0, 1)$. Let the preference be fixed and predictable: $w_k \equiv e_1$ for all $k$. Then the policy that always takes $a_1$ is optimal in every episode and achieves $\text{Reg}(K) = 0$ for all $K$. However, the deployable return set is $\mathcal{C}^* = \text{conv}\{(1, 0), (0, 1)\}$, whose hypervolume in $[0, 1]^2$ equals $1/2$, while the executed set is $\hat{\mathcal{C}}_K = \{(1, 0)\}$ for all $K$ with hypervolume 0. Therefore $\text{HV}(\mathcal{C}^*) - \text{HV}(\hat{\mathcal{C}}_K) = 1/2$, proving the claim with $c_0 = 1/2$. $\qquad\square$

### C.10. Proof of Corollary 6.15 (PFE queries to deployable HV certificate)

Assume the event (1) holds. Fix any $\bar{w} \in \mathcal{W}_L$ and let $\pi_{\bar{w}} := \hat{\pi}(\bar{w})$. By Lemma 4.6 and (1),

$$h_{\mathcal{C}^*}(\bar{w}) = V^*(\bar{w}) \leq V^{\pi_{\bar{w}}}(\bar{w}) + \varepsilon = \langle \bar{w}, v(\pi_{\bar{w}}) \rangle + \varepsilon.$$

Thus the per-weight planning condition in Theorem 6.7 holds with $\varepsilon_{\text{alg}} = \varepsilon$ for the output set $\mathcal{C}_{\text{out}} = \text{conv}(\{v(\pi_{\bar{w}}) : \bar{w} \in \mathcal{W}_L\})$. Applying Theorem 6.7 yields the stated bound. $\qquad\square$

**Estimating $\mathcal{C}_{\text{out}}$ under reward-free feedback.** Corollary 6.15 is a certification statement about the return set induced by the queried policies. In the reward-free protocol, the exploration phase does not observe realized rewards, so computing $\text{HV}(\mathcal{C}_{\text{out}})$ from data typically requires either (i) an additional evaluation phase once rewards are revealed (e.g., rollouts to estimate $v(\pi)$ for queried $\pi$), or (ii) access to an oracle for the mean scalar reward at query time. This practical estimation step is orthogonal to the stability chain, which only requires a uniform bound on the scalarized value errors.

Before proving Theorem 6.2, we make the noise notation explicit. Enumerate episode-steps by $t = (k-1)H + h$. Let $\mathcal{F}_{t-1}$ denote the *pre-reward* filtration at step $t$ (history up to step $t-1$ together with the current $(s_t, a_t)$), consistent with the episode-step filtrations used in Assumption 3.4. For each objective $i \in \{1, \dots, m\}$, define the conditional mean reward

$$\bar{r}_{h,i}(s_t, a_t) := \mathbb{E}[r_{t,i} \mid \mathcal{F}_{t-1}].$$

Define the reward noise component

$$\xi_{t,i} := r_{t,i} - \bar{r}_{h,i}(s_t, a_t).$$

Since $r_{t,i} \in [0, 1]$ almost surely and $\bar{r}_{h,i}(s_t, a_t) \in [0, 1]$ (as a conditional expectation of a $[0, 1]$-valued variable), we have $\xi_{t,i} \in [-1, 1]$ almost surely. Moreover, by the tower property of conditional expectation,

$$\mathbb{E}[\xi_{t,i} \mid \mathcal{F}_{t-1}] = \mathbb{E}[r_{t,i} - \bar{r}_{h,i}(s_t, a_t) \mid \mathcal{F}_{t-1}] = \mathbb{E}[r_{t,i} \mid \mathcal{F}_{t-1}] - \bar{r}_{h,i}(s_t, a_t) = 0.$$

Let $\xi_t := (\xi_{t,1}, \dots, \xi_{t,m})$ denote the $m$-vector reward noise used in Theorem 6.1.

### C.11. Proof of Theorem 6.2 (online regret): modular reward interface and predictable scalarization

This section verifies that the scalar-MDP regret guarantee used by BasePlanner continues to apply under predictable scalarization. In our MORL setting, the scalarized reward in episode $k$ is

$$r_h^{w_k}(s, a) := \langle w_k, r_h(s, a) \rangle,$$

and the preference vector $w_k$ may depend on the past history. Consequently, the mean scalar reward

$$\bar{r}_h^{w_k}(s, a) := \mathbb{E}\big[\langle w_k, r_h(s, a) \rangle \mid \mathcal{F}_{k,0}, s, a\big]$$

is allowed to vary with $k$. We appeal to the modular reward interface established in Appendix B.6: it suffices to construct a high-probability reward confidence event $\mathcal{E}_r$ of the form required there. The reward-modularity requirement for an episodic `LSVI-UCB`-style BasePlanner is verified in Appendix B.6.

**Reward confidence for predictable scalarization via coordinate-wise regression.** It remains to show that our reward estimation procedure produces a reward confidence event $\mathcal{E}_r$ of the form required by Section B.6.

Recall that we observe a vector reward $r_h(s,a) \in \mathbb{R}^m$ with mean $\bar{r}_h(s,a)$ and we define the scalarized mean reward for episode $k$ as

$$\bar{r}_h^{w_k}(s,a) := \langle w_k, \bar{r}_h(s,a) \rangle.$$

In Algorithm 1, we form coordinate-wise linear estimators $\hat{r}_{k,h,i}(s,a)$ for each coordinate $i \in [m]$, and then define the scalar reward estimator

$$\hat{r}_h^{w_k}(s,a) := \sum_{i=1}^m w_{k,i}\, \hat{r}_{k,h,i}(s,a).$$

The corresponding scalar confidence radius is

$$\beta_{k,h}^r(s,a) := \sum_{i=1}^m w_{k,i}\, \beta_{k,h,i}^r(s,a),$$

where $\beta_{k,h,i}^r(s,a)$ is the coordinate-wise confidence radius used in the coordinate-wise regression guarantee.

**Lemma C.5** (Coordinate-wise ridge regression confidence; time-uniform). *Fix a stage $h \in [H]$ and coordinate $i \in [m]$. Assume the (unknown) mean reward is linear in the features: $\bar{r}_{h,i}(s,a) = \langle \phi_h(s,a), \theta_{h,i} \rangle$ with $\|\theta_{h,i}\|_2 \le B_r$ and $\|\phi_h(s,a)\|_2 \le 1$. Let $\lambda \ge 1$ and define the ridge matrix and estimator*

$$\Lambda_{k,h} := \lambda I + \sum_{\tau=1}^{k-1} \phi_h(s_{\tau,h}, a_{\tau,h}) \phi_h(s_{\tau,h}, a_{\tau,h})^\top, \qquad \hat{\theta}_{k,h,i} := \Lambda_{k,h}^{-1} \sum_{\tau=1}^{k-1} \phi_h(s_{\tau,h}, a_{\tau,h})\, r_{\tau,h,i},$$

*and $\hat{r}_{k,h,i}(s,a) := \langle \phi_h(s,a), \hat{\theta}_{k,h,i} \rangle$. If the noise sequence $(r_{\tau,h,i} - \bar{r}_{h,i}(s_{\tau,h}, a_{\tau,h}))_{\tau \ge 1}$ is a bounded martingale difference sequence with $|r_{\tau,h,i} - \bar{r}_{h,i}(\cdot)| \le 1$, then for any $0 < \delta < 1$, with probability at least $1 - \delta$, for all $k \ge 1$ and all $(s,a)$,*

$$\left| \hat{r}_{k,h,i}(s,a) - \bar{r}_{h,i}(s,a) \right| \le \beta_{k,h,i}^r(s,a) := \left( \sqrt{\lambda}\, B_r + \sqrt{\log \frac{\det(\Lambda_{k,h})}{\det(\lambda I)} + 2\log\frac{1}{\delta}} \right) \|\phi_h(s,a)\|_{\Lambda_{k,h}^{-1}}.$$

**Proof.** Fix $(h,i)$. For each episode $\tau \ge 1$, let $\mathcal{F}_{\tau,h}$ be the within-episode $\sigma$-field from Assumption 3.3 (it contains the history up to stage $h-1$ of episode $\tau$ together with the current state–action $(s_{\tau,h}, a_{\tau,h})$, but not the current reward/transition randomness). Define

$$x_\tau := \phi_h(s_{\tau,h}, a_{\tau,h}), \qquad \eta_\tau := r_{\tau,h,i} - \bar{r}_{h,i}(s_{\tau,h}, a_{\tau,h}).$$

Then $x_\tau$ is $\mathcal{F}_{\tau,h}$-measurable (predictable), and $(\eta_\tau)_{\tau \ge 1}$ is a bounded martingale difference sequence with $\mathbb{E}[\eta_\tau \mid \mathcal{F}_{\tau,h}] = 0$ and $|\eta_\tau| \le 1$ almost surely. Moreover, boundedness implies $\eta_\tau$ is conditionally 1-subgaussian given $\mathcal{F}_{\tau,h}$ (Hoeffding's lemma). Thus the assumptions of Theorem B.2 hold for the episode-indexed process $(x_\tau, \eta_\tau)$, and with probability at least $1 - \delta$, for all $k \ge 1$ simultaneously,

$$\Big\| \sum_{\tau=1}^{k-1} x_\tau \eta_\tau \Big\|_{\Lambda_{k,h}^{-1}} \le \sqrt{\log \frac{\det(\Lambda_{k,h})}{\det(\lambda I)} + 2\log\frac{1}{\delta}}. \tag{C.48}$$

Next, expand the ridge estimator error:

$$\hat{\theta}_{k,h,i} - \theta_{h,i} = \Lambda_{k,h}^{-1} \Big( \sum_{\tau=1}^{k-1} x_\tau r_{\tau,h,i} - \Lambda_{k,h} \theta_{h,i} \Big) = \Lambda_{k,h}^{-1} \Big( \sum_{\tau=1}^{k-1} x_\tau \eta_\tau - \lambda \theta_{h,i} \Big).$$

For any $(s,a)$, Cauchy–Schwarz yields

$$\left| \hat{r}_{k,h,i}(s,a) - \bar{r}_{h,i}(s,a) \right| = \left| \langle \phi_h(s,a), \hat{\theta}_{k,h,i} - \theta_{h,i} \rangle \right| \le \|\phi_h(s,a)\|_{\Lambda_{k,h}^{-1}} \Big( \Big\| \sum_{\tau=1}^{k-1} x_\tau \eta_\tau \Big\|_{\Lambda_{k,h}^{-1}} + \sqrt{\lambda}\, \|\theta_{h,i}\|_2 \Big).$$

Combining (C.48) with $\|\theta_{h,i}\|_2 \le B_r$ gives the stated confidence radius. $\qquad \square$

**Lemma C.6** (reward confidence for predictable scalarization via coordinate-wise regression). *With probability at least $1 - \delta_r$, for all $k \in [K]$, all $h \in [H]$, and all $(s, a)$,*

$$\left| \hat{r}_h^{w_k}(s, a) - \bar{r}_h^{w_k}(s, a) \right| \leq \beta_{k,h}^r(s, a).$$

*This is exactly the event $\mathcal{E}_r$ in Section B.6.*

**Proof.** Let

$$\mathcal{E}_{r,\text{coord}} := \left\{ \forall i \in [m], \forall k \in [K], \forall h \in [H], \forall (s, a) : \left| \hat{r}_{k,h,i}(s, a) - \bar{r}_{h,i}(s, a) \right| \leq \beta_{k,h,i}^r(s, a) \right\}.$$

By Lemma C.5 applied to each $(h, i)$ with failure probability $\delta_{h,i} := \delta_r/(Hm)$ (and a union bound over the $Hm$ coordinate-wise regressions), we have $\mathbb{P}(\mathcal{E}_{r,\text{coord}}) \geq 1 - \delta_r$. On $\mathcal{E}_{r,\text{coord}}$, for any $k \in [K]$, $h \in [H]$, and $(s, a)$,

$$\begin{aligned}
\left| \hat{r}_h^{w_k}(s, a) - \bar{r}_h^{w_k}(s, a) \right| &= \left| \sum_{i=1}^{m} w_{k,i} \left( \hat{r}_{k,h,i}(s, a) - \bar{r}_{h,i}(s, a) \right) \right| \\
&\leq \sum_{i=1}^{m} w_{k,i} \left| \hat{r}_{k,h,i}(s, a) - \bar{r}_{h,i}(s, a) \right| \\
&\leq \sum_{i=1}^{m} w_{k,i} \beta_{k,h,i}^r(s, a) = \beta_{k,h}^r(s, a),
\end{aligned}$$

where we used $w_k \in \Delta_m$ so that $w_{k,i} \geq 0$ and $\sum_i w_{k,i} = 1$. This proves Lemma C.6. $\qquad\square$

Note that Lemma C.6 is a deterministic consequence of the simultaneous coordinate-wise event $\mathcal{E}_{r,\text{coord}}$ and the fact that $w_k \in \Delta_m$ has nonnegative coordinates. In particular, the conclusion holds for any (possibly random) preference sequence once $\mathcal{E}_{r,\text{coord}}$ occurs. Predictability of $w_k$ is used elsewhere (e.g., Theorem 6.1) to preserve the martingale structure under scalarization and to ensure $w_k$ is known at the beginning of episode $k$.

**Failure probability allocation and the $\log m$ factor.** We set $\delta_r := \delta/2$ for the reward confidence event $\mathcal{E}_r$. In Lemma C.6, each of the $Hm$ coordinate-wise regressions uses failure probability $\delta_{r,\text{coord}} := \delta_r/(Hm) = \delta/(2Hm)$, which contributes the logarithmic term $\log(1/\delta_{r,\text{coord}}) = \log(2Hm/\delta) = \log(1/\delta) + \log(2H) + \log(m)$. Thus the only dependence on the number of objectives $m$ is an additive $\log(m)$ inside the $\widetilde{O}(\cdot)$ factors. We set $\delta_P := \delta/4$ for the transition-confidence event $\mathcal{E}_P$ and $\delta_{\text{plan}} := \delta/4$ for the planner-analysis event $\mathcal{E}_{\text{plan}}$.

With these allocations, the Modular Reward Interface theorem (Section B.6) guarantees that the regret bound holds with probability at least

$$1 - (\delta_r + \delta_P + \delta_{\text{plan}}) = 1 - \left( \frac{\delta}{2} + \frac{\delta}{4} + \frac{\delta}{4} \right) = 1 - \delta.$$

**Conclusion of the regret proof.** We now combine the pieces.

Let $\mathcal{E}_r$ be the reward confidence event provided by Lemma C.6 with failure probability $\delta_r = \delta/2$. Let $\mathcal{E}_P$ and $\mathcal{E}_{\text{plan}}$ be the transition and planner-analysis success events required by BasePlanner, with failure probabilities $\delta_P = \delta/4$ and $\delta_{\text{plan}} = \delta/4$.

By Lemma C.6,

$$\mathbb{P}(\mathcal{E}_r) \geq 1 - \delta_r = 1 - \delta/2.$$

By the BasePlanner guarantee,

$$\mathbb{P}(\mathcal{E}_P) \geq 1 - \delta_P = 1 - \delta/4, \qquad \mathbb{P}(\mathcal{E}_{\text{plan}}) \geq 1 - \delta_{\text{plan}} = 1 - \delta/4.$$

Define the intersection event $\mathcal{G} := \mathcal{E}_r \cap \mathcal{E}_P \cap \mathcal{E}_{\text{plan}}$. By the union bound,

$$\mathbb{P}(\mathcal{G}) \geq 1 - (\delta_r + \delta_P + \delta_{\text{plan}}) = 1 - \delta.$$

On $\mathcal{G}$, the Modular Reward Interface theorem (Section B.6) applies. Instantiating BasePlanner with `LSVI-UCB` and using its standard regret guarantee for linear MDPs (Jin et al., 2020b), we obtain that, with probability at least $1 - \delta$,

$$\text{Reg}(K) \leq \widetilde{O}\big( d^{3/2} H^2 \sqrt{K} \big).$$

As discussed above, the only dependence on the number of objectives $m$ is through $\delta_{r,\text{coord}} = \delta/(2Hm)$, which contributes an additive $\log(m)$ factor inside the $\widetilde{O}(\cdot)$ notation. This completes the proof of Theorem 6.2. $\qquad\square$

### C.12. Proof of Theorem 6.4 (online regret lower bound)

We reduce to the single-objective regret lower bound for *linear MDPs* (Appendix B.5).

Fix any algorithm $\mathcal{A}$ for the MORL protocol under Assumption 3.3. Consider the special case $m = 2$ and choose a preference sequence that is constant: $w_k = (1, 0)$ for all $k$. This sequence is predictable (Definition 3.5) because it is deterministic.

Define an instance where the second objective is identically zero: $r_{h,2}(s, a) = 0$ always, while the first objective reward $r_{h,1}$ and transitions follow a hard single-objective linear MDP instance. Under this construction, for every policy $\pi$ and every episode $k$, $V^\pi(w_k) = V^\pi((1,0)) = \mathbb{E}[\sum_h r_{h,1}]$, and the optimal value $V^*(w_k)$ is the single-objective optimal value.

Therefore, the MORL regret $\mathrm{Reg}(K)$ equals the single-objective regret for objective 1. Any minimax lower bound for single-objective linear MDPs under the same interaction model applies to this restricted MORL class.

By the external lower bound stated in Theorem B.9 (Appendix B.5), there exists an instance such that $\mathbb{E}[\mathrm{Reg}(K)] \geq c\, d\sqrt{H^3 K}$. This proves Theorem 6.4. □

### C.13. Proof of Theorem 6.10 (PFE decision-optimal)

Theorem 6.10 follows from Theorem B.10 after matching the total-return scale. By Definition 3.2, every scalarized reward has return bounded by $U_{\mathrm{ret}}$. The external theorem is stated for return bound 1; rescaling rewards by $1/U_{\mathrm{ret}}$ rescales all value gaps by the same factor, so an $\varepsilon$-decision-optimal guarantee at return scale $U_{\mathrm{ret}}$ corresponds to accuracy $\varepsilon/U_{\mathrm{ret}}$ in the normalized setting. Since the sample complexity in Theorem B.10 scales as $1/\varepsilon^2$, this introduces a multiplicative factor $U_{\mathrm{ret}}^2$.

The lower bound follows from the lower bound statement in Theorem B.10 by the same rescaling argument.

### C.14. Proof of Theorem 6.13 (explicit-model lower bound)

We prove Theorem 6.13 by a reduction to multinomial distribution estimation at anchors.

**Hard subclass and sample counts.** We consider a one-step anchored instance (horizon $H = 1$) with $|\mathcal{S}| = S_{\mathrm{size}}$ states and $|\mathcal{A}| = d$ actions. The episode starts in a fixed state $s_1$. Choosing action $a = j \in [d]$ draws the terminal state $s_2 \sim P^j \in \Delta(\mathcal{S})$ and ends the episode; rewards are deterministically zero. Define the feature map at the only decision point as $\phi(s_1, j) = e_j \in \mathbb{R}^d$. Then $P(\cdot \mid s_1, j) = P^j$ and every action is an anchor, so Assumption 5.1 holds.

In episode $k$, the algorithm selects $a_k \in [d]$ and observes $X_k \sim P^{a_k}$. Let $N_j := \sum_{k=1}^N \mathbf{1}\{a_k = j\}$ be the (random) number of visits to anchor $j$. Then $\sum_{j=1}^d N_j = N$ deterministically, and conditional on the action sequence $(a_1, \ldots, a_N)$, the observations corresponding to anchor $j$ are $N_j$ i.i.d. samples from $P^j$.

**A constant-probability multinomial lower bound.** We invoke Lemma B.12 (proved in Appendix B.10) with a *constant* failure probability, say $\delta = 13/40$. Since $\log(1/(2\delta))$ is then an absolute constant, Lemma B.12 implies that there exists a universal constant $c_0 > 0$ such that for any estimator $\hat{P}$ based on $n$ i.i.d. samples from an unknown $P \in \Delta(\mathcal{S})$,

$$n < c_0 \cdot \frac{S_{\mathrm{size}} - 1}{\varepsilon_P^2} \quad \implies \quad \exists\, P^* \in \Delta(\mathcal{S}) : \P\big(\|\hat{P} - P^*\|_1 > \varepsilon_P\big) \geq 13/40.$$

**Lifting to $d$ anchors and conclusion.** We now lift the single-anchor lower bound to the $d$-anchor setting using Yao's minimax principle. Fix any (possibly randomized) explicit-model PFE algorithm. By Yao's principle, it suffices to consider a deterministic algorithm against a randomly drawn hard instance.

Let $\{P_v\}_{v \in \{-1, +1\}^M}$ denote the hypercube family constructed in the proof of Lemma B.12 (with the same choice of $M$ and $\alpha$). Consider the following random instance distribution: for each anchor index $j \in [d]$, draw $V^{(j)}$ i.i.d. uniformly from $\{-1, +1\}^M$ and set the (unknown) anchor transition to $P^j := P_{V^{(j)}}$. Run the deterministic algorithm for $N$ episodes and let $N_j$ be the number of visits to anchor $j$.

Let $\bar{N}_j := \mathbb{E}[N_j]$ denote expectation over the random instance draw and the sampled next states. Since $\sum_{j=1}^d N_j = N$ deterministically, we have $\sum_{j=1}^d \bar{N}_j = N$, hence there exists an index $j_0 \in [d]$ such that $\bar{N}_{j_0} \leq N/d$.

We claim that if $N < c_0 \, d(S_{\mathrm{size}} - 1)/\varepsilon_P^2$ (so in particular $\bar{N}_{j_0} < c_0(S_{\mathrm{size}} - 1)/\varepsilon_P^2$), then the algorithm cannot estimate $P^{j_0}$ to $\ell_1$ accuracy $\varepsilon_P$ with success probability exceeding $27/40$ under the above random instance.

To see this, we repeat the Assouad argument from Lemma B.12 for the $j_0$-th anchor, with the only change that the $n$-sample product measure is replaced by the full transcript distribution induced by the (adaptive) choice of anchors. For each $r \in [M]$ and each parameter vector $v = (v^{(1)}, \ldots, v^{(d)})$, let $v^{(j_0, r)}$ be obtained by flipping the $r$-th bit of $v^{(j_0)}$ and denote by $\mathbb{P}_v$ the distribution of the full transcript (actions and next states) under parameter $v$.

By the chain rule for KL divergence and the fact that the two instances differ only when anchor $j_0$ is sampled,

$$\mathrm{KL}(\mathbb{P}_v \,\|\, \mathbb{P}_{v^{(j_0, r)}}) \le \mathbb{E}_v[N_{j_0}] \cdot \mathrm{KL}\big(P_{v^{(j_0)}} \,\|\, P_{(v^{(j_0)})^{(r)}}\big).$$

As in the proof of Lemma B.12, the per-sample KL on the right-hand side is bounded by $16\alpha^2 S_{\mathrm{size}}$, hence Pinsker's inequality yields

$$\mathrm{TV}\big(\mathbb{P}_v, \mathbb{P}_{v^{(j_0, r)}}\big) \le \sqrt{8\alpha^2 S_{\mathrm{size}} \, \mathbb{E}_v[N_{j_0}]}.$$

Averaging over the uniform prior on $v$ and applying Jensen's inequality gives

$$\mathbb{E}_v\big[\mathrm{TV}(\mathbb{P}_v, \mathbb{P}_{v^{(j_0, r)}})\big] \le \sqrt{8\alpha^2 S_{\mathrm{size}} \, \bar{N}_{j_0}} \le 1/8,$$

where the last inequality holds when $\bar{N}_{j_0} \le 1/(512\alpha^2 S_{\mathrm{size}})$, i.e., $\bar{N}_{j_0} < c_0(S_{\mathrm{size}} - 1)/\varepsilon_P^2$.

With this average neighbor TV bound, applying the same Assouad reduction used in Lemma B.12 (from the coordinate-testing argument onward) implies

$$\mathbb{P}\big(\|\widehat{P}^{j_0} - P^{j_0}\|_1 > \varepsilon_P\big) \ge 13/40,$$

where the probability is over the random instance draw and the samples. Consequently, the success probability on the event $\max_{j \in [d]} \|\widehat{P}^j - P^j\|_1 \le \varepsilon_P$ is at most $27/40$.

By Yao's minimax principle, there exists a fixed instance in our class such that the algorithm's success probability is at most $27/40$, which contradicts the definition of explicit-model PFE requiring success probability at least $3/4$ (for $\delta \le 1/4$). Therefore any explicit-model PFE algorithm must use

$$N = \Omega\!\left(\frac{d(S_{\mathrm{size}} - 1)}{\varepsilon_P^2}\right)$$

episodes, completing the proof. $\qquad\square$

## C.15. Proof of Theorem 6.14 (explicit-model PFE upper bound)

We prove Theorem 6.14 by directly invoking the modular argument proved in Appendix B.11.

**Anchor estimation (explicit model).**  Under Assumptions 5.1 and 5.2, we can collect i.i.d. samples from each anchor kernel $P_h^j$ and form the empirical estimate $\hat{P}_h^j$. Appendix B.11 (*Anchor sample complexity*) shows that choosing

$$n_{\mathrm{anc}} = \widetilde{O}\!\left(\frac{S_{\mathrm{size}} - 1}{\varepsilon_P^2}\right)$$

samples per anchor suffices to guarantee $\|\hat{P}_h^j - P_h^j\|_1 \le \varepsilon_P$ for all $(h, j)$ simultaneously with probability at least $1 - \delta$. Since one episode supplies exactly one transition sample per stage, this corresponds to

$$N = d \, n_{\mathrm{anc}} = \widetilde{O}\!\left(\frac{d(S_{\mathrm{size}} - 1)}{\varepsilon_P^2}\right)$$

episodes, which establishes the explicit-model requirement (Definition 6.12).

**Decision optimality conversion.**  The same appendix (*Decision optimality*) proves that on this event, for every preference $w \in \Delta_m$, any policy optimal in the estimated model is $\varepsilon$-decision-optimal in the true environment with $\varepsilon \le 2HU_{\mathrm{ret}}\varepsilon_P$. Setting $\varepsilon_P := \varepsilon/(2HU_{\mathrm{ret}})$ yields

$$N = \widetilde{O}\!\left(\frac{d(S_{\mathrm{size}} - 1)\, H^2 \, U_{\mathrm{ret}}^2}{\varepsilon^2}\right),$$

which is exactly the sample complexity in Theorem 6.14. $\qquad\square$

# D. Additional Discussion, Counterexamples, and Notation

**Purpose and how to navigate.** This appendix collects auxiliary material that is helpful for verification and reproducibility but would interrupt the flow of the main paper.

## D.1. Counterexamples and sanity checks

We provide minimal counterexamples that highlight common failure modes in MORL under adaptive (yet predictable) preferences and in hypervolume-based evaluation.

**Hypervolume is not convexification-invariant.** See Proposition 4.4 in the main text.

**Measurability trap: post-hoc scalarization breaks martingales.** Set $H = 1$, $d = 1$, $m = 2$, and a single action. Let $z_t$ be i.i.d. Bernoulli$(1/2)$. Define the reward vector at time $t$:

$$r_t = (z_t, 1 - z_t).$$

Then the mean reward vector is $(1/2, 1/2)$ and the noise is $\xi_t = r_t - (1/2, 1/2)$.

Let an adversary choose the next preference $w_{t+1}$ as a function of $z_t$:

- if $z_t = 1$, set $w_{t+1} = (1, 0)$,

- if $z_t = 0$, set $w_{t+1} = (0, 1)$.

Then $w_{t+1}$ is measurable w.r.t. the history up to time $t$ (hence predictable for time $t + 1$), but it depends on $\xi_t$. If one retroactively forms the scalarized noise $\langle w_{t+1}, \xi_t \rangle$, it equals $+1/2$ deterministically, so its conditional expectation is not zero. Therefore, any regression that uses *future* $w$ to scalarize *past* noise loses the martingale difference property and invalidates self-normalized concentration arguments.

This is why our algorithm estimates each objective separately and only combines estimates with $w_k$ at query time.

**When hypervolume Lipschitz lower bounds require a local regime.** Let $K = [0, U]^m$ and $K' = [0, U - \varepsilon]^m$. The ratio

$$\frac{\mathrm{Vol}(K) - \mathrm{Vol}(K')}{U^{m-1}\varepsilon} = \frac{1 - (1 - \varepsilon/U)^m}{\varepsilon/U}.$$

If $\varepsilon = U/2$, this ratio is $2(1 - 2^{-m})$, which is at most 2, so it cannot scale like $m$. This shows any linear-in-$m$ lower bound must restrict $\varepsilon$ to be on the order of $U/m$, as in Theorem 6.5.

## D.2. Implementation and computational considerations

**Reward estimator cost.** For each stage $h$:

- maintaining $\Lambda_{k,h}$ requires storing a $d \times d$ matrix;

- maintaining $(b^r_{k,h,i})$ for $i = 1, \ldots, m$ requires $m$ vectors in $\mathbb{R}^d$.

A naive per-step update cost is $O(d^2 + md)$, hence per-episode cost $O(H(d^2 + md))$.

**Base planner cost.** The base planner's computational complexity depends on the chosen instantiation (Appendix B). Rare-switching planners typically reduce the number of expensive global recomputations to $O(dH \log(1 + K/\lambda))$ updates.

**Hypervolume computation.** Given return vectors in $\mathbb{R}^m$, computing the convex hull and its hypervolume can be done in time polynomial in the number of points and exponential in $m$ in the worst case. In MORL practice $m$ is typically small (2–10), making this feasible. Our theory focuses on statistical rates and does not require computing the full Pareto front.

## D.3. Symbol table (single source of truth)

- $m$: number of objectives (Section 3.1)

- $d$: feature dimension (Section 3.1)

- $H$: horizon (Section 3.1)

- $K$: number of episodes (Section 3.1)

- $T$: total steps, $T = K \cdot H$ (Section 3.1)

- $|\mathcal{S}|$: number of states (Section 3.1)

- $|\mathcal{A}|$: number of actions (Section 3.1)

- $w_k$: preference in episode $k$, $w_k \in \Delta_m$, predictable (Definition 3.5)

- $r_h(s, a)$: reward vector in $[0, 1]^m$ (Section 3.2)

- $\bar{r}_h(s, a)$: mean reward vector (Section 3.2)

- $v(\pi)$: expected return vector (Section 3.2)

- $V^\pi(w)$: scalarized value (Section 3.2)

- $V^*(w)$: optimal scalarized value (Section 3.2)

- $U_{\mathrm{ret}}$: total return scale upper bound (Section 3.1)

- $\mathcal{C}^*$: deployable convexified return set $\mathrm{conv}(\mathcal{V}_{\mathrm{det}})$ (Definition 4.1)

- $D_0(X)$: dominated region w.r.t. 0 (Definition 4.3)

- $\mathrm{HV}(X)$: hypervolume of dominated region (Definition 4.3)

- $\mathrm{dist}_\infty, d_{H,\infty}$: metric objects (Section 4)

## D.4. Delta allocations

The online regret bound (Theorem 6.2) is proved by intersecting three high-probability events: reward regression ($\mathcal{E}_r$), transition confidence ($\mathcal{E}_P$), and the base-planner analysis event ($\mathcal{E}_{\mathrm{plan}}$). A concrete (data-independent) failure-probability allocation is:

- set $\delta_r = \delta/2$,

- for each $(h, i)$ use $\delta_{h,i} = \delta_r/(Hm)$ in Theorem B.2 to form $\mathcal{E}_r$,

- set $\delta_P = \delta/4$ for $\mathcal{E}_P$ and $\delta_{\mathrm{plan}} = \delta/4$ for $\mathcal{E}_{\mathrm{plan}}$.

With these choices, a union bound gives $\mathbb{P}(\mathcal{E}_r \cap \mathcal{E}_P \cap \mathcal{E}_{\mathrm{plan}}) \geq 1 - (\delta_r + \delta_P + \delta_{\mathrm{plan}}) = 1 - \delta$.

## D.5. Optional remark: random weights do not improve worst-case covering exponents

Worst-case covering radius of $\Delta_m$ is $\Theta(L^{-1/(m-1)})$ (Theorem 6.8). Randomly drawing $L$ weights (e.g., Dirichlet(1)) typically yields a radius on the order of $(\log L/L)^{1/(m-1)}$ with high probability, which differs only by a logarithmic factor. This remark is not used in any theorem.

