# OpenReview forum: "Near-Minimax Multi-Objective RL under Predictable Adversarial Preferences and Preference-Free Exploration in Linear MDPs"
_ICML.cc/2026/Conference — ICML 2026 regular_

### Official Review · Reviewer_fFDf · 2026-03-05

**Soundness:** 2
**Presentation:** 1
**Significance:** 2
**Originality:** 2
**Overall Recommendation:** 4
**Confidence:** 3

**Summary:**

In this paper, the authors study MORL in finite-horizon linear MDPs. The authors consider two protocols : predictable adversarial preferences revealed before starting episode, and reward-free preference-free exploration. Under these protocols, the paper propose a protocol-safe learning framework that achieves filtration-safe regret bounds with logarithmic dependence on the number of objective $m$, deployable hyper-volume semantics based on convexification of return set, and nearly-minimax sample complexity guarantees for decision-optimal query answering.

**Compliance With Llm Reviewing Policy:**

Affirmed.

**Final Justification:**

As noted in my rebuttal acknowledgement, several of my concerns have been addressed in light of the authors’ responses. However, I remain uncertain about the degree of technical novelty and how the proposed contributions are clearly distinguished from prior work. Therefore, I conclude my evaluation with a weak accept.

**Key Questions For Authors:**

1. In Proposition 6.9, the hypervolume is measured using only the policies that are observed. Since the learner does not observe all possible reward vectors or policies, is it appropriate to evaluate the hypervolume in this manner?

2. From Theorem 6.10 to Corollary 6.15, the procedure appears to explore policies corresponding to many different preference weights. How does this differ from simply sampling random preference weights during exploration?

**Limitations:**

Yes.

**Strengths And Weaknesses:**

Strengths

- The paper explains the problem of post-hoc scalarization: re-scalarizing the past rewards using future weights can break the martingale structure, which makes sense.

- Compared to previous MORL works, the regret bound depends on $\log(m)$, improving upon the $poly(m)$ dependence.

- Convexifying hypervolume regret is reasonable enough to address the deployment problem.

Weaknesses

- Aside from the contributions highlighted in this work, it is difficult to identify significant technical novelty. While improving the dependence from $poly(m)$ to $\log(m)$ and estimating rewards objective-wise are valuable pursuits in MORL, the core idea closely resembles existing approaches in multi-objective bandit problems [1,2,3] where rewards are estimated independently.

- The main ideas of the paper are not presented in a unified manner, which makes the manuscript difficult to follow. The paper simultaneously discusses predictable adversarial preferences, the newly introduced hypervolume evaluation, and linear MDP regret analysis. Although the primary focus appears to be on predictable adversarial preferences, it is difficult to understand how these components are conceptually connected.

- The motivation for studying predictable adversarial preferences, as opposed to random or fixed preferences, is not sufficiently explained in the introduction. While I understand the technical motivation, the paper provides limited discussion of practical scenarios in which such preference sequences would naturally arise.

- The terminology “reward-free” and “preference-free” is potentially confusing. I initially expected these terms to refer to algorithms that do not observe rewards or preferences during learning. However, the learner still observes reward vectors and preference weights, which creates ambiguity.

---
References

[1] S Lu, “Multi-Objective Generalized Linear Bandits” 2019

[2] M Xu, “Pareto Regret Analyses in Multi-objective Multi-armed Bandit” 2023

[3] S Park, “Thompson Sampling for Multi-Objective Linear Contextual Bandit” 2025

---

> ### Author Rebuttal · Authors · 2026-03-31
>
> Thank you for the careful reading. We agree that the manuscript did not make the main storyline explicit enough.
>
> A concise way to read the paper is:
> (M1) predictable preferences ask how to learn safely when the weight can change by episode;
> (M2) hypervolume asks what return set is actually deployable after learning;
> (M3) reward-free PFE asks what must be learned now to answer future preference queries.
> The common principle is that preference is a query-time object.
>
> You raised four questions.
>
> **1. On Proposition 6.9 and “observed-policy HV”.**
>
> Directly: no—Proposition 6.9 is a counterexample about the executed set, not the deployment metric we advocate. Our actual target is the deployable convexified return set $C^* = \mathrm{conv}(V_{\mathrm{det}})$, because episode-start randomization makes the convex hull deployable, and Proposition 4.4 shows that $\mathrm{HV}(X)$ and $\mathrm{HV}(\mathrm{conv}(X))$ can differ. So Proposition 6.9 is meant to show that zero regret on the realized preference sequence does not imply coverage of the deployable frontier.
>
> Plainly, observed-only HV asks what was executed during training, while deployable $\mathrm{HV}(\mathrm{conv}(\cdot))$ asks what can actually be deployed after learning.
>
> In an additional rebuttal-only diagnostic, under a concentrated schedule (`constant_e1`), observed-only evaluation gives `hv_hat = 9.3229`, `hv_gap = 3.7545`, and `worst_support_gap = 1.1290`, whereas oracle queried + lattice cover gives `hv_hat = 13.0760`, `hv_gap = 0.0013`, and `worst_support_gap = 0.0017`. Under a broad schedule (`iid_dirichlet1`), the two nearly coincide (`hv_gap = 0.0013` in both cases). Here `hv_gap` is the remaining deployable-HV deficit, and `worst_support_gap` is the largest utility loss over future preferences; smaller is better for both. So observed-only HV is not always wrong, but it can substantially understate deployable coverage when training preferences are concentrated.
>
> **2. On “many queried weights” versus random preference sampling.**
>
> Directly: no—the weights in Theorem 6.10 to Corollary 6.15 are queried after exploration for certification, not sampled during exploration. The theorem-level object is a finite queried set $W_L$, whose quality is measured by its covering radius $\eta(W_L)$: Theorem 6.7 turns per-query planning error plus covering error into an HV certificate, and Theorem 6.8 gives the simplex covering rate. So the point is not to sample many random preferences during learning, but to choose post-hoc query weights that cover the simplex well under a fixed query budget.
>
> In an additional rebuttal-only comparison, at $N = 500$ and $L = 8$, random Dirichlet queries give `hv_gap = 0.9181 ± 0.7585` and `worst_support_gap = 0.3088 ± 0.3581`, whereas lattice covering gives `hv_gap = 0.4002 ± 0.2995` and `worst_support_gap = 0.1087 ± 0.0585`. So covering reduces the point estimate of the deployable-HV deficit by `0.5179` and the worst-case utility loss by `0.2001`, without any extra interaction.
>
> **3. On novelty.**
>
> We do not claim a new scalar planner. The new technical point is the protocol-safe reward interface under predictable preferences. If one learns only the scalar reward $\langle w_k, r \rangle$ online, then the target itself changes with $w_k$. If one learns reward coordinates first and combines them with $w_k$ only at episode start, the learned object stays stable and scalarization happens only at query time. This is what preserves the martingale/confidence structure.
>
> This is also the sharp difference from multi-objective bandits: there, objective-wise estimation is often enough because there is no learned transition model or Bellman confidence transfer. Here the issue is filtration-safe confidence transfer under adaptive predictable weights while transitions are learned; naive post-hoc scalarization breaks exactly that step.
>
> Figure 4 isolates this point: the feature map and optimistic planner are the same, but the baseline learns only $\langle w_k, r \rangle$ online. So the difference is the reward interface itself.
>
> **4. On motivation and terminology.**
>
> Predictable preferences are meant to model an online system facing a sequence of users with different utilities: the next user’s weight can depend on past outcomes, but is fixed before the current episode starts. “Reward-free / preference-free” refers only to the exploration stage in Definition 3.6: during exploration, no rewards are observed; after exploration, a queried preference reveals the corresponding mean scalar reward and must be answered without further interaction. It does not mean that every protocol in the paper ignores rewards or preferences.
>
> We hope this clarifies the intended storyline: M1 studies protocol-safe online learning, M2 identifies the deployment-consistent evaluation object, and M3 shows what reward-free exploration must learn to support future preference queries.

---

> > ### Author Rebuttal · Reviewer_fFDf · 2026-04-03
> >
> > I understand the main point of this paper, especially the difference between queried weights and random preference sampling. I will raise my score since my main questions have been resolved, but I still remain uncertain about this particular aspect of novelty, as how this perspective advances beyond existing multi-objective RL literature.

---

> > > ### Author Response · Authors · 2026-04-03
> > >
> > > Thank you for the follow-up. After your feedback, we now understand your concern more clearly. In our main line $M1$, the question is how online learning should be done when the preference can change by episode. The discussion below focuses on $M1$, since this is the part most directly related to your novelty question, but the paper also has $M2$ and $M3$.
> > >
> > > Our paper's claim has never been that we invented the high-level idea of estimating each reward coordinate separately. Our claim is that, in our MORL setting, this idea cannot be used directly. The difficult part is how to make it work when (i) preferences can change from episode to episode, (ii) transitions are unknown, and (iii) decisions are multi-step rather than one-shot.
> > >
> > > A small two-step example makes this concrete. Suppose the two objectives are safety and speed. In step 1, action $A$ reaches a "safe" region $g$ with probability $0.8$ and a "risky" region $b$ with probability $0.2$; action $B$ does the opposite. In step 2, at $g$ we can choose $\mathrm{conservative}=(0.9,0.2)$ or $\mathrm{aggressive}=(0.7,0.8)$, and at $b$ we can choose $\mathrm{conservative}=(0.4,0.1)$ or $\mathrm{aggressive}=(0.1,1.0)$.
> > >
> > > If $w=(0.9,0.1)$, then at $g$ we prefer conservative since $0.9\cdot 0.9 + 0.1\cdot 0.2 = 0.83$, whereas aggressive gives $0.71$. At $b$ we also prefer conservative since $0.9\cdot 0.4 + 0.1\cdot 0.1 = 0.37$, whereas aggressive gives $0.19$. So the step-1 values are
> > > $A: 0.8\cdot 0.83 + 0.2\cdot 0.37 = 0.738$
> > > and
> > > $B: 0.2\cdot 0.83 + 0.8\cdot 0.37 = 0.462$,
> > > so $A$ is optimal.
> > >
> > > If $w=(0.1,0.9)$, then at $g$ we prefer aggressive since $0.1\cdot 0.7 + 0.9\cdot 0.8 = 0.79$, whereas conservative gives $0.27$. At $b$ we also prefer aggressive since $0.1\cdot 0.1 + 0.9\cdot 1.0 = 0.91$, whereas conservative gives $0.13$. So the step-1 values become
> > > $A: 0.8\cdot 0.79 + 0.2\cdot 0.91 = 0.814$
> > > and
> > > $B: 0.2\cdot 0.79 + 0.8\cdot 0.91 = 0.886$,
> > > so $B$ is now optimal.
> > >
> > > This is the key difference from the cited multi-objective bandit papers. There, one round is a single arm pull with an immediate vector reward. Here, one action changes future states, and the current preference changes which later action is optimal. So in MORL, "estimate each objective separately" is only the beginning; one must also specify how that estimate is connected to multi-step planning.
> > >
> > > Once we follow this seemingly simple idea all the way through, three concrete pitfalls appear.
> > >
> > > **(1) We cannot use a future preference to re-scalarize past noisy rewards.**
> > > Appendix D.1 gives a counterexample where the next preference is chosen from the previous outcome; if one then uses that later weight to re-scalarize the earlier noise, the resulting "noise" is no longer mean-zero. So past data cannot be rewritten using later preferences.
> > >
> > > **(2) Once the preference changes, previously computed values or $Q$-functions may no longer be valid.**
> > > In the two-step example above, the optimal first action flips from $A$ to $B$ when the preference changes. So it is not enough to "learn the coordinates once" and keep using old values. One must inject the current preference only at the start of the current episode and replan for that episode. This is exactly the role of Algorithm 1 and the modular planner interface in Appendix B.6.
> > >
> > > **(3) Discretizing the preference simplex quickly explodes.**
> > > A natural idea is to replace the continuous preference space by a finite grid, so that one only needs to solve finitely many scalarized problems. Theorem 6.8 formalizes this curse, and Appendix A.4/Table 1 gives concrete numbers for the simplex-lattice size
> > > $L=\binom{q+m-1}{m-1}$.
> > > Already $m=10,q=10$ gives $92{,}378$ weights; $m=20,q=10$ gives $20{,}030{,}010$; and $m=100,q=10$ gives $42{,}634{,}215{,}112{,}710$, which is infeasible.
> > >
> > > This is why our $M1$ contribution is not just "split the vector reward." Our $M1$ contribution is a protocol-safe reward interface for predictable preferences: learn stable reward coordinates from vector feedback, introduce the current preference only at episode start, and then pass the resulting scalar reward oracle to an episodic optimistic planner. Figure 4 was designed as a controlled comparison: same feature map, same optimistic planner, only the reward interface changes.
> > >
> > > The paper also has two other main lines. $M2$ gives the deployment-consistent $HV(\mathrm{conv}(\cdot))$ semantics. $M3$ gives the reward-free PFE separation between decision-optimal query answering and explicit model recovery.
> > >
> > > So our point is not that the high-level intuition "separate the objectives" is new. Our point is that, in this MORL setting, using that intuition correctly is not obvious. One must avoid re-scalarizing past noise with future weights, replan when preferences change, and avoid weight-space explosion. Our paper's contribution is to solve these issues in a unified theory for online learning ($M1$), deployment-consistent evaluation ($M2$), and reward-free future-query support ($M3$).

---

### Official Review · Reviewer_xjEY · 2026-03-10

**Soundness:** 3
**Presentation:** 2
**Significance:** 3
**Originality:** 3
**Overall Recommendation:** 4
**Confidence:** 2

**Summary:**

This paper studies multi-objective reinforcement learning in finite-horizon linear MDPs when preferences are not fixed in advance. It considers two protocols: (i) predictable adversarial preferences revealed before each episode, and (ii) reward-free preference-free exploration, where only transitions are observed during exploration and arbitrary preference queries must be answered later. The main idea is to avoid protocol-unsafe post-hoc scalarization by learning reward coordinates separately and scalarizing only at query time. The paper also proposes evaluating deployable performance via the hypervolume of the convex hull of achievable return vectors, and gives near-minimax guarantees for online regret and preference-free query answering, plus a separation between decision-optimal query answering and explicit transition-model recovery.

**Compliance With Llm Reviewing Policy:**

Affirmed.

**Final Justification:**

The authors' rebuttal effectively addressed my concern. Especially, the clear explanation on the necessity of the vector feedback helped me well understood the scope of the paper and its reasonability. The additional experiments was also helpful to strengthen my belief on the effectiveness of the algorithm. I am happy to recommend its acceptance.

**Key Questions For Authors:**

1. How essential is vector reward feedback for the main online guarantee? Would any part of the approach extend to scalar-only feedback?
2. Can the authors provide more intuition for why decision-optimal preference-query answering is strictly easier than explicit model recovery?
3. How important is the deployable hypervolume perspective to the algorithm itself, versus mainly to evaluation and certification?
4. Could the authors strengthen the empirical section with more direct comparisons to standard MORL baselines under changing preferences?

**Limitations:**

yes

**Strengths And Weaknesses:**

**Strengths**

- The paper addresses an important MORL setting where preferences may change online or be specified only after data collection.
- The filtration-safety issue is meaningful and well-motivated; the coordinate-wise regression + query-time scalarization interface is a clean fix.
- The theoretical results are strong: near-minimax regret in linear MDPs, logarithmic dependence on the number of objectives, and a sharp PFE separation result.
- The deployable hypervolume viewpoint through $HV(\mathrm{conv}(\cdot))$ is conceptually interesting and likely useful beyond this paper.
- Overall, the authors explore a broad aspect of MORL under preference uncertainty, spanning online learning, evaluation, and reward-free exploration.
- The authors explore a central concept: preferences should be treated as query-time objects rather than baked into past stochastic rewards.

**Weaknesses**

- The paper is ambitious and somewhat dense; the combination of protocol safety, hypervolume semantics, and PFE makes the narrative heavy.
- The strongest guarantees rely on linear MDPs, known features, and vector feedback, which may limit applicability.
- The empirical section appears supportive but not especially strong relative to the breadth of the theory.
- The practical consequences of the convex-hull hypervolume metric could be better explained, especially when policy randomization is not realistic.

---

> ### Author Rebuttal · Authors · 2026-03-31
>
> Thank you for the careful reading and the concrete questions. We agree that the roles of the three parts can be made clearer.
>
> A concise way to read it is:
> (M1) predictable preferences ask how to learn safely when the weight can change from episode to episode;
> (M2) deployable hypervolume asks what return set is actually deployable after learning;
> (M3) reward-free PFE asks what must be learned now in order to answer future preference queries.
> The common principle is that preference should be treated as a query-time object.
>
> You raised four questions.
>
> **1. How essential is vector reward feedback for the main online guarantee? Would any part extend to scalar-only feedback?**
>
> For the strongest online guarantee: yes, vector reward feedback is essential in the current paper. This is already explicit in Assumption 3.3, and the Limitations section states that extending filtration-safe learning to scalar-only feedback while retaining the same logarithmic dependence on m is open.
>
> The intuition is simple. Our interface first estimates a stable object — the reward coordinates — from past data, and only then combines them with the current episode’s preference at episode start. With scalar-only feedback, the learner directly observes only a changing projection $\langle w_k, r \rangle$, not the underlying coordinate vector, so the same confidence construction is not currently available in our analysis.
>
> Figure 4 probes exactly this: against a feature-aware scalar baseline with the same feature map and optimistic planner, the changed ingredient is the reward interface. This is why we view Figure 4 as evidence that vector-style coordinate estimation is the key practical ingredient in our current theorem setting.
>
> **2. Why is decision-optimal preference-query answering strictly easier than explicit model recovery?**
>
> Because choosing a near-best action is weaker than reconstructing the whole environment model. “Pick a good route” is easier than “reconstruct the whole road network.”
>
> This is also how Section 6.3 is organized. Theorem 6.10 asks only for uniform near-optimal query answering over future preferences. In contrast, Definition 6.12 and Theorems 6.13–6.14 ask for the stronger requirement of recovering the explicit model itself (the base kernels in the anchored branch) to TV accuracy. Since explicit model output is strictly stronger than decision quality, it necessarily carries the extra model-size dependence.
>
> **3. How important is the deployable hypervolume perspective to the algorithm itself, versus mainly to evaluation and certification?**
>
> It is mainly an evaluation/certification layer rather than the optimization target of the online algorithm itself. Algorithm 1 is the online interaction algorithm for M1. Section 4 plays a different role: it provides the stability chain that turns scalar planning errors into deployable hypervolume certificates. In other words, the algorithm learns through scalarized planning, while $HV(\operatorname{conv}(\cdot))$ tells us how to evaluate the deployable trade-off set that learning makes available.
>
> If two corner policies are both deployable, then episode-start randomization can also deploy the intermediate trade-offs between them, even if those intermediate points were never executed as single deterministic policies.
>
> You also asked about applications where policy randomization may not be realistic. We agree this matters. Our contribution is not that $HV(\operatorname{conv}(\cdot))$ is universally the right metric; it is that once the deployment interface allows episode-start randomization, $HV(\operatorname{conv}(\cdot))$ is the deployment-consistent one, and Proposition 4.4 shows that $HV(X)$ and $HV(\operatorname{conv}(X))$ can differ materially. If an application forbids randomization, then $HV(X)$ or another non-convex operational metric would indeed be more appropriate. Our theory is explicitly scoped to the randomization-allowed deployment protocol.
>
> **4. Could the empirical section include more direct comparisons to standard MORL baselines under changing preferences?**
>
> We agree that this is helpful. In an additional rebuttal-only benchmark on MO-Gymnasium fishwood-v0 under the same changing-preference evaluation, our method achieved scalarized return / deployable $HV(\operatorname{conv}(\cdot))$ of 91.007 ± 3.438 / 2132.83 ± 427.43, compared with 83.473 ± 19.641 / 881.81 ± 836.97 for Envelope and 30.627 ± 34.626 / 211.77 ± 373.49 for PQL. In a supporting replicate, it remained highest at 93.273 ± 1.452 / 1937.23 ± 606.14, versus 92.993 ± 1.529 / 1814.91 ± 607.31 for Envelope and 61.573 ± 17.666 / 1106.31 ± 623.80 for PQL.
>
> We hope this clarifies the division of roles in the paper: vector feedback is essential for the strongest online theorem; the PFE separation is about decision quality versus explicit model output; $HV(\operatorname{conv}(\cdot))$ is primarily the deployment-consistent evaluation layer; and we now add a direct standard-benchmark comparison.

---

> > ### Author Rebuttal · Reviewer_xjEY · 2026-04-02
> >
> > Thank you very much for the clarification with the further experiments.
> > The clarification, especially the additional experiments, resolved my concern.
> > I will maintain my current positive evaluation.

---

> > > ### Author Response · Authors · 2026-04-06
> > >
> > > Thank you very much for the helpful follow-up and for the careful reading. We are glad that our rebuttal helped clarify the main points. In the final version, we will revise the presentation to address the issues you raised more clearly, especially by improving readability and reducing the amount of effort needed for a first pass through the paper. In particular, we will make the motivation, scope, and empirical takeaways more direct, and we will further simplify the exposition so that the core message is easier to follow.

---

### Official Review · Reviewer_BUBh · 2026-03-12

**Soundness:** 3
**Presentation:** 3
**Significance:** 3
**Originality:** 3
**Overall Recommendation:** 4
**Confidence:** 4

**Summary:**

This paper studies multi-objective reinforcement learning (MORL) in linear MDPs under two settings: predictable adversarial preferences and reward-free preference-free exploration. The authors propose a protocol-safe framework that performs coordinate-wise reward estimation and scalarization at query time. The paper establishes near-minimax regret guarantees for predictable preferences and provides sample complexity results for preference-free exploration together with hypervolume-based evaluation guarantees.

**Compliance With Llm Reviewing Policy:**

Affirmed.

**Final Justification:**

The rebuttal addressed my concerns, and I will maintain my original score of weak accept.

**Key Questions For Authors:**

1. The regret bound matches the standard near-minimax rate for linear MDPs. Could the authors clarify which aspects of the algorithm design or analysis are essential for achieving this result in the multi-objective setting with predictable preferences? In particular, it would be helpful to provide a proof sketch highlighting the main new techniques.

2. The theoretical results rely on the linear MDP assumption with a known feature map. Could the authors discuss whether the proposed protocol-safe reward interface could be combined with other function approximation settings (e.g., general function approximation), and what technical challenges might arise in extending the analysis beyond linear MDPs?

**Limitations:**

The experiments focus mainly on synthetic environments designed to validate theoretical results. Do the authors expect the proposed framework to provide practical advantages over existing MORL approaches in more realistic environments? Additional discussion or experiments could help clarify the practical implications of the method.

**Strengths And Weaknesses:**

Strengths

1. The paper establishes near-minimax regret guarantees for MORL in linear MDPs under predictable adversarial preferences, together with matching lower bounds up to logarithmic factors.

2. The work studies both online learning with predictable preferences and reward-free preference-free exploration within a unified framework.

Weaknesses

1. While the paper establishes strong theoretical guarantees, it is sometimes difficult to clearly identify which parts of the analysis are new
compared with existing regret analyses for linear MDPs. A clearer explanation or proof sketch highlighting the main new technical ingredients would help improve readability.

2. The experimental evaluation is conducted only on synthetic environments with simulator access and exact dynamic programming evaluation. It would strengthen the paper to include experiments in more realistic or benchmark MORL environments to demonstrate the practical benefits of the proposed framework.

---

> ### Author Rebuttal · Authors · 2026-03-31
>
> Thank you for the thoughtful questions. We agree that two things should be made clearer: what is genuinely new in the predictable-preference MORL setting, and whether the method has practical value beyond our synthetic anchored family.
>
> A concise way to read the paper is:
> (M1) predictable preferences ask how to learn safely when the weight can change from episode to episode;
> (M2) deployable hypervolume asks what return set is actually deployable after learning;
> (M3) reward-free PFE asks what must be learned now in order to answer future preference queries.
> The common principle is that preference should be treated as a query-time object, not baked into past stochastic rewards.
>
> You asked two technical questions and one practical one. We address them in that order.
>
> **1. Which ingredients are actually essential for the near-minimax online result?**
>
> The key new ingredient is not a new scalar planner but a protocol-safe reward interface that keeps confidence estimates valid when the weight changes over episodes and may depend on past data. If episode 1 uses $w=(1,0)$ and episode 2 uses $w=(0,1)$, then a scalar-only learner is chasing a target that changes with the weight; a coordinate-wise learner is estimating the same underlying reward vector and only reweights it at episode start.
>
> A short proof sketch is:
> (i) for each objective coordinate, estimate rewards separately so each noise process stays adapted to the episode-step filtration;
> (ii) when $w_k$ is revealed before episode $k$, combine those coordinate estimates only at episode start to form the scalar reward model for that episode;
> (iii) this preserves the confidence transfer needed by an optimistic linear-MDP planner, with only logarithmic dependence on m from the coordinate-wise confidence bookkeeping;
> (iv) plug this reward interface into a standard episodic optimistic planner that recomputes value functions every episode; this yields the paper's general online-regret guarantee in Theorem 6.2, and when the base planner is instantiated with an episodic near-minimax planner that recomputes each episode, it recovers Corollary 6.3;
> (v) the lower bound is a single-objective embedding: keep only one nonzero objective and choose a constant predictable preference, so the problem reduces to the standard single-objective linear-MDP lower bound.
>
> So we do not claim a new scalar planner. The new technical point is the protocol-safe reward interface that lets a standard scalar planner remain valid under predictable changing preferences.
>
> This is also what Figure 4 in the submitted paper tests: it compares our method against a feature-aware scalar baseline with the same feature map and the same optimistic planner, but which learns only the scalarized reward $\langle w_k, r \rangle$ online. Because planner and features are matched, the difference is the reward interface itself.
>
> **2. Could the same interface extend beyond linear MDPs?**
>
> We view the interface as modular. In principle, the same idea can be paired with richer function-approximation backends. The technical difficulty is no longer scalarization itself; it is maintaining valid coordinate-wise uncertainty estimates and optimism after query-time aggregation under adaptive preferences. Linear MDPs with known features are the clean setting where this can be proved end-to-end, which is why the current theorem-level guarantee is intentionally stated there.
>
> **3. Does the framework have practical value beyond the synthetic anchored family?**
>
> We agree this is important. In an additional rebuttal-only benchmark on MO-Gymnasium fishwood-v0, under the same changing-preference evaluation and with the same scalarized-return / deployable $HV(\operatorname{conv}(\cdot))$ metrics, our method obtains 91.007 ± 3.438 / 2132.83 ± 427.43, compared with 83.473 ± 19.641 / 881.81 ± 836.97 for Envelope and 30.627 ± 34.626 / 211.77 ± 373.49 for PQL. In a supporting replicate, it remains highest with 93.273 ± 1.452 / 1937.23 ± 606.14, compared with 92.993 ± 1.529 / 1814.91 ± 607.31 for Envelope and 61.573 ± 17.666 / 1106.31 ± 623.80 for PQL.
>
> Here scalarized return means the utility achieved under changing preferences, and $HV(\operatorname{conv}(\cdot))$ measures the size of the trade-off menu that can actually be deployed after learning when episode-start randomization is allowed; larger is better for both. We do not present this as a universal benchmark-superiority claim, but it does show that the protocol-safe approach is practically viable beyond our anchored synthetic family.
>
> We hope this makes the paper’s contribution easier to identify: the essential new ingredient is the filtration-safe reward interface under changing preferences, and the additional benchmark result shows that the approach is not purely theorem-facing.

---

> > ### Author Rebuttal · Reviewer_BUBh · 2026-04-03
> >
> > I thank the authors for addressing my concern. I will keep my positive score.

---

> > > ### Author Response · Authors · 2026-04-06
> > >
> > > Thank you very much for the helpful follow-up and for the careful reading. We are glad that our rebuttal helped clarify the main points. In the final version, we will revise the presentation to address the issues you raised more clearly, especially by improving readability and reducing the amount of effort needed for a first pass through the paper. In particular, we will make the motivation, scope, and empirical takeaways more direct, and we will further simplify the exposition so that the core message is easier to follow.

---

### Official Review · Reviewer_32Ug · 2026-03-12

**Soundness:** 3
**Presentation:** 3
**Significance:** 3
**Originality:** 3
**Overall Recommendation:** 4
**Confidence:** 4

**Summary:**

This paper focuses on multi-objective RL in linear MDPs under dynamic preferences. This paper proposes a protocol-safe reward interface through coordinate wise regression to ensure statistical safety. In theoretical analysis, this paper proves near minimax regret bounds and also analyzes reward-free exploration. Synthetic experiments are conducted to validate the proposed method’s efficiency and performance across different preference protocols.

**Compliance With Llm Reviewing Policy:**

Affirmed.

**Final Justification:**

I have read the authors' rebuttal, and it effectively addresses my previous concerns. The additional experiments on anchor violation $(\rho>0)$ and the $d=10$ non-anchored study convince me that the separation between decision-optimal PFE and explicit-model recovery is a fundamental property. Furthermore, the clarification that the $U^{m-1}$ factor is a worst-case sharp limit of the hypervolume metric itself is theoretically sound. Therefore, I decide to maintain my postive evaluation and raise my confidence.

**Key Questions For Authors:**

Main questions (the responses will likely change my evaluation):

1. Based on Weakness 1, could you clarify the necessity of the anchor assumptions (5.1 and 5.2)? Does the information theoretic separation in Section 6.3 still hold without these assumptions? I would like to know if this result is mainly a byproduct of the anchored structure.

2. Based on Weakness 2, I am interested in seeing how the performance changes when the feature dimension $d$ is increased or when the anchor assumptions are slightly violated. This would help validate the value of the protocol-safe wrapper beyond synthetic setup.

3. In Theorem 6.7, the hypervolume gap has the order of $U^{m-1}$, which is exponential with respect to $m$. Is there any way to provide a tighter bound or can you discuss the challenge to reduce this in analysis?

**Limitations:**

yes

**Strengths And Weaknesses:**

Strengths:

1. This paper addresses Pitfall P1, where rescalarizing past rewards with future weights could breaks structure ofmartingale. This paper proposes a filtration safe reward interface through the coordinatewise regression. In this design, w-net discretization can be avoided and this paper provides near minimax regret. The regret bound is tighter with respect to $m$.

2. The proposed theoretical framework for hypervolume (M2) is rigorous. This paper gives a stability chain taht connects support function error to hypervolume error via Hausdorff distance.

Weaknesses:

1. My main concern lies on the reliance on Assumption 5.1 and Assumption 5.2 for the results in section6.3. These assumptions require that there exist specific state-action pairs that can represent the base kernels of the MDP. I think it is a bit strong to have these assumptions, which may not hold in general linear MDPs. A discussion on these assumptions are necessary.

2. The experiments is limited to a synthetic anchored simplex-mixture benchmark. I would like to see how the method performs when the anchor assumptions are relaxed or when the feature dimension $d$ is high in linear MDP.

---

> ### Author Rebuttal · Authors · 2026-03-31
>
> Thank you for these questions; they go directly to the scope of Section 6.3.
>
> You asked three things: (1) how necessary Assumptions 5.1 and 5.2 are, and whether the Section 6.3 separation still holds without them; (2) whether the protocol-safe wrapper still has value when anchors are relaxed; and (3) whether the $U^{m-1}$ factor in the hypervolume bound can be tightened. We answer each point directly.
>
> **1. On Assumptions 5.1 / 5.2 and whether the separation is “just an anchor artifact”.**
>
> Directly: the current explicit-model theorem is anchored, but the separation is not merely an anchor artifact. The theorem-level explicit-model statement is scoped to the anchored identifiable branch. Remark 3.7 introduces the anchored simplex-mixture structure only for the explicit-model branch in Section 6.3; Definition 6.12 and Theorems 6.13–6.14 are stated only under Assumptions 5.1–5.2; by contrast, Theorem 6.10 (decision-optimal PFE) is stated for the general reward-free PFE protocol. So we do not claim the same base-kernel recovery theorem without anchors.
>
> Assumptions 5.1 and 5.2 also play different roles. Assumption 5.1 is the substantive identifiability condition: it makes the base kernels $\{P_h^j\}$ into well-defined explicit objects. Assumption 5.2 mainly simplifies the explicit-model upper bound. It is not the source of the lower-bound hardness: in Appendix C.14, the hard instance is one-step and choosing action $j$ directly samples from $P_j$, so the lower-bound mechanism already works under ordinary interaction.
>
> The separation itself is not merely an anchor artifact. Anchors make the explicit object canonical; they do not create the gap. Without anchors, recovering the base kernels themselves is no longer the right stronger target, because $\{P_h^j\}$ are not canonical. The right anchor-free stronger target is uniform recovery of the full transition map: $\max_{h,s,a}\|\widehat P_h(\cdot\mid s,a)-P_h(\cdot\mid s,a)\|_1 \le \varepsilon_P$. Any algorithm that achieves this on all linear MDPs must also succeed on the anchored subclass used in our lower bound. On that subclass, uniform transition-map recovery contains recovery of the $d$ anchor kernels as a special case, so the same anchored-subclass argument gives $\Omega(d(|S|-1)/\varepsilon_P^2)$ scaling for this stronger anchor-free target as well. Thus the decide-vs-fully-recover separation is broader than the anchored parameterization.
>
> Plainly: choosing a near-best action for any queried preference is like picking a good route for any destination; explicit-model recovery is like reconstructing the whole road network. The second task is strictly stronger.
>
> In an additional rebuttal-only robustness study, at $N=5000$, the explicit-model recovery error increases from $0.1399 \pm 0.0081$ at $\rho=0$ to $0.1668 \pm 0.0070$ at $\rho=0.1$ and $0.2112 \pm 0.0106$ at $\rho=0.2$. In contrast, the decision-quality gap increases only from $0.000329 \pm 0.000043$ to $0.000465 \pm 0.000107$ and $0.000509 \pm 0.000103$. So mild anchor violation hurts “recover the exact model” much more than “choose a near-best action.”
>
> **2. On relaxing anchors / non-anchor robustness.**
>
> In an additional rebuttal-only non-anchor check at $d=10$, our method still improves over the scalar-only baseline on a linear-MDP family with no exact anchor pairs: final cumulative regret is $312.71 \pm 9.34$ for ours versus $339.20 \pm 13.37$ for scalar-only, with paired delta $-26.49$ and 95% CI $[-35.62,-17.36]$. The late-stage regret shows the same pattern: the mean regret over the last 50 episodes is $0.1744 \pm 0.0221$ versus $0.2384 \pm 0.0303$, with paired delta $-0.0640$ and 95% CI $[-0.1024,-0.0257]$. Here cumulative regret is the total scalar utility lost along the changing-preference sequence, so smaller is better.
>
> **3. On the $U^{m-1}$ factor in Theorem 6.7.**
>
> At this level of generality, $mU^{m-1}$ is the optimal universal Lipschitz coefficient, not a proof artifact. Theorem 4.10 gives the upper bound. For $K=[0,U]^m$ and $K_\varepsilon=[0,U-\varepsilon]^m$ in Appendix C.6, $d_{H,\infty}(K,K_\varepsilon)=\varepsilon$ and $(HV(K)-HV(K_\varepsilon))/\varepsilon=(U^m-(U-\varepsilon)^m)/\varepsilon \to mU^{m-1}$ as $\varepsilon\to0$. Hence no universal improvement in the dependence on $U$ or $m$ is possible without extra geometric structure.
>
> Theorem 6.5 gives the finite-$\varepsilon$ version: for $\varepsilon\le U/m$, the same box family already yields the lower bound $(m/2)U^{m-1}\varepsilon$. So $mU^{m-1}$ itself is worst-case sharp.
>
> We hope this clarifies the intended scope: the current base-kernel theorem is anchored; the broader decide-vs-fully-recover separation is not merely an anchor artifact; and the $U^{m-1}$ dependence is worst-case sharp.

---

> > ### Author Rebuttal · Reviewer_32Ug · 2026-04-04
> >
> > Thank you to the author for the detailed response. All of my questions have been resolved. I've decided to maintain my positive rating and increase my confidence.

---

> > > ### Author Response · Authors · 2026-04-06
> > >
> > > Thank you very much for the helpful follow-up and for the careful reading. We are glad that our rebuttal helped clarify the main points. In the final version, we will revise the presentation to address the issues you raised more clearly, especially by improving readability and reducing the amount of effort needed for a first pass through the paper. In particular, we will make the motivation, scope, and empirical takeaways more direct, and we will further simplify the exposition so that the core message is easier to follow.

---

### Decision · Program_Chairs · 2026-04-30

**Decision:**

Accept (regular)

**Comment:**

This work studies multi-objective reinforcement learning under adversarial preference vectors. The authors consider two different adversarial protocols in the linear MDP setting and show that, under both, near-optimal sample complexity can be achieved. They also provide experimental results across several settings. I agree with the reviewers that this paper makes a solid contribution to the MORL literature. My main concern is that the presentation is quite dense, which makes the core contribution difficult to distill. In addition, the paper would benefit from a clearer discussion of its technical novelty relative to prior work, particularly Wu et al.